# ENZYMEFLOW: GENERATING REACTION-SPECIFIC ENZYME CATALYTIC POCKETS THROUGH FLOW MATCHING AND CO-EVOLUTIONARY DYNAMICS

## ABSTRACT

Enzyme design is a critical area in biotechnology, with applications ranging from drug development to synthetic biology. Traditional methods for enzyme function prediction or protein binding pocket design often fall short in capturing the dynamic and complex nature of enzyme-substrate interactions, particularly in catalytic processes. To address the challenges, we introduce EnzymeFlow, a generative model that employs flow matching with hierarchical pre-training and enzyme-reaction co-evolution to generate catalytic pockets for specific substrates and catalytic reactions. Additionally, we introduce a large-scale, curated, and validated dataset of enzyme-reaction pairs, specifically designed for the catalytic pocket generation task, comprising a total of $328,192$ pairs. By incorporating evolutionary dynamics and reaction-specific adaptations, EnzymeFlow becomes a powerful model for designing enzyme pockets, which is capable of catalyzing a wide range of biochemical reactions. Experiments on the new dataset demonstrate the model's effectiveness in designing high-quality, functional enzyme catalytic pockets, paving the way for advancements in enzyme engineering and synthetic biology. The EnzymeFlow code can be found at https://anonymous.4open.science/r/EnzymeFlow-7420.

## 1 INTRODUCTION

Proteins are fundamental to life, participating in many essential interactions for biological processes (Whitford, 2013). Among proteins, enzymes stand out as a specialized class that serves as catalysts, driving and regulating nearly all chemical reactions and metabolic pathways across living organisms, from simple bacteria to complex mammals (Kraut, 1988; Murakami et al., 1996; Copeland, 2023) (visualized in Fig. 1). Their catalytic power is central to biological functions, enabling the efficient production of complex organic molecules in biosynthesis (Ferrer et al., 2008; Liu & Wang, 2007) and the creation of novel biological pathways in synthetic biology (Girvan & Munro, 2016; Keasling, 2010; Hodgman & Jewett, 2012). Examining enzyme functions across the tree of life deepens our understanding of the evolutionary processes that shape metabolic networks and enable organisms to adapt to their environments (Jensen, 1976; Glasner et al., 2006; Campbell et al., 2016; Pinto et al., 2022). Consequently, studying enzyme-substrate interactions is essential for comprehending biological processes and designing effective products.

Traditional methods have primarily focused on enzyme function prediction, annotation (Gligorijević et al., 2021; Yu et al., 2023), or enzyme-reaction retrieval (Mikhael et al., 2024; Hua et al., 2024b; Yang et al., 2024). These approaches lack the ability to design new enzymes that catalyze specific biological processes. Recent studies suggest that current function prediction models struggle to generalize to unseen enzyme reaction data (de Crecy-Lagard et al., 2024; Kroll et al., 2023a), limiting their utility in enzyme design. To effectively design enzymes, it is crucial not only to predict protein functions but also to identify and generate enzyme catalytic pockets specific to particular substrates and reactions, thereby enabling potentially valuable biological processes.

Figure 1: Enzyme-substrate Mechanism.

On the other hand, recent advances in deep generative models have significantly improved pocket design for protein-ligand complexes (Stärk et al., 2023; Zhang et al., 2023b; 2024d; Krishna et al., 2024), generating diverse and functional binding pockets for ligand molecules. However, these models cannot generalize directly to the design of enzyme catalytic pockets for substrates involved in catalytic processes. Unlike protein-ligand complexes, where ligand binding typically does not lead to

a chemical transformation, **enzyme-substrate interactions result in a chemical change where the substrate is converted into a product,** which has significantly different underlying mechanisms. More specifically, in protein-ligand binding, the ligand may induce a conformational change in the protein, affect its interactions with other molecules, or modulate its activity; in contrast, the formation of an enzyme-substrate complex is a precursor to a catalytic reaction, where the enzyme lowers the activation energy, facilitating the transformation of the substrate into a product. After the reaction, the enzyme is free to catalyze another substrate molecule. Therefore, current generative models for pocket design are restricted and limited to static ligand-binding interactions, failing to describe such dynamic transformations and the complex nature of enzyme-substrate interactions.

To address these limitations, we propose EnzymeFlow (demonstrated in Fig. 3), a flow matching model (Lipman et al., 2022; Liu et al., 2022; Albergo & Vanden-Eijnden, 2023) with enzyme-reaction co-evolution and structure-based pre-training for enzyme catalytic pocket generation. Our major contributions follow: **(1) EnzymeFlow—Flow Model for Enzyme Catalytic Pocket Design:** We define conditional flows for enzyme catalytic pocket generation based on backbone frames, amino acid types, and Enzyme Commission (EC) class. The generative flow process is conditioned on specific substrates and products, enabling potential catalytic processes. **(2) Enzyme-Reaction Co-Evolution:** Since enzyme-substrate interactions involve dynamic chemical transformations of substrate molecules, which is distinct from static protein-ligand interactions, we propose enzyme-reaction co-evolution with a new co-evolutionary transformer (*coEvoFormer*). The co-evolution is used to capture substrate-specificity in catalytic reactions. It encodes how enzymes and reactions evolve together, allowing the model to operate on evolutionary dynamics, which naturally comprehends the catalytic process. **(3) Structure-Based Hierarchical Pre-Training:** To leverage the vast data of geometric structures from existing proteins and protein-ligand complexes, we propose a structure-based hierarchical pre-training. This method progressively learns from protein backbones to protein binding pockets, and finally to enzyme catalytic pockets. This hierarchical learning of protein structures enhances geometric awareness within the model. **(4) EnzymeFill—Large-scale Pocket-specific Enzyme-Reaction Dataset with Pocket Structures:** Current enzyme-reaction datasets are based on full enzyme sequences or structures and lack precise geometry for how enzyme pockets catalyze the substrates. To address this, we construct a structure-based, curated, and validated enzyme catalytic pocket-substrate dataset, specifically designed for the catalytic pocket generation task.

## 2 RELATED WORK

### 2.1 PROTEIN EVOLUTION

Protein evolution learns how proteins change over time through processes such as mutation, selection, and genetic drift (Pál et al., 2006; Bloom & Arnold, 2009), which influence protein functions. Studies on protein evolution focus on understanding the molecular mechanisms driving changes in protein sequences and structures. Zuckerkandl & Pauling (1965) introduce the concept of the molecular clock, which postulates that proteins evolve at a relatively constant rate over time, providing a framework for estimating divergence times between species. DePristo et al. (2005) show that evolutionary rates are influenced by functional constraints, with regions critical to protein function (*e.g.*, active sites, binding interfaces) evolving more slowly due to purifying selection. This understanding leads to the development of methods for detecting functionally important residues based on evolutionary conservation. Understanding protein evolution has practical applications in protein engineering. By studying how natural proteins evolve to acquire new functions, researchers design synthetic proteins with desired properties (Xia & Levitt, 2004; Jäckel et al., 2008). Additionally, deep learning models increasingly integrate evolutionary principles to predict protein function and stability, design novel enzymes, and guide protein engineering (Yang et al., 2019; AlQuraishi, 2019; Jumper et al., 2021).

### 2.2 GENERATIVE MODELS FOR PROTEIN AND POCKET DESIGN

Recent advancements in generative models have advanced the field of protein design and binding pocket design, enabling the creation of proteins or binding pockets with desired properties and functions (Yim et al., 2023a;b; Chu et al., 2024; Hua et al., 2024a; Abramson et al., 2024). For example, RFDiffusion (Watson et al., 2023) employs denoising diffusion in conjunction with RoseTTAFold (Baek et al., 2021) for *de novo* protein structure design, achieving wet-lab-level generated structures that can be extended to binding pocket design. RFDiffusionAA (Krishna et al., 2024) extends RFDiffusion for joint modeling of protein and ligand structures, generating ligand-binding proteins and further leveraging MPNNs for sequence design. Additionally, FAIR (Zhang et al., 2023b) and

PocketGen (Zhang et al., 2024d) use a two-stage coarse-to-fine refinement approach to co-design pocket structures and sequences. Recent models leveraging flow matching frameworks have shown promising results in these tasks. For instance, FoldFlow (Bose et al., 2023) introduces a series of flow models for protein backbone design, improving training stability and efficiency. FrameFlow (Yim et al., 2023a) further enhances sampling efficiency and demonstrates success in motif-scaffolding tasks using flow matching, while MultiFlow (Campbell et al., 2024) advances to structure and sequence co-design. These flow models, initially applied to protein backbones, have been further generalized to binding pockets. For example, PocketFlow (Zhang et al., 2024e) combines flow matching with physical priors to explicitly learn protein-ligand interactions in binding pocket design, achieving stronger results compared to RFDiffusionAA. While these models excel in protein and binding pocket design, they primarily focus on static protein(-ligand) interactions and lack the ability to model the chemical transformations involved in enzyme-catalyzed reactions. This limitation may reduce their accuracy and generalizability in designing enzyme pockets for catalytic reactions. In EnzymeFlow, we aim to address these current limitations. An extended discussion of related works on AI-driven protein engineering can be found in App. C.

**Discussion regarding PocketFlow.** PocketFlow (Zhang et al., 2024e) has demonstrated strong performance in protein-ligand design, showing generalizability across various protein pocket categories. However, it falls short when applied to the design of enzyme catalytic pocket with specific substrates. One key limitation is that protein-ligand interactions are static, meaning that the training data and model design do not capture or describe the chemical transformations, such as the conversion or production of new molecules, that occur during enzyme-catalyzed reactions. This dynamic aspect of enzyme-substrate interactions is missing in current models. Another limitation is that PocketFlow fixes the overall protein backbone structure before designing the binding pocket, treating the pocket as a missing element to be filled in. This approach may not align with practical needs, as the overall protein backbone structure is often unknown before pocket design. Ideally, the design process should be reversed: the pocket should be designed first, influencing the overall protein structure. Despite these challenges, PocketFlow remains a good and leading work in pocket design. With EnzymeFlow, we aim to address these limitations, particularly in the context of catalytic pocket design.

## 3 ENZYMEFILL: LARGE-SCALE ENZYME POCKET-REACTION DATASET

A key limitation of current datasets, such as ESP (Kroll et al., 2023b), EnzymeMap (Heid et al., 2023), CARE (Yang et al., 2024), or ReactZyme (Hua et al., 2024b), is the lack of precise pocket information. These datasets typically provide enzyme-reaction data, including protein sequences and SMILES representations, which is used to predict EC numbers in practice. To address it, we introduce a new synthetic dataset, EnzymeFill, which includes precise pocket structures with substrate conformations. EnzymeFill is specifically introduced for enzyme catalytic pocket design.

**Data Source.** We construct a curated and validated dataset of enzyme-reaction pairs by collecting data from the Rhea (Bansal et al., 2022), MetaCyc (Caspi et al., 2020), and Brenda (Schomburg et al., 2002) databases. For enzymes in these databases, we exclude entries missing UniProt IDs or protein sequences. For reactions, we apply the following procedures: (1) remove cofactors, small ion groups, and molecules that appear in both substrates and products within a single reaction; (2) exclude reactions with more than five substrates or products; and (3) apply OpenBabel (O'Boyle et al., 2011) to standardize canonical SMILES. Ultimately, we obatin a total of $328,192$ enzyme-reaction pairs, comprising $145,782$ unique enzymes and $17,868$ unique reactions; we name it EnzymeFill.

**Catalytic Pocket with AlphaFill.** We identify all enzyme catalytic pockets using AlphaFill (Hekkelman et al., 2023), an AF-based algorithm that uses sequence and structure similarity to transplant ligand molecules from experimentally determined structures to predicted protein models. We download the AlphaFold structures for all enzymes and apply AlphaFill to extract the enzyme pockets. Simultaneously, we determine the reaction center by using atom-atom mapping of the reactions. During the pocket extraction process, AlphaFill first identifies homologous proteins of the target enzyme in the PDB-REDO database, along with their complexes with ligands (van Beusekom et al., 2018). It then transplants the ligands from the homologous protein complexes to the target enzyme through structural alignment (illustrated in Fig. 2(a)). After transplantation, we select the appropriate ligand molecule based on the number of atoms and its frequency of occurrence, and extract the pocket using a pre-defined radius of $10\text{Å}$ . We also perform clustering analysis on the extracted pockets using Foldseek (van Kempen et al., 2022), which reveals that enzyme catalytic pockets capture functional information more effectively than full structures (illustrated in Fig. 2(b)). For the extraction of

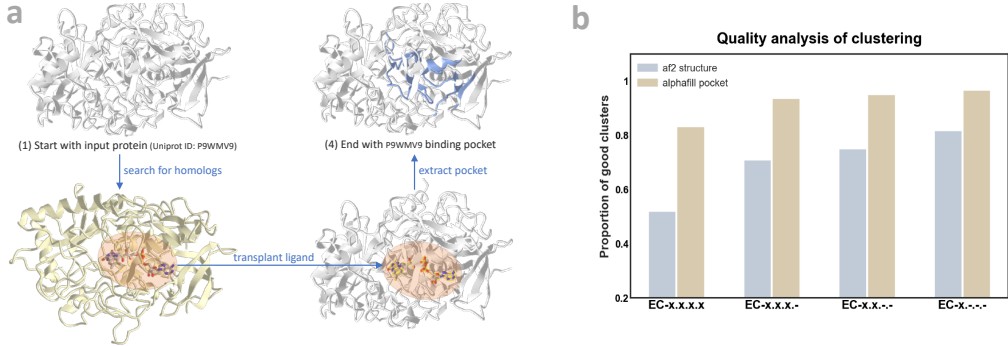

Figure 2: **(a)** Enzyme pocket extraction workflow with AlphaFill. **(b)** Quality analysis of clustering between enzyme pockets and full structures; good clusters have high functional concentration.

reaction centers, we first apply RXNMapper to extract atom-atom mappings (Schwaller et al., 2021), which maps the atoms between the substrates and products. We then identify atoms where changes occurred in chemical bonds, charges, and chirality, labeling these atoms as reaction centers.

**Data Debiasing for Generation.** To ensure the quality of catalytic pocket data for the design task, we exclude pockets with fewer than 32 residues[1], resulting in $232,520$ enzyme-reaction pairs. Additionally, enzymes and their catalytic pockets can exhibit significant sequence similarity. When enzymes that are highly similar in sequence appear too frequently in the dataset, they tend to belong to the same cluster or homologous group, which can introduce substantial biases during model training. To mitigate this issue and ensure a more balanced dataset, it is important to reduce the number of homologous enzymes by clustering and selectively removing enzymes from the same clusters. This helps to debias the data and improve the model's generalizability. We perform sequence alignment to cluster enzymes and identify homologous ones (Steinegger & Söding, 2017). We then revise the dataset into five major categories based on enzyme sequence similarity, resulting in: (1) $19,379$ pairs with at most $40\%$ homology, (2) $34,750$ pairs with at most $50\%$ homology, (3) $53,483$ pairs with at most $60\%$ homology, (4) $100,925$ pairs with at most $80\%$ homology, and (5) $132,047$ pairs with at most $90\%$ homology. In EnzymeFlow, we choose to use the clustered data with at most $60\%$ homology with $53,483$ samples for training. We provide more dataset statistics in App. H

## 4 ENZYMEFLOW

We introduce EnzymeFlow, a flow matching model with hierarchical pre-training and enzyme-reaction co-evolution for enzyme catalytic pocket design, conditioned on specific catalytic reactions and trained on EnzymeFill. We demonstrate the pipeline in Fig. 3, discuss the EnzymeFlow with co-evolution in Sec. 4.1, further introduce the structure-based hierarchical pre-training in Sec. 4.2.

### 4.1 ENZYME CATALYTIC POCKET GENERATION WITH FLOW MATCHING

**EnzymeFlow on Catalytic Pocket.** Following Yim et al. (2023a), we refer to the protein structure as the backbone atomic coordinates of each residue. A pocket with number of residues $N_r$ can be parameterized into SE(3) residue frames $\{(x^i, r^i, c^i)\}_{i=1}^{N_r}$, where $x^i \in \mathbb{R}^3$ represents the position (translation) of the $C_\alpha$ atom of the $i$-th residue, $r^i \in \text{SO}(3)$ is a rotation matrix defining the local frame relative to a global reference frame, and $c^i \in \{1, \ldots, 20\} \cup \{\times\}$ denotes the amino acid type, with an additional $\times$ indicating a *masking state* of the amino acid type. We refer to the residue block as $T^i = (x^i, r^i, c^i)$, and the entire pocket is described by a set of residues $\mathbf{T} = \{T^i\}_{i=1}^{N_r}$. Additionally, we denote the graph representations of substrate and product molecules in the catalytic reaction as $l_s$ and $l_p$, respectively. An enzyme-reaction pair can therefore be described as $(\mathbf{T}, l_s, l_p)$.

Following flow matching literature (Yim et al., 2023a; Campbell et al., 2024), we use time $t = 1$ to denote the source data. The conditional flow on the enzyme catalytic pocket $p_t(\mathbf{T}_t | \mathbf{T}_1)$ for a time step $t \in (0, 1]$ can be factorized into the probability density over continuous variables (translations and rotations) and the probability mass function over discrete variables (amino acid types) as:

$$p_t(\mathbf{T}_t | \mathbf{T}_1) = \prod_{i=1}^{N_r} p_t(x_t^i | x_1^i) \, p_t(r_t^i | r_1^i) \, p_t(c_t^i | c_1^i), \tag{1}$$

---

[1]32 residues are chosen based on LigandMPNN (Dauparas et al., 2023), ensuring high-quality interactions.

Figure 3: Overview of EnzymeFlow with hierarchical pre-training and enzyme-reaction co-evolution. **(1)** Flow model pre-trained on protein backbones and amino acid types. **(2)** Flow model further pre-trained on protein binding pockets, conditioned on ligand molecules with geometry-specific optimization. **(3)** Flow model fine-tuned on enzyme catalytic pockets, and conditioned on substrate and product molecules, with enzyme-reaction co-evolution and EC-class generation.

where the translation, rotation, and amino acid type at time $t$ are derived as:

$$x_t^i = (1-t)x_0^i + tx_1^i,\ x_0^i \sim \mathcal{N}(0, I);\ \ r_t^i = \exp_{r_0^i}(t \log_{r_0^i} r_1^i),\ r_0^i \sim \mathcal{U}_{\text{SO(3)}};$$
$$c_t^i \sim p_t(c_t^i | c_1^i) = \text{Cat}(t\,\delta(c_1^i, c_t^i) + (1-t)\,\delta(\textsf{X}, c_t^i)), \tag{2}$$

where $\delta(a, b)$ is the Kronecker delta, which equals to 1 if a = b and 0 if a $\neq$ b; Cat is a categorical distribution for the sampling of discrete amino acid type, with probabilities $t\delta(c_1^i, c_t^i) + (1-t)\delta(\textsf{X}, c_t^i)$. The discrete flow interpolates from the *masking state* $\textsf{X}$ at $t = 0$ to the actual amino acid type $c_1^i$ at $t = 1$ (Campbell et al., 2024). In a catalytic process, enzymes interact with substrates to produce specific products. In practical enzyme design, we typically know the substrates $l_s$ (as 3D atom point clouds) and the desired products $l_p$ (as 2D molecular graphs or SMILES). Therefore, the formation of the enzyme catalytic pocket should be conditioned on both substrates and products. Our enzyme flow matching model is conditioned on these two ligand molecules $l_s, l_p$, ensuring that the predictions of vector fields $v_\theta(\cdot)$ and loss functions account for the substrate and product molecules:

$$\mathcal{L}_{\text{trans}} = \sum_{i=1}^{N_r} \|v_\theta^i(x_t^i, t, l_s, l_p) - (x_1^i - x_0^i)\|_2^2;\ \ \mathcal{L}_{\text{rot}} = \sum_{i=1}^{N_r} \|v_\theta^i(r_t^i, t, l_s, l_p) - \frac{\log_{r_t^i} r_1^i}{1-t}\|_{\text{SO(3)}}^2;$$
$$\mathcal{L}_{\text{aa}} = -\sum_{i=1}^{N_r} \log p_\theta(c_1^i | v_\theta^i(c_t^i, t, l_s, l_p)). \tag{3}$$

To design the enzyme pocket and model protein-ligand interactions, we implement 3D and 2D GNNs to encode the substrate and product, respectively (implemented in App. E). The main vector field network applies cross-attention to model protein-ligand interactions and incorporates Invariant Point Attention (IPA) (Jumper et al., 2021) to encode protein features and make predictions. Following tricks in Yim et al. (2023a); Campbell et al. (2024), we let the the model predict the final structure at $t = 1$ and interpolates to compute the vector fields (discussed in App. F).

**EnzymeFlow on EC-Class.** The Enzyme Commission (EC) classification is crucial for categorizing enzymes based on the reactions they catalyze. Understanding the EC-class of an enzyme-reaction pair can help predict its function in various biochemical pathways (Bansal et al., 2022). Given its importance, EnzymeFlow leverages EC-class to enhance its generalizability across various enzymes and catalytic reactions. Therefore, our model incorporates EC-class, $y_{\text{ec}} \in \{1, \dots, 7\} \cup \{\textsf{X}\}$, as a discrete factor in the design process. The EC-class is sampled from a Categorical distribution with probabilities $t\delta(y_{\text{ec}_1}, y_{\text{ec}_t}) + (1-t)\delta(\textsf{X}, y_{\text{ec}_t})$. The discrete flow on EC-class interpolates from the *masking state* $\textsf{X}$ at $t = 0$ to the actual EC-class $y_{\text{ec}_1}$ at $t = 1$. The prediction and loss function are conditioned on the pocket frames and the substrate and product molecules:

$$\mathcal{L}_{\text{ec}} = -\log p_\theta(y_{\text{ec}_1} | v_\theta(\mathbf{T}_t, t, l_s, l_p, y_{\text{ec}_t})). \tag{4}$$

The model predicts the final EC-class at $t = 1$ and interpolates to compute its vector field. For EC-class prediction, we first employ a EC-class embedding network to encode $y_{\text{ec}_t}$. The final predicted EC-class is obtained by pooling cross-attention between the encoded enzyme and EC-class features.

Figure 4: Catalytic pocket design example using EnzymeFlow (UniProt: Q7U4P2). The pocket generation is conditioned on reaction CN[C@H](C(=O)C)CS.C/C=C\\1/C(=C/c2[nH]c(c(c2C)CCC(=O)O)/C=C/2\\N=C(C(=C2CCC (=O)O)C)C[C@H]2NC(=O)C(=C2C)C=C)/NC(=O)[C@H]1C → CN[C@H](C(=O)C)CSC(C1=C(C)C(=O)N[C@H]1Cc1[nH]c(c(c1C)CCC(=O) O)/C=C/1\\N=C(C(=C1CCC(=O)O)C)C[C@H]1NC(=O)C(=C1C)C=C)C of EC4 (ligase enzyme), from $t = 0$ to $t = 1$.

### 4.1.1 ENZYMEFLOW WITH ENZYME-REACTION CO-EVOLUTION

Enzyme (protein) evolution refers to the process by which enzyme structures and functions change over time due to genetic variations, such as mutations, duplications, and recombinations. These changes can lead to alterations in amino acids, potentially affecting the enzyme structure, function, stability, and interactions (Pál et al., 2006; Sikosek & Chan, 2014). Reaction evolution, on the other hand, refers to the process by which chemical reactions or substrates, particularly those catalyzed by enzymes, change and diversify within biological systems over time (illustrated in Fig. 3(3)(d)).

**Co-Evolutionary Dynamics.** Enzymes can co-evolve with the metabolic or biochemical pathways they are part of, adapting to changes in substrate availability, the introduction of new reaction steps, or the need for more efficient flux through the pathway. As pathways evolve, enzymes within them may develop new catalytic functions or refine existing ones to better accommodate these changes (Noda-Garcia et al., 2018). This process frequently involves the co-evolution of enzymes and their substrates. As substrates change—whether due to the introduction of new compounds in the environment or mutations in other metabolic pathways—enzymes may adapt to catalyze reactions with these new substrates, leading to the emergence of entirely new reactions. Understanding enzyme-substrate interactions, therefore, requires considering their evolutionary dynamics, as these interactions are shaped by the evolutionary history and adaptations of both enzymes and their substrates. This co-evolutionary process is crucial for explaining how enzymes develop new functions and maintain efficiency in response to ongoing changes in their biochemical environment.

To capture the evolutionary dynamics, we introduce the concept of enzyme-reaction co-evolution into EnzymeFlow. We compute the enzyme and reaction evolution by applying multiple sequence alignment (MSA) to enzyme sequences and reaction SMILES, respectively (Steinegger & Söding, 2017). The co-evolution of an enzyme-reaction pair is represented by a matrix $U \in \mathbb{R}^{N_{\text{MSA}} \times N_{\text{token}}}$, which combines the MSA results of enzyme sequences and reaction SMILES (illustrated in Fig. 3(3)(d) & Fig. 9), where $N_{\text{MSA}}$ denotes the number of MSA sequences and $N_{\text{token}}$ denotes the length of the MSA alignment preserved. And each element $u^{mn} \in \{1, \ldots, 64\} \cup \{\times\}$ in $U$ denotes a tokenized character from our co-evolution vocabulary, with additional $\times$ indicating the *masking state*.

**EnzymeFlow on Co-Evolution.** The flow for co-evolution follows a similar approach to that used for amino acid types and EC-class, treating it as a discrete factor in the design process. The co-evolution is sampled from a Categorical distribution, where each element has probabilities $t\delta(u_1^{mn}, u_t^{mn}) + (1 - t)\delta(\times, u_t^{mn})$. Each element flows independently, reflecting the natural independence of amino acid mutations (Boyko et al., 2008). The discrete flow on co-evolution interpolates from the *masking state* $\times$ at $t = 0$ to the actual character $u_1^{mn}$ at $t = 1$. The prediction and loss function are conditioned on the pocket frames and the substrate and product molecules:

$$\mathcal{L}_{\text{coevo}} = - \sum_{m=1}^{N_{\text{MSA}}} \sum_{n=1}^{N_{\text{token}}} \log p_\theta(u_1^{mn} | v_\theta(\mathbf{T}_t, t, l_s, l_p, u_t^{mn})). \quad (5)$$

The model predicts the final co-evolution at $t = 1$ and interpolates to compute its vector field. For co-evolution prediction, we first introduce a co-evolutionary MSA transformer (coEvoFormer) to encode $U_t$ (implemented in App. D). The final predicted co-evolution is obtained by computing cross-attention between the encoded enzyme and ligand, and the encoded co-evolution features.

We can therefore express EnzymeFlow with co-evolutionary dynamics for catalytic pocket design as:

$$p_t(\mathbf{T}_t, U_t, y_{\text{ec}_t} | \mathbf{T}_1, U_1, y_{\text{ec}_1}, l_s, l_p) = p_t(y_{\text{ec}_t} | y_{\text{ec}_1}, \mathbf{T}_t) \, p_t(U_t | U_1, \mathbf{T}_t) \, p_t(\mathbf{T}_t | \mathbf{T}_1, l_s, l_p). \quad (6)$$

The final EnzymeFlow model performs flows on protein backbones, amino acid types, EC-class, and enzyme-reaction co-evolution. Given the SE(3)-invariant prior and the main SE(3)-equivariant network in EnzymeFlow, the pocket generation process is also SE(3)-equivariant (proven in App. G).

## 4.2 STRUCTURE-BASED HIERARCHICAL PRE-TRAINING

In addition to the standard EnzymeFlow for enzyme pocket design, we propose a hierarchical pre-training strategy to enhance the generalizability of the model across different enzyme categories. The term *hierarchical pre-training* is used because the approach first involves training the flow model to understand protein backbone generation, followed by training it to learn the geometric relationships between proteins and ligand molecules, which form protein binding pockets. After the flow model learns these prior knowledge, we fine-tune it specifically on an enzyme-reaction dataset to generate enzyme catalytic pockets. The term *hierarchical* reflects the progression from protein backbone generation, to protein binding pocket formation, and finally to enzyme catalytic pocket generation.

Specifically, we begin by pre-training the flow model on a protein backbones. Once the model learns it, we proceed to post-train it on a protein-ligands, with the objective of generating binding pockets conditioned on the ligand molecules. Finally, the model is fine-tuned on our EnzymeFlow dataset to generate valid enzyme catalytic pockets for specific substrates and catalytic reactions.

### 4.2.1 PROTEIN BACKBONE PRE-TRAINING

The initial step involves pre-training the model on a protein backbone dataset (illustrated in Fig. 3(1)). We use the backbone dataset discussed in FrameFlow (Yim et al., 2023a). This pre-training focuses solely on SE(3) backbone frames and discrete amino acid types, allowing the flow model to acquire foundational knowledge of protein backbone geometry and structure.

### 4.2.2 PROTEIN-LIGAND PRE-TRAINING

Following the protein backbone pre-training, we proceed to pre-train the flow model on a protein-ligand dataset (illustrated in Fig. 3(2)). Specifically, we use PDBBind2020 (Wang et al., 2004). This pre-training focuses on binding pocket frames, with the flow model conditioned on the 3D representations of ligand molecules $l$ consisting of $N_l$ atoms. Additionally, binding affinity $y_{kd} \in \mathbb{R}$ and atomic-level pocket-ligand distance $D^i \in \mathbb{R}^{4 \times N_l}$ for the $i$-th residue frame serve as optimization factors. The parametrization is similar to Eq. 6, with conditioning on the ligand molecule as follows:

$$p_t(\mathbf{T}_t, y_{kd}|\mathbf{T}_1, l) = p_t(y_{kd}|\mathbf{T}_t, l) \, p_t(\mathbf{T}_t|\mathbf{T}_1, l). \tag{7}$$

In addition to the flow matching losses in Eq. 3, we introduce a loss of protein-ligand interaction to prevent the collision during the binding in generation process. Conceptually, this ensures that the generated pocket atoms do not come into contact with the surface of the ligand molecule. Following previous work on protein-ligand binding (Lin et al., 2022), the surface of a ligand $\{a_j | j \in \mathbb{N}(N_l)\}$ is defined as $\{a \in \mathbb{R}^3 | S(a) = \gamma\}$, where $S(a) = -\rho \log(\sum_{j=1}^{N_l} \exp(-|a - a_j|^2/\rho))$. The interior of the ligand molecule is thus defined by $\{a \in \mathbb{R}^3 | S(a) < \gamma\}$, and the binding pocket atoms are constrained to lie within $\{a \in \mathbb{R}^3 | S(a) > \gamma\}$. We also introduce a protein-ligand distance loss to regularize pairwise atomic distances, along with a binding affinity loss to enforce the generation of more valid protein-ligand pairs. These objectives are defined as follows:

$$\mathcal{L}_{\text{inter}} = \sum_{i=1}^{N_r} \max(0, \gamma - S(\hat{A}_t^i)), \ \mathcal{L}_{\text{dist}} = \sum_{i=1}^{N_r} \frac{\|\mathbf{1}\{D_1^i < 8\text{Å}\}(D_1^i - \hat{D}_t^i)\|_2^2}{\sum \mathbf{1}_{D_1^i < 8\text{Å}}}, \ \mathcal{L}_{\text{kd}} = \|y_{kd} - \hat{y}_{kd}\|^2, \quad (8)$$

where $\hat{A}^i \in \mathbb{R}^{4 \times 3}$ denotes the predicted atomic positions of $i$-th residue frame, $\gamma = 6$ and $\rho = 2$ are hyperparameters, and $\hat{y}_{kd}$ is the predicted binding affinity for a generated pair. $\hat{D}^i \in \mathbb{R}^{4 \times N_l}$ is defined similarly to $D^i$, based on the distance between the predicted atomic positions and ligand positions for the $i$-th residue frame. The predicted affinity $\hat{y}_{kd}$ is obtained by pooling the encoded protein and ligand features. These additional losses are incorporated to improve the model's generalizability, enforcing more constrained geometries for more valid protein pocket design.

## 5 EXPERIMENT — GENERATING CATALYTIC POCKET CONDITIONED ON REACTIONS AND SUBSTRATES

EnzymeFlow is essentially a *function-based* protein design model, where the intended function is defined by the reaction the enzyme will catalyze. Here, we demonstrate that EnzymeFlow outperforms current *structure-based* substrate-conditioned protein design models in both the structural and functional aspects, showing its capability and advantage in enzyme catalytic pocket design.

Table 1: EnzymeFlow Evaluation Data Statistics.

| Data | Pair #pair | Enzyme #enzyme | Substrate #substrate | #avg atom | Product #product | #avg atom | EC1 | EC2 | EC3 | EC4 | EC5 | EC6 | EC7 |
|------|------|------|------|------|------|------|------|------|------|------|------|------|------|
| Raw | 232520 | 97912 | 7259 | 30.81 | 7664 | 30.34 | 44881 (19.30) | 75944 (32.66) | 37728 (16.23) | 47242 (20.32) | 8315 (3.58) | 18281 (7.86) | 129 (0.06) |
| Train | 53483 | 22350 | 6112 | 30.95 | 6331 | 30.34 | 11674 (21.83) | 18419 (34.44) | 11394 (21.30) | 5555 (10.39) | 2194 (4.10) | 4200 (7.85) | 47 (0.09) |
| Eval | 100 | 100 | 100 | 30.7 | 94 | 28.84 | 17 (17.00) | 17 (17.00) | 17 (17.00) | 17 (17.00) | 16 (16.00) | 16 (16.00) | 0 (0.00) |

We compare EnzymeFlow with state-of-the-arts representative baselines, including template-matching method DEPACT (Chen et al., 2022), deep equivariant and iterative refinement model PocketGen (Zhang et al., 2024d), golden-standard diffusion model RFDiffusionAA (Krishna et al., 2024), and the most recent PocketFlow[2] (Zhang et al., 2024e). For RFDiffusionAA-designed pockets, we apply LigandMPNN (Dauparas et al., 2023) to inverse fold and predict the sequences post-hoc. We provide EnzymeFlow code at `https://anonymous.4open.science/r/EnzymeFlow-7420`.

**Evaluation Data.** We use MMseqs2 to perform clustering with a $10\%$ homology threshold, selecting the center of each cluster as the initial dataset, resulting in a total of $3,417$ pairs. After de-duplicating both repeated substrates and UniProt entries, we are left with 839 unique enzyme-reaction pairs. We then uniformly sample data across different EC classes, selecting 17 pairs from EC1 to EC4 classes and 16 pairs from EC5 and EC6 classes, respectively, resulting in a total of 100 unique catalytic pockets and 100 unique reactions. Each enzyme-reaction pair is labeled with a ground-truth EC-class from EC1 to EC6. We present the EC-class distribution in the evaluation set in Tab. 1.

**Reaction-conditioned Generation.** For pocket design and model sampling, we perform conditional generation on each reaction (or substrate), generating 100 catalytic pockets for each reaction in the evaluation set. We evaluate the generated pockets for their structures and functions (*i.e.,* EC-class).

## 5.1 CATALYTIC POCKET STRUCTURE EVALUATION

We begin by assessing the structural validity of generated catalytic pockets. While enzyme function determines whether the designed pocket can catalyze a specific reaction, the structure determines whether the substrate conformation can properly bind to the catalytic pocket. We provide some visual examples of designed pockets in Fig. 5 and Fig. 14.

**Metrics.** We use the following metrics to evaluate and compare the structural validity of the generated pockets. Constrained-site RMSD (`cRMSD`): The structural distance between the ground-truth and generated pockets, as proposed in Hayes et al. (2024). `TM-score`: The topological similarity between the generated and ground-truth pockets in local deviations. Aggregated Chai Score (`chai`): The confidence and structural validity of the pocket-substrate complex by running Chai (Chai, 2024). It is calculated as $0.2 \times pTM + 0.8 \times ipTM - 100 \times clash$, where $pTM$ is the predicted template modeling score, $ipTM$ is the interface predicted template modeling score (as used in Jumper et al. (2021)), and the definition of `chai` is proposed by Chai (2024). Binding Affinity (`Kd`): The binding affinity between the generated catalytic pocket and the substrate conformation is computed using AutoDock Vina (Trott & Olson, 2010). Amino Acid Recovery (`AAR`): The overlap ratio between the predicted and ground-truth amino acid types in the generated pocket. Enzyme Commission Accuracy (`ECacc`): The accuracy of matching the EC-class of generated pockets with the ground-truth EC-class.

Table 2: Evaluation of structural validity of EnzymeFlow- and baseline-generated catalytic pockets. The binding affinities (`Kd`) and structural confidence (`chai`) are computed by performing docking on the catalytic pocket and substrate conformation using Vina (Trott & Olson, 2010) and Chai (Chai, 2024), respectively. We highlight top three results in **bold**, underline, and *italic*, respectively.

| Model | cRMSD (↓) Top1 | Top10 | Median | TM-score (↑) Top1 | Top10 | Median | Kd (↓) | chai (↑) | AAR (↑) | ECacc (↑) |
|------|------|------|------|------|------|------|------|------|------|------|
| Eval Data | | - | | | - | | -4.65 | - | - | - |
| DEPACT | 9.25 | 9.75 | 11.16 | 0.238 | 0.206 | 0.149 | -5.46 | 0.125 | 0.112 | 0.149 |
| PocketGen | 7.65 | 8.14 | 10.45 | 0.260 | 0.233 | 0.193 | -5.01 | 0.121 | 0.176 | 0.152 |
| RFDiffusionAA | 9.13 | 9.77 | 11.92 | 0.269 | 0.245 | 0.198 | **-12.71** | **0.232** | 0.153 | 0.170 |
| PocketFlow | 7.42 | 8.09 | 10.01 | 0.268 | *0.260* | 0.197 | -4.93 | 0.123 | *0.207* | 0.166 |
| EnzymeFlow (T=50) | **6.94** | **7.57** | 9.04 | **0.290** | **0.262** | **0.209** | -5.03 | 0.129 | **0.216** | **0.280** |
| w/o coevo | 7.02 | 7.60 | *9.15* | 0.288 | *0.260* | *0.205* | -4.86 | 0.123 | 0.196 | 0.246 |
| w/o pretraining | *7.01* | 7.69 | 9.29 | *0.286* | 0.261 | 0.207 | -4.33 | *0.134* | 0.202 | *0.255* |
| w/o coevo+pretraining | 7.05 | 7.81 | 9.43 | 0.278 | 0.255 | 0.204 | -4.72 | 0.125 | 0.154 | 0.221 |
| EnzymeFlow (T=100) | 6.97 | **7.57** | **9.02** | 0.283 | 0.258 | 0.207 | *-5.31* | 0.135 | 0.215 | 0.273 |

**Results.** We compare the structural validity between EnzymeFlow- and baseline-generated catalytic pockets in Tab. 2. EnzymeFlow and its ablation models outperform baseline models, including leading

---

[2]PocketFlow is not open-sourced yet, we implement and train it on EnzymeFill without fixing the backbones.

Reference — RFDiffAA-designed — PocketFlow-designed — EnzymeFlow-designed

Superimposition

TM-score: 0.20  RMSD: 3.36

TM-score: 0.34  RMSD: 3.02

TM-score: 0.34  RMSD: 1.21

(Uniprot ID: B8MXP5)

Figure 5: Case study of catalytic pocket design (UniProt: B8MXP5). We show the reference and designed pockets of different models. The pocket generation is conditioned on reaction `OC[C@H]1O[C@@H](Oc2ccccc2/C=C\\C(=O)O)[C@@H]([C@H]([C@@H]1O)O)O → OC(=O)/C=C\\c1ccccc1O` of EC3.

models like RFDiffusionAA and PocketFlow, with significant improvements in `cRMSD`, `TM-score`, and `ECacc`, and competitive performance in `AAR`. This demonstrates that EnzymeFlow is capable of generating more structurally valid catalytic pockets, aligning with the enzyme function analysis presented in Fig. 6. The average improvements over RFDiffusionAA in `cRMSD`, `TM-score`, `AAR`, and `EC-Acc` are 23.9%, 7.8%, 41.1%, and 64.7%, respectively. Additionally, EnzymeFlow slightly outperforms PocketFlow in catalytic-substrate binding, showing improved affinity scores (`Kd`) and structural confidence (`chai`) by 2.1% and 9.8%, respectively.

However, EnzymeFlow underperforms RFDiffusionAA in binding scores, reflected by lower affinities and structural confidence. However, considering that the affinities of EnzymeFlow-generated catalytic pockets (-5.03) are close to those of enzyme-reaction pairs in the evaluation set (-4.65), the binding of EnzymeFlow remains acceptable, as enzymes and substrates do not always require tight binding to catalyze reactions because of the kinetic mechanism (Cleland, 1977; Arcus & Mulholland, 2020).

## 5.2 QUANTITATIVE ANALYSIS OF ENZYME FUNCTION

The key question is how we can *quantitatively* assess enzyme functions, *i.e.,* catalytic ability, of the generated pockets for a given reaction. To answer this, we perform enzyme function analysis on the designed catalytic pockets. Accurate annotated enzyme function is important for catalytic pocket design because it helps identify the functionality and the active sites that should be preserved or modified to improve catalytic efficiency (Rost, 2002; Barglow & Cravatt, 2007; Yu et al., 2023).

**Enzyme Function Comparison.** In EnzymeFlow, we co-annotate the enzyme function alongside the catalytic pocket design, allowing their functions to directly influence the structure generation. This integration of enzyme function annotation into EnzymeFlow ensures functionality control throughout the design. For baselines that design general proteins rather than enzyme-specific pockets, we perform enzyme function annotation post-hoc using CLEAN (Yu et al., 2023) to classify and annotate the EC-class of the generated pockets. After labeling each generated pocket with a EC-class, we compare it to the ground-truth EC-class associated with the actual reaction to compute EC-class accuracy, which quantifies how well the generated pockets align with the intended enzyme functions.

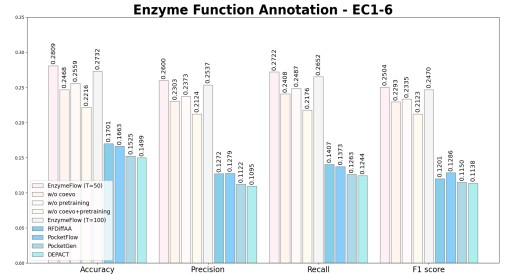

Figure 6: Quantitative comparison of annotated enzyme functions between EnzymeFlow- and baseline-generated catalytic pockets across all EC classes, using four multi-label accuracy metrics. Light color represents EnzymeFlow and its ablation models, blue color represents baseline pocket design models.

**Results.** We quantitatively compare the annotated enzyme functions between EnzymeFlow- and baseline-generated catalytic pockets across all EC classes in Fig. 6, and compare the per-class performance in Fig. 7. These figures allow us to interpret the functions of enzyme catalytic pockets designed by different models. From Fig. 6, EnzymeFlow and its ablation models achieve the highest values across various multi-label accuracy metrics, including accuracy (0.2809), precision (0.2600), recall (0.2722), and F1 score (0.2504), outperforming models like RFDiffusionAA and PocketFlow. Additionally, Fig. 7 illustrates per-class enzyme function accuracy, where EnzymeFlow demonstrates strong performance in EC2, EC4, EC5, and EC6, competitive performance in EC3, but slightly weaker performance in EC1 compared to baseline models. Baseline models tend to perform poorly in EC5 and EC6, with per-class occurrence and accuracy showing values close to 0. In contrast,

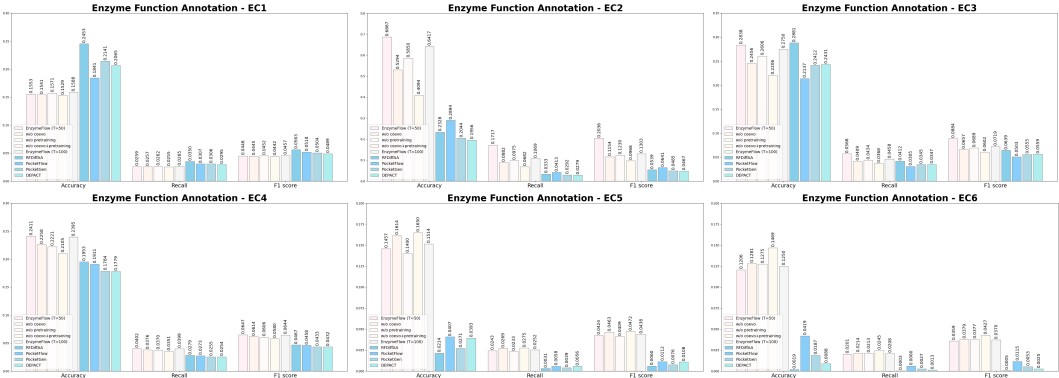

Figure 7: Quantitative comparison of annotated enzyme functions between EnzymeFlow- and baseline-generated catalytic pockets per EC-class, using accuracy, recall, and F1 score. Light color represents EnzymeFlow and ablation models, blue color represents baseline pocket design models.

EnzymeFlow generates more functionally diverse and accurate catalytic pockets, maintaining higher accuracy across different EC classes.

Additionally, for a fairer comparison, in Fig. 8, we compare EnzymeFlow with co-generated enzyme functions, EnzymeFlow with functions annotated post hoc by CLEAN, and baseline models with functions also annotated post hoc by CLEAN. This comparison aims to evaluate the enzyme functions of generated catalytic pockets of different pocket design models using post-hoc function annotation via CLEAN. We observe that EnzymeFlow outperforms the baselines in multi-label accuracy metrics, even when functions are annotated post hoc.

In conclusion, EnzymeFlow generates catalytic pockets that are better compared to other pocket design models, providing more accurate and diverse enzyme functions, which suggests enhanced catalytic potential. From both functional and structural perspectives, the *function-based, reaction-conditioned* EnzymeFlow outperforms current *structure-based, substrate-conditioned* protein design models in both structural validity and intended function design (catalytic ability). EnzymeFlow leverages enzyme-reaction co-evolution to effectively capture the dynamic changes in catalytic reactions as substrates are transformed into products. This approach enables function-based enzyme design, resulting in the generation of more functionally and structurally valid catalytic pockets for specific reactions.

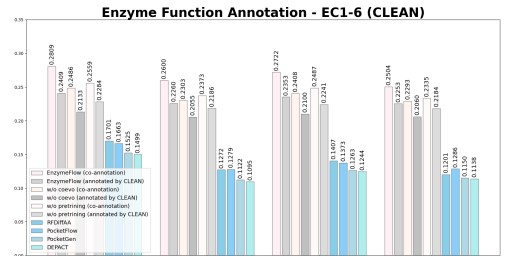

Figure 8: Quantitative comparison of annotated enzyme functions between EnzymeFlow- and baseline-generated catalytic pockets across all EC classes, using four multi-label accuracy metrics. Light color represents EnzymeFlow with enzyme function co-annotation, gray color represents EnzymeFlow with enzyme functions annotated by CLEAN post hoc, blue color represents baseline pocket design models with enzyme functions annotated by CLEAN post hoc.

## 6 LIMITATION AND FUTURE WORK

EnzymeFlow addresses key challenges in designing enzyme catalytic pockets for specific reactions, but several limitations remain. The first limitation is that EnzymeFlow currently generates only the catalytic pocket residues, rather than the entire enzyme structure. Ideally, the catalytic pocket should be designed first, followed by the design or reconstruction of the full enzyme structure based on the pocket. While we are developing to use ESM3 (Hayes et al., 2024) to reconstruct the full enzyme structure based on the designed catalytic pocket (discussed in App. I), this is not the most ideal solution. ESM3 is not specifically trained for enzyme-related tasks, which may limit its performance in enzyme design. In future versions of EnzymeFlow, we are working to fine-tune large biological models like ESM3 (Hayes et al., 2024), RFDiffusionAA (Krishna et al., 2024), or Genie2 (Lin et al., 2024) to specialize them for enzyme-related tasks, particularly for inpainting functional motifs of enzymes (enzyme catalytic motif scaffolding). Additionally, we aim to create an end-to-end model that combines EnzymeFlow with these large models, enabling catalytic pocket generation and functional motif inpainting in a single step, rather than in a two-step process. The second limitation, though minor, is that EnzymeFlow currently operates only on enzyme backbones and does not model or generate enzyme side chains. In future work, we plan to incorporate models like DiffPack (Zhang et al., 2024c) or develop a full-atom model to address this.

## REPRODUCIBILITY STATEMENT

We provide our code and data examples with demonstrations at `https://anonymous.4open.science/r/EnzymeFlow-7420`. In particular, a Jupyter notebook demonstrating the *de novo* design of enzyme catalytic pockets conditioned on specific reactions is available at `https://anonymous.4open.science/r/EnzymeFlow-7420/enzymeflow_demo.ipynb`. For those who prefer not to dive into the full codebase, we have also open-sourced key model components in App. E, App. D, and other appendix sections.

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

## A  FUTURE WORK IN PROGRESS: AI-DRIVEN ENZYME DESIGN PLATFORM

As discussed in Sec. 6, there are several limitations in the current version of EnzymeFlow. Here, we briefly outline the next steps and improvements we are actively working on for the upcoming version. Currently, EnzymeFlow generates only catalytic pocket residues rather than full enzyme structures. Ideally, the catalytic pocket should be designed first, followed by the reconstruction of the full enzyme structure based on the pocket. While we currently use ESM3 (Hayes et al., 2024) for this reconstruction, this approach is not ideal. Fine-tuning ESM3 or RFDiffusionAA (Krishna et al., 2024) would be preferable, but unfortunately, training scripts for these wonderful models are not provided, making it impossible to directly fine-tune them on our EnzymeFill dataset.

To address this, we are borrowing concepts from Wang et al. (2021) and Lin et al. (2024), which focuses on inpainting proteins and scaffolding functional motifs. We are working to integrate this concept into EnzymeFlow's pipeline, as part of our primary design. Our goal is to develop an end-to-end automated AI-driven enzyme discovery system that works as follows:

- 1. **Catalytic Pocket Design**: The system will first design enzyme catalytic pockets.
- 2. **Scaffolding Functional Motifs**: Next, it will scaffold the functional motifs to generate full enzyme structures.
- 3. **Substrate Docking**: Using methods like DiffDock (Corso et al., 2022), DynamicBind (Lu et al., 2024), or fine-tuned Chai (Chai, 2024) on EnzymeFill, the system will bind substrates to the catalytic pockets.
- 4. **Inverse Folding**: The enzyme-substrate complex will undergo inverse folding using LigandMPNN (Dauparas et al., 2023).
- 5. **Computational Screening**: Finally, the system will perform computational screening to select the best-generated enzymes.

This entire process is being developed into an integrated, end-to-end solution for AI-driven enzyme design. We are very excited about the potential of this project and look forward to achieving a fully automated enzyme design system in the near future.

## B  OPEN DISCUSSION: WHY IS SUBSTRATE/REACTION-SPECIFIED ENZYME DESIGN NEEDED?

EnzymeFlow is unique in its leading approach to function-based *de novo* protein design. Currently, most protein design models, whether focused on backbone generation (Yim et al., 2023a;b; Bose et al., 2023; Campbell et al., 2024; Krishna et al., 2024) or pocket design (Zhang et al., 2023b;a; 2024d;e), are structure-based. These models aim to design or modify proteins to achieve a specific 3D structure, prioritizing stability, folding, and molecular interactions. The design process typically involves optimizing a protein structure to minimize energy and achieve a stable structural conformation (Khoury et al., 2014; Pelay-Gimeno et al., 2015).

In contrast, function-based protein design focuses on creating proteins that perform specific bio-chemical tasks, such as catalysis, signaling, or even binding (Martin et al., 1998; Thornton et al., 1999). These models are driven by the need for proteins to carry out particular functions rather than adopt a specific 3D structure. Function-based design often targets the active site or binding pockets, optimizing them for specific molecular interactions—in our case, the enzyme's catalytic pockets.

Our philosophy is that protein function determines its structure, meaning that a protein folds into a specific 3D shape to achieve its intended function, and the resulting structure can then be translated into a proper sequence—essentially, *protein function → protein structure → protein sequence*. EnzymeFlow follows this philosophy. Specifically, the function of an enzyme is determined by its ability to catalyze a specific reaction or interact with a specific substrate. Therefore, our enzyme pocket design process begins with the reaction or substrate in mind, incorporating reaction/substrate specificity into the generation process. The reaction or substrate represents the functional target for the generated enzyme pockets.

In this approach, EnzymeFlow generates enzyme pocket structures specified for the desired protein function, which contrasts with current generative methods that prioritize structure first. These existing

methods operate on the idea that *protein structure → protein function → protein sequence*. However, proteins should be designed primarily for their functionality, not just their structures. EnzymeFlow's focus on function-based design could serve as an inspiration for future advancements, leading the way toward more purposeful, function-driven protein design.

# C RELATED WORK

## C.1 PROTEIN REPRESENTATION LEARNING

Graph representation learning emerges as a potent strategy for representing and learning about proteins and molecules, focusing on structured, non-Euclidean data (Satorras et al., 2021; Luan et al., 2020; 2022; Hua et al., 2022a;b; Luan et al., 2024b;a). In this context, proteins and molecules can be effectively modeled as 2D graphs or 3D point clouds, where nodes correspond to individual atoms or residues, and edges represent interactions between them (Gligorijević et al., 2021; Zhang et al., 2022; Hua et al., 2023; Zhang et al., 2024a). Indeed, representing proteins and molecules as graphs or point clouds offers a valuable approach for gaining insights into and learning the fundamental geometric and chemical mechanisms governing protein-ligand interactions. This representation allows for a more comprehensive exploration of the intricate relationships and structural features within protein-ligand structures (Tubiana et al., 2022; Isert et al., 2023; Zhang et al., 2024b).

## C.2 PROTEIN FUNCTION ANNOTATION

Protein function prediction aims to determine the biological role of a protein based on its sequence, structure, or other features. It is a crucial task in bioinformatics, often leveraging databases such as Gene Ontology (GO), Enzyme Commission (EC) numbers, and KEGG Orthology (KO) annotations (Bairoch, 2000; Consortium, 2004; Mao et al., 2005). Traditional methods like BLAST, PSI-BLAST, and eggNOG infer function by comparing sequence alignments and similarities (Altschul et al., 1990; 1997; Huerta-Cepas et al., 2019). Recently, deep learning has introduced more advanced approaches for protein function prediction (Ryu et al., 2019; Kulmanov & Hoehndorf, 2020; Bonetta & Valentino, 2020). There are two major types of function prediction models, one uses only protein sequence as their input, while the other also uses experimentally-determined or predicted protein structure as input. Typically, these methods predict EC or GO annotations to approximate protein functions, rather than describing the exact catalyzed reaction, which is a limitation of these approaches.

## C.3 PROTEIN EVOLUTION

Protein evolution learns how proteins change over time through processes such as mutation, selection, and genetic drift (Pál et al., 2006; Bloom & Arnold, 2009), which influence protein functions. Studies on protein evolution focus on understanding the molecular mechanisms driving changes in protein sequences and structures. Zuckerkandl & Pauling (1965) introduce the concept of the molecular clock, which postulates that proteins evolve at a relatively constant rate over time, providing a framework for estimating divergence times between species. DePristo et al. (2005) show that evolutionary rates are influenced by functional constraints, with regions critical to protein function (*e.g.*, active sites, binding interfaces) evolving more slowly due to purifying selection. This understanding leads to the development of methods for detecting functionally important residues based on evolutionary conservation. Understanding protein evolution has practical applications in protein engineering. By studying how natural proteins evolve to acquire new functions, researchers design synthetic proteins with desired properties (Xia & Levitt, 2004; Jäckel et al., 2008). Additionally, deep learning models increasingly integrate evolutionary principles to predict protein function and stability, design novel enzymes, and guide protein engineering (Yang et al., 2019; AlQuraishi, 2019; Jumper et al., 2021).

## C.4 GENERATIVE MODELS FOR PROTEIN AND POCKET DESIGN

Recent advancements in generative models have advanced the field of protein design and binding pocket design, enabling the creation of proteins or binding pockets with desired properties and functions (Yim et al., 2023a;b; Chu et al., 2024; Hua et al., 2024a; Abramson et al., 2024). For example, RFDiff (Watson et al., 2023) employs denoising diffusion in conjunction with RoseTTAFold (Baek et al., 2021) for *de novo* protein structure design, achieving wet-lab-level generated structures that can be extended to binding pocket design. RFDiffusionAA (Krishna et al., 2024) extends RFDiff for joint modeling of protein and ligand structures, generating ligand-binding proteins and further leveraging GNNs for sequence design. Additionally, FAIR (Zhang et al., 2023b) and PocketGen

(Zhang et al., 2024d) use a two-stage coarse-to-fine refinement approach to co-design pocket structures and sequences. Recent models leveraging flow matching frameworks have shown promising results in these tasks. For instance, FoldFlow (Bose et al., 2023) introduces a series of flow models for protein backbone design, improving training stability and efficiency. FrameFlow (Yim et al., 2023a) further enhances sampling efficiency and demonstrates success in motif-scaffolding tasks using flow matching, while MultiFlow (Campbell et al., 2024) advances to structure and sequence co-design. These flow models, initially applied to protein backbones, have been further generalized to binding pockets. For example, PocketFlow (Zhang et al., 2024e) combines flow matching with physical priors to explicitly learn protein-ligand interaction types in pocket design, achieving superior results compared to RFDiffusionAA. While these models excel in protein and binding pocket design, they primarily focus on static protein(-ligand) interactions and lack the ability to model the chemical transformations involved in enzyme-substrate interactions. This limitation may reduce their accuracy and generalizability in designing enzyme pockets for catalytic reactions.

## D  CO-EVOLUTIONARY MSA TRANSFORMER

Co-evolution captures the dynamic relationship between an enzyme and its substrate during a catalytic reaction. AlphaFold2 (Jumper et al., 2021) has demonstrated the critical importance of leveraging protein evolution, specifically through multiple sequence alignments (MSA) across protein sequences, to enhance a model's generalizability and expressive power. Previous works, such as MSA Transformer (Rao et al., 2021) and EvoFormer (Jumper et al., 2021), have focused on encoding and learning protein evolution from MSA results. Proper co-evolution encodings of enzymes and reactions are essential for capturing the dynamic changes that occur during catalytic processes, not only in our EnzymeFlow model but in other models as well.

Figure 9: Enzyme-reaction co-evolution and tokenized representation.

### D.1  CO-EVOLUTION VOCABULARY

We provide our co-evolution dictionary for tokenization and encoding following:

```
{'<pad>': 0, ' ': 1, '#': 2, '%': 3, '(': 4, ')': 5, '*': 6, '+': 7, '-': 8, '.': 9,
 '/': 10, '0': 11, '1': 12, '2': 13, '3': 14, '4': 15, '5': 16, '6': 17, '7': 18,
 '8': 19, '9': 20, '=': 21, '>': 22, '@': 23, 'A': 24, 'B': 25, 'C': 26, 'D': 27,
 'E': 28, 'F': 29, 'G': 30, 'H': 31, 'I': 32, 'J': 33, 'K': 34, 'L': 35, 'M': 36,
 'N': 37, 'O': 38, 'P': 39, 'Q': 40, 'R': 41, 'S': 42, 'T': 43, 'U': 44, 'V': 45,
 'W': 46, 'X': 47, 'Y': 48, 'Z': 49, '[': 50, '\\': 51, ']': 52, 'c': 53, 'e': 54,
 'g': 55, 'i': 56, 'l': 57, 'n': 58, 'o': 59, 'r': 60, 's': 61, 'u': 62, '<unk>': 63}
```

Figure 10: EnzymeFlow co-evolution dictionary.

### D.2  coEvoFormer IMPLEMENTATION

Here, we introduce a new co-evolutionary MSA transformer, coEvoFormer. The co-evolution of an enzyme-reaction pair is represented by a matrix $U \in \mathbb{R}^{N_{\text{MSA}} \times N_{\text{token}}}$, which combines the MSA results of enzyme sequences and reaction SMILES (illustrated in Fig. 3(3d)). In this matrix, $N_{\text{MSA}}$ denotes the number of MSA sequences, and $N_{\text{token}}$ denotes the length of the preserved MSA alignment. Each element $u^{mn} \in \{1, \ldots, 64\} \cup \{\times\}$ in $U$ represents a tokenized character from our co-evolution vocabulary (provided in App. D.1), with $\times$ indicating the *masking state*. The coEvoFormer takes the co-evolution matrix $U$ as input and outputs an embedded co-evolution representation $H_U \in \mathbb{R}^{N_{\text{MSA}} \times N_{\text{token}} \times D_{H_U}}$, where $D_{H_U}$ denotes the hidden dimension size.

The code for coEvoFormer follows directly:

```python
import math, copy
import numpy as np

import torch
import torch.nn as nn
import torch.nn.functional as F
from torch.autograd import Variable

## Co-Evolution Transformer (coEvoFormer)

## (12) Layer Norm
class ResidualNorm(nn.Module):
    def __init__(self, size, dropout):
        super(ResidualNorm, self).__init__()
        self.norm = LayerNorm(size)
        self.dropout = nn.Dropout(dropout)

    def forward (self, x, sublayer):
        return x + self.dropout(sublayer(self.norm(x)))

## (11) Residual Norm
class LayerNorm(nn.Module):
    def __init__(self, features, eps=1e-6):
        super(LayerNorm, self).__init__()
        self.a_2 = nn.Parameter(torch.ones(features))
        self.b_2 = nn.Parameter(torch.zeros(features))
        self.eps = eps

    def forward(self, x):
        mean = x.mean(-1, keepdim=True)
        std = x.std(-1, keepdim=True)
        x = self.a_2 * (x - mean) / (std + self.eps) + self.b_2
        return x

## (10) 2-layer MLP
class MLP(nn.Module):
    def __init__(self, model_depth, ff_depth, dropout):
        super(MLP, self).__init__()
        self.w1 = nn.Linear(model_depth, ff_depth)
        self.w2 = nn.Linear(ff_depth, model_depth)
        self.dropout = nn.Dropout(dropout)
        self.silu = nn.SiLU()

    def forward(self, x):
        return self.w2(self.dropout(self.silu(self.w1(x))))

## (9) Attention
def attention(Q,K,V, mask=None):
    dk = Q.size(-1)
    T = (Q @ K.transpose(-2, -1))/math.sqrt(dk)
    if mask is not None:
        T = T.masked_fill_(mask.unsqueeze(1)==0, -1e9)
    T = F.softmax(T, dim=-1)
    return T @ V

## (8) Multi-Head Attention
class MultiHeadAttention(nn.Module):
    def __init__ (self,
                    num_heads,
                    embed_dim,
                    bias=False
                    ):
        super(MultiHeadAttention, self).__init__()
        self.num_heads = num_heads
        self.dk = embed_dim//num_heads
        self.WQ = nn.Linear(embed_dim, embed_dim, bias=bias)
        self.WK = nn.Linear(embed_dim, embed_dim, bias=bias)
        self.WV = nn.Linear(embed_dim, embed_dim, bias=bias)
        self.WO = nn.Linear(embed_dim, embed_dim, bias=bias)

    def forward(self, x, kv, mask=None):
        batch_size = x.size(0)
        Q = self.WQ(x ).view(batch_size, -1, self.num_heads, self.dk).transpose(1,2)
        K = self.WK(kv).view(batch_size, -1, self.num_heads, self.dk).transpose(1,2)
        V = self.WV(kv).view(batch_size, -1, self.num_heads, self.dk).transpose(1,2)
```

```python
        if mask is not None:
            if len(mask.shape) == 2:
                mask = torch.einsum('bi,bj->bij', mask, mask)
        x = attention(Q, K, V, mask=mask)

        x = x.transpose(1, 2).contiguous().view(batch_size, -1, self.num_heads*self.dk)
        return self.WO(x)

## (7) Positional Embedding
class PositionalEncoding(nn.Module):
    def __init__(self, model_depth, max_len=5000):
        super(PositionalEncoding, self).__init__()

        pe = torch.zeros(max_len, model_depth)
        position = torch.arange(0.0, max_len).unsqueeze(1)
        div_term = torch.exp(torch.arange(0.0, model_depth, 2) *
                             -(math.log(10000.0) / model_depth))
        pe[:, 0::2] = torch.sin(position * div_term)
        pe[:, 1::2] = torch.cos(position * div_term)
        pe = pe.unsqueeze(0)
        self.register_buffer('pe', pe)

    def forward(self, x):
        return x + Variable(self.pe[:, :x.size(1)], requires_grad=False)

## (6) Embedding
class Embedding(nn.Module):
    def __init__(self, vocab_size, model_depth):
        super(Embedding, self).__init__()
        self.lut = nn.Embedding(vocab_size, model_depth)
        self.model_depth = model_depth
        self.positional = PositionalEncoding(model_depth)

    def forward(self, x):
        emb = self.lut(x) * math.sqrt(self.model_depth)
        return self.positional(emb)

## (5) Encoder Layer
class EncoderLayer(nn.Module):
    def __init__(self,
                 n_heads,
                 model_depth,
                 ff_depth,
                 dropout=0.0
                 ):
        super(EncoderLayer, self).__init__()
        self.self_attn = MultiHeadAttention(embed_dim=model_depth, num_heads=n_heads)
        self.resnorm1 = ResidualNorm(model_depth, dropout)
        self.ff = MLP(model_depth, ff_depth, dropout)
        self.resnorm2 = ResidualNorm(model_depth, dropout)

    def forward(self, x, mask):
        x = self.resnorm1(x, lambda arg: self.self_attn(arg, arg, mask))
        x = self.resnorm2(x, self.ff)
        return x

## (4) Encoder
class Encoder(nn.Module):
    def __init__ (self,
                  n_layers,
                  n_heads,
                  model_depth,
                  ff_depth,
                  dropout
                  ):
        super(Encoder, self).__init__()
        self.layers = nn.ModuleList([EncoderLayer(n_heads, model_depth, ff_depth, dropout) for
        i in range(n_layers)])
        self.lnorm = LayerNorm(model_depth)

    def forward(self, x, mask):
        for layer in self.layers:
            x = layer(x, mask)
        return self.lnorm(x)
```

```python
## (3)Generator
class Generator(nn.Module):
    def __init__(self,
                 model_depth,
                 vocab_size
                 ):
        super(Generator, self).__init__()
        self.ff = nn.Linear(model_depth, vocab_size)

    def forward(self, x):
        return F.log_softmax(self.ff(x), dim=-1)

## (2)coEvoEmbedder
class CoEvoEmbedder(nn.Module):
    def __init__(self,
                 vocab_size,
                 n_layers=2,
                 n_heads=4,
                 model_depth=64,
                 ff_depth=64,
                 dropout=0.0,
                 ):
        super(CoEvoFormer, self).__init__()

        self.model_depth = model_depth
        self.encoder = Encoder(n_layers=n_layers,
                               n_heads=n_heads,
                               model_depth=model_depth,
                               ff_depth=ff_depth,
                               dropout=dropout,
                               )

        if vocab_size is not None:
            if isinstance(vocab_size, int):
                self.set_vocab_size(vocab_size)

            else:
                self.set_vocab_size(vocab_size[0], vocab_size[1])

    def set_vocab_size(self, src_vocab_size):
        self.src_embedder = Embedding(src_vocab_size, self.model_depth)
        self.generator = Generator(self.model_depth, src_vocab_size)

        for p in self.parameters():
            if p.dim() > 1:
                nn.init.xavier_uniform_(p)

    def forward(self, src, src_mask=None):
        enc_out = self.encoder(self.src_embedder(src), src_mask)

        return enc_out

## (1)coEvoFormer
class CoEvoFormer(nn.Module):
    def __init__(self, model_conf):
        super(CoEvoFormer, self).__init__()
        torch.set_default_dtype(torch.float32)
        self._model_conf = model_conf
        self._msa_conf = model_conf.msa

        self.msa_encoder = CoEvoEmbedder(
                        vocab_size=self._msa_conf.num_msa_vocab,
                        n_layers=self._msa_conf.msa_layers,
                        n_heads=self._msa_conf.msa_heads,
                        model_depth=self._msa_conf.msa_embed_size,
                        ff_depth=self._msa_conf.msa_hidden_size,
                        dropout=self._model_conf.dropout,
                        )

        self.col_attn = MultiHeadAttention(
                num_heads=self._msa_conf.msa_heads,
                embed_dim=self._msa_conf.msa_embed_size,
            )

        self.row_attn = MultiHeadAttention(
                num_heads=self._msa_conf.msa_heads,
                embed_dim=self._msa_conf.msa_embed_size,
            )
```

```
241  def forward(
242      self,
243      msa_feature,
244      msa_mask=None,
245  ):
246      bs, n_msa, n_token = msa_feature.size()
247      msa_feature = msa_feature.reshape(bs*n_msa, n_token)
248      msa_embed = self.msa_encoder(msa_feature).reshape(bs, n_msa, n_token, -1)
249      msa_embed = msa_embed.transpose(1, 2).reshape(bs*n_token, n_msa, -1)
250
251      if msa_mask is not None:
252          msa_mask = msa_mask.transpose(1, 2).reshape(bs*n_token, n_msa)
253
254      msa_embed = self.col_attn(msa_embed, msa_embed, mask=msa_mask).reshape(bs, n_token,
         n_msa, -1).transpose(1, 2)
255      msa_embed = msa_embed.reshape(bs*n_msa, n_token, -1)
256
257      if msa_mask is not None:
258          msa_mask = msa_mask.reshape(bs, n_token, n_msa)
259          msa_mask = msa_mask.transpose(1, 2).reshape(bs*n_msa, n_token)
260
261      msa_embed = self.row_attn(msa_embed, msa_embed, mask=msa_mask).reshape(bs, n_msa,
         n_token, -1)
262
263      return msa_embed
```

Listing 1: Pytorch Implementation of coEvoFormer.

# E MOLECULE GNN

## E.1 3D MOLECULE GNN

The 3D molecule GNN plays a crucial role in EnzymeFlow. During the structure-based hierarchical pre-training, it encodes ligand molecule representations, learning the constrained geometry between protein binding pockets and ligand molecules. This pre-training process makes the 3D molecule GNN transferable. When the flow model is fine-tuned, the 3D molecule GNN is also fine-tuned, transferring its prior knowledge about ligand molecules to substrate molecules in enzyme-catalyzed reactions. This allows for substrate-specific encodings while leveraging the knowledge learned from protein-ligand interactions.

Consider a molecule $l_s$ with $N_{l_s}$ atoms; this could be a ligand conformation in a protein-ligand pair or a substrate conformation in an enzyme-substrate pair. The molecule $l_s$ can be viewed as a set of atomic point clouds in 3D Euclidean space, where each atom is characterized by its atomic type. There is a distance relationship between each atom pair in the point cloud, which can be processed as bonding features. In our 3D molecule GNN, we use a radial basis function to process these pairwise atomic distances, a technique commonly employed to ensure equivariance and invariance in model design (Hua et al., 2023; Zhang et al., 2024a;b). The 3D molecule GNN takes a molecule conformation $l_s$ as input and outputs an embedded molecule representation $H_{l_s} \in \mathbb{R}^{N_{l_s} \times D_{H_{l_s}}}$, where $D_{H_{l_s}}$ denotes the hidden dimension size.

The code for 3D Molecule GNN follows directly:

```
1   import math
2   import numpy as np
3
4   import torch
5   import torch.nn as nn
6   from torch.nn import functional as F
7
8   ## (1)3D Molecule GNN
9   class MolEmbedder3D(nn.Module):
10      def __init__(self, model_conf):
11          super(MolEmbedder3D, self).__init__()
12          torch.set_default_dtype(torch.float32)
13          self._model_conf = model_conf
14          self._embed_conf = model_conf.embed
15
16          node_embed_dims = self._model_conf.num_atom_type
17          node_embed_size = self._model_conf.node_embed_size
18          self.node_embedder = nn.Sequential(
19              nn.Embedding(node_embed_dims, node_embed_size, padding_idx=0),
20              nn.SiLU(),
21              nn.Linear(node_embed_size, node_embed_size),
```

```
22              nn.LayerNorm(node_embed_size),
23          )
24
25      self.node_aggregator = nn.Sequential(
26          nn.Linear(node_embed_size + self._model_conf.edge_embed_size, node_embed_size),
27          nn.SiLU(),
28          nn.Linear(node_embed_size, node_embed_size),
29          nn.SiLU(),
30          nn.Linear(node_embed_size, node_embed_size),
31          nn.LayerNorm(node_embed_size),
32          )
33
34      self.dist_min = self._model_conf.ligand_rbf_d_min
35      self.dist_max = self._model_conf.ligand_rbf_d_max
36      self.num_rbf_size = self._model_conf.num_rbf_size
37      self.edge_embed_size = self._model_conf.edge_embed_size
38
39      self.edge_embedder = nn.Sequential(
40          nn.Linear(self.num_rbf_size + node_embed_size + node_embed_size, self.
     edge_embed_size),
41          nn.SiLU(),
42          nn.Linear(self._model_conf.edge_embed_size, self._model_conf.edge_embed_size),
43          nn.SiLU(),
44          nn.Linear(self._model_conf.edge_embed_size, self._model_conf.edge_embed_size),
45          nn.LayerNorm(self._model_conf.edge_embed_size),
46          )
47
48      mu = torch.linspace(self.dist_min, self.dist_max, self.num_rbf_size)
49      self.mu = mu.reshape([1, 1, 1, -1])
50
51      self.sigma = (self.dist_max - self.dist_min) / self.num_rbf_size
52
53  # Distance function -- pair-wise distance computation
54  def coord2dist(self, coord, edge_mask):
55      n_batch, n_atom = coord.size(0), coord.size(1)
56      radial = torch.sum((coord.unsqueeze(1) - coord.unsqueeze(2)) ** 2, dim=-1)
57      dist = torch.sqrt(
58              radial + 1e-10
59          ) * edge_mask
60
61      radial = radial * edge_mask
62      return radial, dist
63
64  # RBF function -- distance encoding
65  def rbf(self, dist):
66      dist_expand = torch.unsqueeze(dist, -1)
67      _mu = self.mu.to(dist.device)
68      rbf = torch.exp(-(((dist_expand - _mu) / self.sigma) ** 2))
69      return rbf
70
71  def forward(
72      self,
73      ligand_atom,
74      ligand_pos,
75      edge_mask,
76  ):
77      num_batch, num_atom = ligand_atom.shape
78
79      # Atom Embbedding
80      node_embed = self.node_embedder(ligand_atom)
81
82      # Edge Feature Computation
83      radial, dist = self.coord2dist(
84                      coord=ligand_pos,
85                      edge_mask=edge_mask,
86                  )
87      edge_embed = self.rbf(dist) * edge_mask[..., None]
88      src_node_embed = node_embed.unsqueeze(1).repeat(1, num_atom, 1, 1)
89      tar_node_embed = node_embed.unsqueeze(2).repeat(1, 1, num_atom, 1)
90      edge_embed = torch.cat([src_node_embed, tar_node_embed, edge_embed], dim=-1)
91
92      # Edge Embedding
93      edge_embed = self.edge_embedder(edge_embed.to(torch.float))
94
95      # Message-Passing
96      src_node_agg = (edge_embed.sum(dim=1) / (edge_mask[..., None].sum(dim=1)+1e-10)) *
     ligand_atom.clamp(max=1.)[..., None]
97      src_node_agg = torch.cat([node_embed, src_node_agg], dim=-1)
98
99      # Residue Connection
100     node_embed = node_embed + self.node_aggregator(src_node_agg)
```

```
101
102        return node_embed, edge_embed
```

Listing 2: Pytorch Implementation of 3D Molecule GNN.

## E.2 2D MOLECULE GNN

Like the 3D molecule GNN, the 2D molecule GNN is also important in our EnzymeFlow implementation. In an enzyme-catalyzed reaction, the substrate molecule is transformed into a product molecule, with enzyme-substrate interactions driving this chemical transformation. The 2D molecule GNN plays a key role in modeling and encoding this transformation during the catalytic process, making it equally important as our use of co-evolutionary dynamics. While the 3D molecule GNN encodes the substrate, the 2D molecule GNN encodes the product, guiding the design of the enzyme catalytic pocket.

Consider a product molecule $l_p$ with $N_{l_p}$ atoms in a catalytic reaction. This molecule can be represented as a graph, where nodes correspond to atoms and edges represent bonds. In our 2D molecule GNN, we use fingerprints with attention mechanisms (Xiong et al., 2019) to facilitate message passing between atoms, enabling effective communication across the molecule. The 2D molecule GNN takes this molecular graph $l_p$ as input and outputs an embedded molecule representation $H_{l_p} \in \mathbb{R}^{N_{l_p} \times D_{H_{l_p}}}$, where $D_{H_{l_p}}$ denotes the hidden dimension size.

The code for 2D Molecule GNN follows directly:

```
1  import torch
2  import torch.nn as nn
3  from torch_geometric.nn.models import AttentiveFP
4
5  ## (1)2D Molecule GNN
6  class MolEmbedder2D(nn.Module):
7      def __init__(self, model_conf):
8          super(MolEmbedder2D, self).__init__()
9          torch.set_default_dtype(torch.float32)
10         self._model_conf = model_conf
11
12         self.node_embed_dims = self._model_conf.mpnn.mpnn_node_embed_size
13         self.edge_embed_dims = self._model_conf.mpnn.mpnn_edge_embed_size
14
15         self.node_embedder = nn.Sequential(
16             nn.Embedding(self._model_conf.num_atom_type, self.node_embed_dims),
17             nn.SiLU(),
18             nn.Linear(self.node_embed_dims, self.node_embed_dims),
19             nn.LayerNorm(self.node_embed_dims),
20             )
21
22         self.edge_embedder = nn.Sequential(
23             nn.Embedding(self._model_conf.mpnn.num_edge_type, self.edge_embed_dims),
24             nn.SiLU(),
25             nn.Linear(self.edge_embed_dims, self.edge_embed_dims),
26             nn.LayerNorm(self.edge_embed_dims),
27             )
28
29         # Message Passing with Atttention and Fingerprint
30         self.mpnn = AttentiveFP(
31                 in_channels=self.node_embed_dims,
32                 hidden_channels=self.node_embed_dims,
33                 out_channels=self.node_embed_dims,
34                 edge_dim=self.edge_embed_dims,
35                 num_layers=self._model_conf.mpnn.mpnn_layers,
36                 num_timesteps=self._model_conf.mpnn.n_timesteps,
37                 dropout=self._model_conf.mpnn.dropout,
38                 )
39
40     # Dense Edge Matrix to Sparse Edge Matrix
41     def dense_to_sparse(
42         self,
43         mol_atom,
44         mol_edge,
45         mol_edge_feat,
46         mol_atom_mask,
47         mol_edge_mask,
48     ):
49         mol_atom_list = mol_atom[mol_atom_mask]
50         mol_edge_feat_list = mol_edge_feat[mol_edge_mask]
```

```
51
52      if mol_edge.size(dim=1) == 2:
53          mol_edge = mol_edge.transpose(1,2)
54      mol_edge_list = [edge[mask] for edge, mask in zip(mol_edge, mol_edge_mask)]
55
56      n_nodes = mol_atom_mask.sum(dim=1, keepdim=True)
57      cum_n_nodes = torch.cumsum(n_nodes, dim=0)
58      new_mol_edge_list = [mol_edge_list[0]]
59      for edge, size in zip(mol_edge_list[1:], cum_n_nodes[:-1]):
60          new_mol_edge = edge + size
61          new_mol_edge_list.append(new_mol_edge)
62
63      new_mol_edge_list = torch.cat(new_mol_edge_list, dim=0)
64
65      if new_mol_edge_list.size(dim=1) == 2:
66          new_mol_edge_list = new_mol_edge_list.transpose(1,0)
67
68      idx = 0
69      batch_mask = []
70      for size in n_nodes:
71          batch_mask.append(torch.zeros(size, dtype=torch.long) + idx)
72          idx += 1
73      batch_mask = torch.cat(batch_mask).to(mol_atom.device)
74
75      return mol_atom_list, new_mol_edge_list, mol_edge_feat_list, batch_mask
76
77  def forward(
78      self,
79      mol_atom,
80      mol_edge,
81      mol_edge_feat,
82      mol_atom_mask,
83      mol_edge_mask,
84  ):
85      n_batch = mol_atom.size(0)
86
87      mol_atom_mask = mol_atom_mask.bool()
88      mol_edge_mask = mol_edge_mask.bool()
89      mol_atom, mol_edge, mol_edge_feat, batch_mask = self.dense_to_sparse(mol_atom,
       mol_edge, mol_edge_feat, mol_atom_mask, mol_edge_mask)
90      assert mol_edge.size(1) == mol_edge_feat.size(0)
91
92      # Atom Embedding
93      mol_atom = self.node_embedder(mol_atom)
94
95      # Edge Embedding
96      mol_edge_feat = self.edge_embedder(mol_edge_feat)
97
98      # Message-Passing
99      mol_rep = self.mpnn(mol_atom, mol_edge, mol_edge_feat, batch_mask)
100
101     return mol_rep
```

Listing 3: Pytorch Implementation of 2D Molecule GNN.

## F  VECTOR FIELD COMPUTATION AND SAMPLING

Here, we describe how to compute vectors fields and perform sampling for catalytic pocket residues frames, EC-class, as well as the enzyme-reaction co-evolution.

### F.1  BACKGROUND

**Catalytic Pocket Frame.** We refer to the protein structure as the backbone atomic coordinates of each residue. A pocket of length $N_r$ can be parameterized into SE(3) residue frames $\{(x^i, r^i, c^i)\}_{i=1}^{N_r}$, where $x^i \in \mathbb{R}^3$ represents the position (translation) of the $C_\alpha$ atom of the $i$-th residue, $r^i \in SO(3)$ is a rotation matrix defining the local frame relative to a global reference frame, and $c^i \in \{1, \dots, 20\} \cup \{\times\}$ denotes the amino acid type, with additional $\times$ indicating a *masking state* of the amino acid type. We refer to the residue block as $T^i = (x^i, r^i, c^i)$, and the entire pocket is described by a set of residues $\mathbf{T} = \{T^i\}_{i=1}^{N_r}$. Additionally, we denote the graph representations of substrate and product molecules in the catalytic reaction as $l_s$ and $l_p$, respectively. An enzyme-reaction pair can therefore be described as $(\mathbf{T}, l_s, l_p)$. For simplicity, we omit $i$.

**EC-Class.** An EC-class is denoted as $y_{ec} \in \{1, \dots, 7\} \cup \{\times\}$, with $\times$ indicating the *masking state*.

**Co-evolution.** The co-evolution of an enzyme-reaction pair is represented by a matrix $U \in \mathbb{R}^{N_{\mathrm{MSA}} \times N_{\mathrm{token}}}$, which combines the MSA results of enzyme sequences and reaction SMILES, where $N_{\mathrm{MSA}}$ denotes the number of MSA sequences and $N_{\mathrm{token}}$ denotes the length of the MSA alignment preserved. And each element $u^{mn} \in \{1, \ldots, 64\} \cup \{\times\}$ in $U$ denotes a tokenized character from our co-evolution vocabulary, with additional $\times$ indicating the *masking state*.

**Vector Field.** flow matching describes a process where a flow transforms a simple distribution $p_0$ into the target data distribution $p_1$ (Lipman et al., 2022). The goal in flow matching is to train a neural network $v_\theta(\epsilon_t, t)$ that approximates the vector field $u_t(\epsilon)$, which measures the transformation of the distribution $p_t(\epsilon_t)$ as it evolves toward $p_1(\epsilon_t)$ over time $t \in [0, 1)$. The process is optimized using a regression loss defined as $\mathcal{L}_{\mathrm{FM}} = \mathbb{E}_{t \sim \mathcal{U}[0,1], p_t(\epsilon_t)} \|v_\theta(\epsilon_t, t) - u_t(\epsilon)\|^2$. However, directly computing $u_t(\epsilon)$ is often intractable in practice. Instead, a conditional vector field $u_t(\epsilon | \epsilon_1)$ is defined, and the conditional flow matching objective is computed as $\mathcal{L}_{\mathrm{CFM}} = \mathbb{E}_{t \sim \mathcal{U}[0,1], p_t(\epsilon_t)} \|v_\theta(\epsilon_t, t) - u_t(\epsilon | \epsilon_1)\|^2$. Notably, $\nabla_\theta \mathcal{L}_{\mathrm{FM}} = \nabla_\theta \mathcal{L}_{\mathrm{CFM}}$.

During inference or sampling, an ODEsolver, *e.g.*, Euler method, is typically used to solve the ODE governing the flow, expressed as $\epsilon_1 = \mathtt{ODEsolver}(\epsilon_0, v_\theta, 0, 1)$, where $\epsilon_0$ is the initial data and $\epsilon_1$ is the generated data. In actual training, rather than directly predicting the vector fields, it is more common to use the neural network to predict the final state at $t = 1$, then interpolates to calculate the vector fields. This approach has been shown to be more efficient and effective for network optimization (Yim et al., 2023a; Bose et al., 2023; Campbell et al., 2024).

### F.2 Continuous Variable Trajectory

Given the predictions for translation $\hat{x}_1$ and rotation $\hat{r}_1$ at $t = 1$, we interpolate and their corresponding vector fields are computed as follows:

$$v_\theta(x_t, t) = \frac{\hat{x}_1 - x_t}{1 - t}, \quad v_\theta(r_t, t) = \frac{\log_{r_t} \hat{r}_1}{1 - t}. \tag{9}$$

The sampling or trajectory can then be computed using Euler steps with a step size $\Delta t$, as follows:

$$x_{t+\Delta t} = x_t + v_\theta(x_t, t) \cdot \Delta t, \quad r_{t+\Delta t} = r_t + v_\theta(r_t, t) \cdot \Delta t, \tag{10}$$

where the prior of $x_0, r_0$ are chosen as the uniform distribution on $\mathbb{R}^3$ and SO(3), respectively.

### F.3 Discrete Variable Trajectory

For the discrete variables, including amino acid types, EC-class, and co-evolution, we follow Campbell et al. (2024) to use continuous time Markov chains (CTMC).

**Continuous Time Markov Chain.** A sequence trajectory $\epsilon_t$ over time $t \in [0, 1]$ that follows a CTMC alternates between resting in its current state and periodically jumping to another randomly chosen state. The frequency and destination of the jumps are determined by the rate matrix $R_t \in \mathbb{R}^{N \times N}$ with the constraint its off-diagonal elements are non-negative. The probability of $\epsilon_t$ jumping to a different state $s$ follows $R_t(\epsilon_t, s)\mathrm{d}t$ for the next infinitesimal time step $\mathrm{d}t$. We can express the transition probability as

$$p_{t+\mathrm{d}t}(s | \epsilon_t) = \delta\{\epsilon_t, s\} + R_t(\epsilon_t, s)\mathrm{d}t, \tag{11}$$

where $\delta(\mathrm{a}, \mathrm{b})$ is the Kronecker delta, equal to 1 if $\mathrm{a} = \mathrm{b}$ and 0 if $\mathrm{a} \neq \mathrm{b}$, and $R_t(\epsilon_t, \epsilon_t) = -\sum_{\gamma \neq \epsilon}(\epsilon_t, \gamma)$ (Campbell et al., 2024). Therefore, $p_{t+\mathrm{d}t}$ is a Categorical distribution with probabilities $\delta(\epsilon_t, \cdot) + R_t(\epsilon_t, \cdot)\mathrm{d}t$ with notation $s \sim \mathrm{Cat}(\delta(\epsilon_t, s) + R_t(\epsilon_t, s)\mathrm{d}t)$.

For finite time intervals $\Delta t$, a sequence trajectory can be simulated with Euler steps following:

$$\epsilon_{t+\Delta t} \sim \mathrm{Cat}(\delta(\epsilon_t, \epsilon_{t+\Delta t}) + R_t(\epsilon_t, \epsilon_{t+\Delta t})\Delta t). \tag{12}$$

The rate matrix $R_t$ along with an initial distribution $p_0$ define CTMC. Furthermore, the probability flow $p_t$ is the marginal distribution of $\epsilon_t$ at every time $t$, and we say the rate matrix $R_t$ generates $p_t$ if $\partial_t p_t = R_t^T p_t, \forall t \in [0, 1]$.

In the actual training, Campbell et al. (2024) show that we can train a neural network to approximate the true denoising distribution using the standard cross-entropy:

$$\mathcal{L}_{\mathrm{CE}} = \mathbb{E}_{t \sim \mathcal{U}[0,1], p_t(\epsilon_t)}[\log p_\theta(\epsilon_1 | \epsilon_t)], \tag{13}$$

which leads to our neural network objectives for amino acid types, EC-class, and co-evolution as:

$$\mathcal{L}_{\text{aa}} = \mathbb{E}_{t \sim \mathcal{U}[0,1], p_t(c_t)}[\log p_\theta(c_1|c_t)], \mathcal{L}_{\text{ec}} = \mathbb{E}_{t \sim \mathcal{U}[0,1], p_t(y_{\text{ec}_t})}[\log p_\theta(y_{\text{ec}_1}|y_{\text{ec}_t})],$$
$$\mathcal{L}_{\text{coevo}} = \mathbb{E}_{t \sim \mathcal{U}[0,1], p_t(u_t)}[\log p_\theta(u_1|u_t)]. \tag{14}$$

**Rate Matrix for Inference.** The conditional rate matrix $R_t(\epsilon_t, s|s_1)$ generates the conditional flow $p_t(\epsilon_t|\epsilon_1)$. And $R_t(\epsilon_t, s) = \mathbb{E}_{p_1(\epsilon_1|\epsilon_t)}[R_t(\epsilon_t, s|\epsilon_1)]$, for which the expectation is taken over $p_1(\epsilon_1|\epsilon_t) = \frac{p_t(\epsilon_t|\epsilon_1)p_1(\epsilon_1)}{p_t(\epsilon_t)}$. With the conditional rate matrix, the sampling can be performed:

$$R_t(\epsilon_t, \cdot) \leftarrow \mathbb{E}_{p_1(\epsilon_1|\epsilon_t)}[R_t(\epsilon_t, \cdot|\epsilon_1)],$$
$$\epsilon_{t+\Delta t} \sim \text{Cat}(\delta(\epsilon_t, \epsilon_{t+\Delta t}) + R_t(\epsilon_t, \epsilon_{t+\Delta t})\Delta t). \tag{15}$$

The rate matrix generates the probability flow for discrete variables.

Campbell et al. (2024) define the conditional rate matrix starting with

$$R_t(\epsilon_t, s|\epsilon_t) = \frac{\text{ReLU}(\partial_t p_t(s|\epsilon_1) - \partial_t p_t(\epsilon_t|\epsilon_1))}{N \cdot p_t(\epsilon_t|\epsilon_1)}. \tag{16}$$

In practice, the closed-form of conditional rate matrix with *masking state* ✗ is defined as:

$$R_t(\epsilon_t, s|\epsilon_1) = \frac{\delta(\epsilon_1, s)}{1-t}\delta(\epsilon_t, \text{✗}). \tag{17}$$

With the definition of the conditional rate matrix $R_t(\epsilon_t, s|\epsilon_1)$, we can perform sampling and inference for amino acid types, EC-class, and co-evolution following:

$$c_{t+\Delta t} \sim \text{Cat}(\delta(c_t, c_{t+\Delta t}) + R_t(c_t, c_{t+\Delta t}|v_\theta(c_t, t)) \cdot \Delta t),$$
$$y_{\text{ec}_{t+\Delta t}} \sim \text{Cat}(\delta(y_{\text{ec}_t}, y_{\text{ec}_{t+\Delta t}}) + R_t(y_{\text{ec}_t}, y_{\text{ec}_{t+\Delta t}}|v_\theta(y_{\text{ec}_t}, t)) \cdot \Delta t), \tag{18}$$
$$u_{t+\Delta t} \sim \text{Cat}(\delta(u_t, u_{t+\Delta t}) + R_t(u_t, u_{t+\Delta t}|v_\theta(u_t, t)) \cdot \Delta t).$$

## G    ENZYMEFLOW SE(3)-EQUIVARIANCE

**Theorem.** *Let $\phi$ denote an SE(3) transformation. The catalytic pocket design in EnzymeFlow, represented as $p_\theta(\mathbf{T}|l_s)$, is SE(3)-equivariant, meaning that $p_\theta(\phi(\mathbf{T})|\phi(l_s)) = p_\theta(\mathbf{T}|l_s)$, where $\mathbf{T}$ represents the generated catalytic pocket, and $l_s$ denotes the substrate conformation.*

*Proof.* Given an SE(3)-invariant prior, such that $p(\mathbf{T}_0, l_s) = p(\phi(\mathbf{T}_0), \phi(l_s))$, and an SE(3)-equivariant transition state for each time step $t$ via an SE(3)-equivariant neural network, such that $p_\theta(\mathbf{T}_{t+\Delta t}, l_s) = p_\theta(\phi(\mathbf{T}_{t+\Delta t}), \phi(l_s))$, it follows that for the total time steps $T$, we have:

$$\begin{aligned}
p_\theta(\phi(\mathbf{T}_1)|\phi(l_s)) &= \int p_\theta(\phi(\mathbf{T}_0, l_s)) \prod_{n=0}^{T-1} p_\theta(\phi(\mathbf{T}_{n\Delta t+\Delta t}, l_s)|\phi(\mathbf{T}_{n\Delta t}, l_s)) \\
&= \int p_\theta(\mathbf{T}_0, l_s) \prod_{n=0}^{T-1} p_\theta(\phi(\mathbf{T}_{n\Delta t+\Delta t}, l_s)|\phi(\mathbf{T}_{n\Delta t}, l_s)) \\
&= \int p_\theta(\mathbf{T}_0, l_s) \prod_{n=0}^{T-1} p_\theta(\mathbf{T}_{n\Delta t+\Delta t}, l_s|\mathbf{T}_{n\Delta t}, l_s) \\
&= p_\theta(\mathbf{T}_1|l_s).
\end{aligned} \tag{19}$$

$\square$

## H    ENZYMEFLOW DATASET STATISTICS

**Data Source.** We construct a curated and validated dataset of enzyme-reaction pairs by collecting data from the Rhea (Bansal et al., 2022), MetaCyc (Caspi et al., 2020), and Brenda (Schomburg et al., 2002) databases. For enzymes in these databases, we exclude entries missing UniProt IDs or protein sequences. For reactions, we apply the following procedures: (1) remove cofactors, small

| Data | Reaction | Enzyme | Substrate | | Product | | Enzyme Commission Class | | | | | | |
|---|---|---|---|---|---|---|---|---|---|---|---|---|---|
| | #reaction | #enzyme | #substrate | #avg atom | #product | #avg atom | EC1 | EC2 | EC3 | EC4 | EC5 | EC6 | EC7 |
| Rawdata | 232520 | 97912 | 7259 | 30.81 | 7664 | 30.34 | 44881 (19.30%) | 75944 (32.66%) | 37728 (16.23%) | 47242 (20.32%) | 8315 (3.58%) | 18281 (7.86%) | 129 (0.06%) |
| 40% Homo | 19379 | 6922 | 4798 | 31.06 | 4897 | 30.24 | 4754 (24.53%) | 5857 (30.22%) | 4839 (24.97%) | 1764 (9.10%) | 759 (3.92%) | 1379 (7.12%) | 27 (0.14%) |
| 50% Homo | 34750 | 13442 | 5675 | 31.45 | 5871 | 30.75 | 8184 (23.55%) | 11174 (32.16%) | 8050 (23.17%) | 3203 (9.22%) | 1357 (3.91%) | 2752 (7.92%) | 30 (0.09%) |
| 60% Homo | 53483 | 22350 | 6112 | 30.95 | 6331 | 30.34 | 11674 (21.83%) | 18419 (34.44%) | 11394 (21.30%) | 5555 (10.39%) | 2194 (4.10%) | 4200 (7.85%) | 47 (0.09%) |
| 80% Homo | 100925 | 43458 | 6619 | 30.46 | 6943 | 29.95 | 21308 (21.11%) | 34344 (34.03%) | 18925 (18.75%) | 14010 (13.88%) | 3901 (3.87%) | 8371 (8.29%) | 66 (0.07%) |
| 90% Homo | 132047 | 55697 | 6928 | 30.32 | 7298 | 29.81 | 28833 (21.84%) | 43287 (32.78%) | 23989 (18.17%) | 20070 (15.20%) | 5015 (3.80%) | 10766 (8.15%) | 87 (0.07%) |

Table 3: EnzymeFill Dataset Statistics.

| Data | Reaction | Enzyme | Substrate | | Product | | Enzyme Commision | | | | | | |
|---|---|---|---|---|---|---|---|---|---|---|---|---|---|
| | #reaction | #enzyme | #substrate | #avg atom | #product | #avg atom | EC1 | EC2 | EC3 | EC4 | EC5 | EC6 | EC7 |
| Rawdata | 232520 | 97912 | 7259 | 30.81 | 7664 | 30.34 | 44881 (19.30) | 75944 (32.66) | 37728 (16.23) | 47242 (20.32) | 8315 (3.58) | 18281 (7.86) | 129 (0.06) |
| Train Data | 53483 | 22350 | 6112 | 30.95 | 6331 | 30.34 | 11674 (21.83) | 18419 (34.44) | 11394 (21.30) | 5555 (10.39) | 2194 (4.10) | 4200 (7.85) | 47 (0.09) |
| Eval Data | 100 | 100 | 100 | 30.7 | 94 | 28.84 | 17 (17.00) | 17 (17.00) | 17 (17.00) | 17 (17.00) | 16 (16.00) | 16 (16.00) | 0 (0.00) |

Table 4: EnzymeFlow Evaluation Data Statistics.

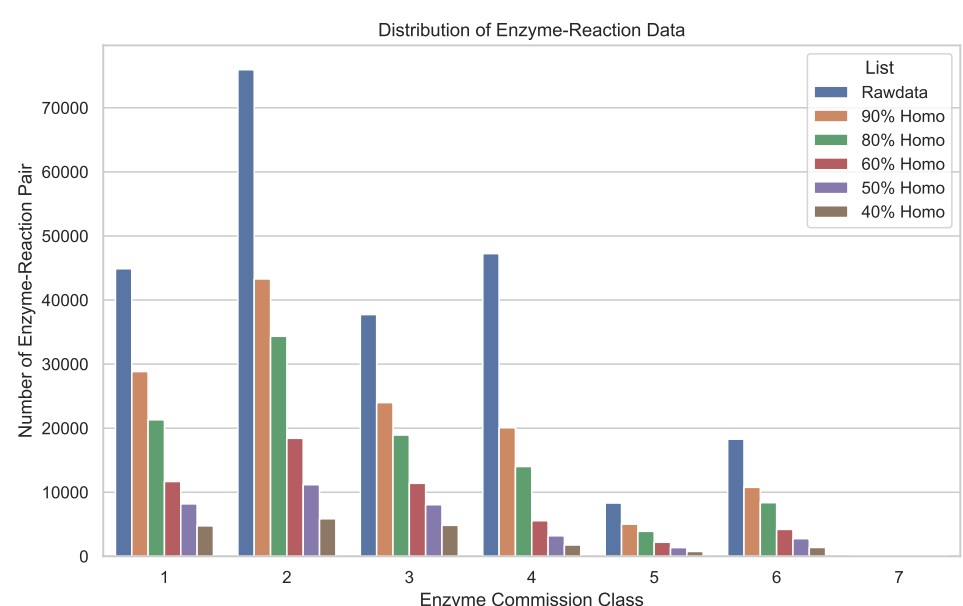

Figure 11: Distribution of enzyme-reaction pairs over EC-class.

ion groups, and molecules that appear in both substrates and products within a single reaction; (2) exclude reactions with more than five substrates or products; and (3) apply OpenBabel (O'Boyle et al., 2011) to standardize molecular SMILES. Ultimately, we obatin a total of $328,192$ enzyme-reaction pairs, comprising $145,782$ unique enzymes and $17,868$ unique reactions.

**Debiasing.** To ensure the quality of catalytic pocket data, we exclude pockets with fewer than 32 residues, resulting in $232,520$ enzyme-reaction pairs. Additionally, enzymes and their catalytic pockets can exhibit significant sequence similarity. When enzymes that are highly similar in sequence appear too frequently in the dataset, they tend to belong to the same cluster or homologous group, which can introduce substantial biases during model training. To mitigate this issue and ensure a more balanced dataset, it is important to reduce the number of homologous enzymes by clustering and selectively removing enzymes from the same clusters. This helps to debias the data and improve the model's generalizability. We perform sequence alignment to cluster enzymes and identify homologous ones (Steinegger & Söding, 2017). We then revise the dataset into five major categories based on enzyme sequence similarity, resulting in: (1) $19,379$ pairs with at most $40\%$ homology, (2) $34,750$ pairs with at most $50\%$ homology, (3) $53,483$ pairs with at most $60\%$ homology, (4) $100,925$ pairs with at most $80\%$ homology, and (5) $132,047$ pairs with at most $90\%$ homology. We provide data statistics, including the EC-class distribution, in Table 3, and visualize the distribution in Figure 11.

From the data, we observe that EC1, EC2, EC3, and EC4 contribute the most enzyme-reaction pairs to our dataset. Specifically, EC1 refers to oxidation/reduction reactions, involving the transfer of hydrogen, oxygen atoms, or electrons from one substance to another. EC2 involves the transfer of a functional group (such as methyl, acyl, amino, or phosphate) from one substance to another. EC3 is associated with the formation of two products from a substrate through hydrolysis, while EC4

involves the non-hydrolytic addition or removal of groups from substrates, potentially cleaving C-C, C-N, C-O, or C-S bonds. Our dataset distribution closely follows the natural enzyme-reaction enzyme commission class distribution, with Transferases (EC2) being the most dominant.

# I WORK IN PROGRESS: ENZYME POCKET-REACTION RECRUITMENT WITH ENZYME CLIP MODEL

In addition to evaluating the catalytic pockets generated from the functional and structural perspectives, we may raise a key question of how we *quantitatively* determine whether the generated pockets can catalyze a specific reaction. To answer it, we are working to train an enzyme-reaction CLIP model using enzyme-reaction pairs (with pocket-specific information) from the 60%-clustered data, excluding the 100 evaluation samples from training. All enzymes not annotated to catalyze a specific reaction are treated as negative samples, following the approach in Yang et al. (2024); Mikhael et al. (2024). For the 100 generated catalytic pockets of each reaction, we select the Top-1 pocket with the highest TM-score for evaluation using the enzyme CLIP model.

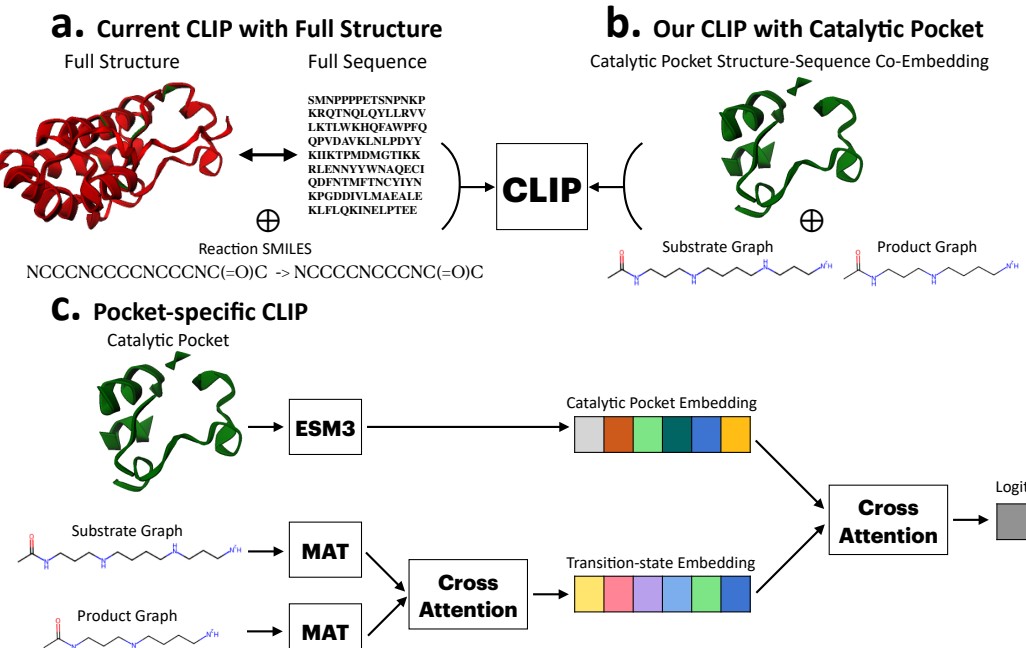

Figure 12: Enzyme-Reaction CLIP model comparison. (a) Existing CLIP models use the full enzyme structure or full enzyme sequence, paired with reaction SMILES as input. (b) Our pocket-specific CLIP model focuses on catalytic pockets, using both their structures and sequences paired with molecular graphs of reactions. The pocket-specific CLIP approach learns from enzyme active sites, which exhibit higher functional concentration. (c) Overview of Pocket-specific CLIP model.

**Pocket-specific CLIP.** Unlike existing methods that typically train on full enzyme structures or sequences (Yu et al., 2023; Mikhael et al., 2024), our pocket-specific CLIP approach is designed to focus specifically on catalytic pockets, including both their structures and sequences, paired with molecular graphs of catalytic reactions (illustrated in Fig. 12). As shown in Fig. 2(b), catalytic pockets are usually the regions that exhibit high functional concentration, while the remaining parts tend to be less functionally important. Therefore, focusing on catalytic pockets is more applicable and effective for enzyme CLIP models. The advantage of the pocket-specific CLIP is that it learns from active sites that are highly meaningful both structurally and sequentially.

We illustrate our pocket-specific enzyme CLIP approach in Fig. 12. In our pocket-specific CLIP model, we encode the pocket structure and sequence using ESM3 (Hayes et al., 2024), and the substrate and product molecular graphs using MAT (Maziarka et al., 2020). Cross-attention is applied to compute the transition state of the reaction, capturing the transformation of the substrate into the product, as proposed in Hua et al. (2024b). This is followed by another cross-attention mechanism to

learn the interactions between the catalytic pocket and the reaction. The model is trained by enforcing high logits for positive enzyme-reaction pairs and low logits for negative enzyme-reaction pairs.

**Metrics.** To evaluate the catalytic ability of the designed pockets for a given reaction, we employ retrieval-based ranking as proposed in Hua et al. (2024b). This ranking-based evaluation ensures fairness and minimizes biases. The metrics include: Top-k Acc, which quantifies the proportion of instances in which the catalytic pocket is ranked within the CLIP's top-k predictions; Mean Rank, which calculates the average position of the pocket in the retrieval list; Mean Reciprocal Rank (MRR), which measures how quickly the pocket is retrieved by averaging the reciprocal ranks of the first correct pocket across all reactions. These metrics help assess whether a catalytic pocket designed for a specific reaction ranks highly in the recruitment list, indicating its potential to catalyze the reaction.

### I.1 Inpainting Catalytic Pocket with ESM3 for Full Enzyme Recruitment

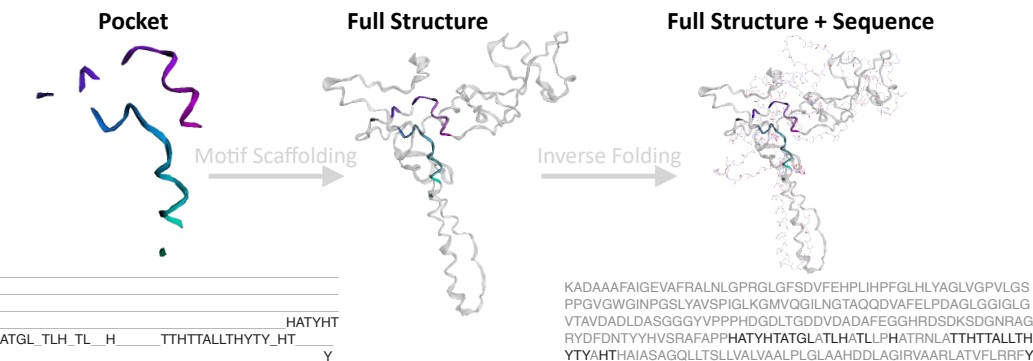

Figure 13: Inpainting catalytic pocket using ESM3.

ESM3 (Hayes et al., 2024) can inpaint missing structures and sequences with functional motifs. In this context, we train a separate full enzyme CLIP model for the enzyme recruitment task. This model is trained using the same 60%-clustered data but incorporates full enzyme structures and sequences. For generated catalytic pockets and those in the evaluation set, we use ESM3 to inpaint them, completing the structures and sequences predicted by ESM3. These ESM3-inpainted enzymes are then evaluated using the full enzyme CLIP model, applying the same retrieval-based ranking metrics as before. We illustrate the catalytic pocket inpainting pipeline in Fig. 13.

In conclusion, we are developing a pocket-specific enzyme CLIP model for pocket-based enzyme recruitment tasks and a full-enzyme CLIP model using ESM3 for inpainting and pocket scaffolding in full enzyme recruitment tasks. However, we recognize that directly using ESM3 for catalytic pocket inpainting lacks domain-specific knowledge, making fine-tuning necessary. To address this, we are working on a fine-tuning open-source large biological model, *e.g.,* Genie2 (Lin et al., 2024), on our EnzymeFill dataset. Genie2, pre-trained on FoldSeek-clustered AlphaFold- and Protein-DataBank proteins for *de novo* protein design and (multi-)motif scaffolding, aligns well with our catalytic pocket scaffolding task. Fine-tuning Genie2 on EnzymeFill will enhance its performance in catalytic pocket inpainting. The development of EnzymeFlow, aimed at achieving an AI-driven automated enzyme design platform, is discussed in App. A.

## J   RFDiffusionAA-design vs. EnzymeFlow-design

In Fig. 14, we visualize and compare the RFDiffusionAA-generated pockets (Krishna et al., 2024) with EnzymeFlow-generated catalytic pockets, both aligned to the ground-truth reference pockets. In RFDiffusionAA, the generation is conditioned on the substrate conformation, and the pocket sequence is computed post hoc using LigandMPNN (Dauparas et al., 2023). In contrast, EnzymeFlow conditions the generation on the reaction, with the pocket sequence co-designed alongside the pocket structure. In addition to visualization, we report TM-score, RMSD, and AAR, where EnzymeFlow outperforms RFDiffusionAA across all three metrics, demonstrating EnzymeFlow's ability to generate more structurally valid enzyme catalytic pockets.

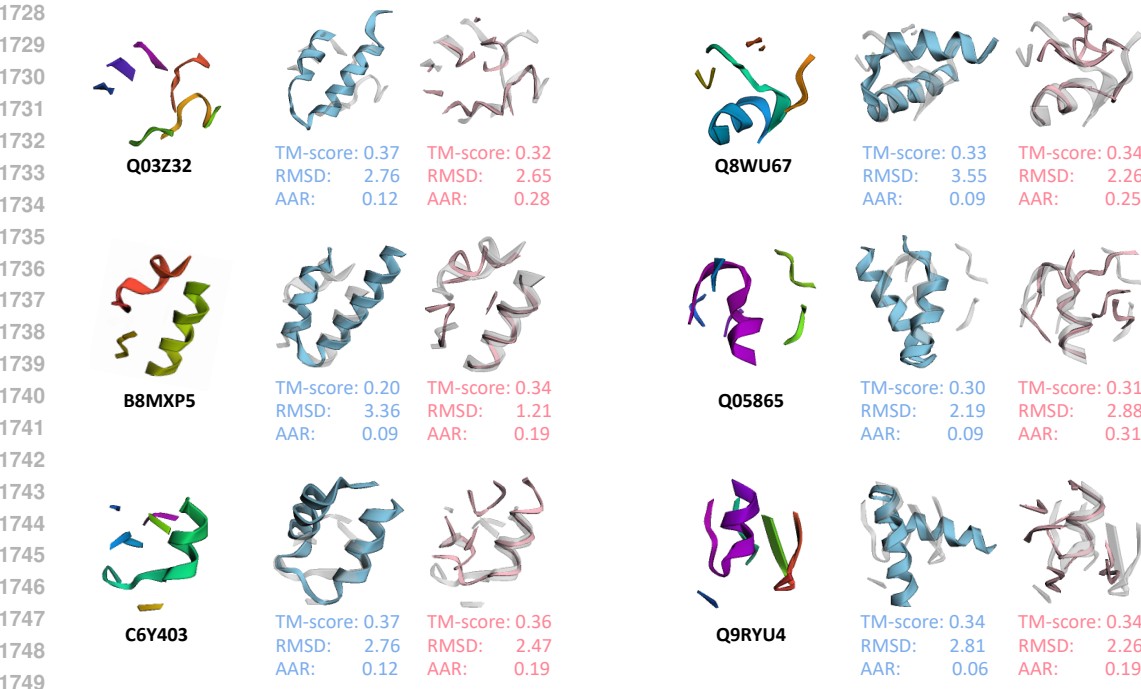

Figure 14: Visualization and comparison between RFDiffusionAA-designed pockets and EnzymeFlow-designed pockets after superimposition with ground-truth pockets. Light color represents EnzymeFlow-designed pockets, blue color represents RFDiffusionAA-designed pockets, spectral color represents the ground-truth reference pockets. TM-score, RMSD, AAR are reported.

## K  ENZYMEFLOW NEURAL NETWORK IMPLEMENTATION

The equivariant neural network is based on the Invariant Point Attention (IPA) implemented in AlphaFold2 (Jumper et al., 2021). In the following, we detail how enzyme catalytic pockets, substrate molecules, product molecules, EC-class, and co-evolution interact within our network.

The code for EnzymeFlow main network follows directly:

```python
import functools as fn
import math

import torch
import torch.nn as nn
from torch.nn import functional as F

from ofold.utils.rigid_utils import Rigid

from model import ipa_pytorch
from flowmatch.data import all_atom
from flowmatch.data import utils as du

## EnzymeFlow Main Network

## (8) Distogram
def calc_distogram(pos, min_bin, max_bin, num_bins):
    dists_2d = torch.linalg.norm(pos[:, :, None, :] - pos[:, None, :, :], axis=-1)[
        ..., None
    ]
    lower = torch.linspace(min_bin, max_bin, num_bins, device=pos.device)
    upper = torch.cat([lower[1:], lower.new_tensor([1e8])], dim=-1)
    dgram = ((dists_2d > lower) * (dists_2d < upper)).type(pos.dtype)
    return dgram

## (7) Index Embedding
def get_index_embedding(indices, embed_size, max_len=2056):
    K = torch.arange(embed_size // 2, device=indices.device)
    pos_embedding_sin = torch.sin(
        indices[..., None] * math.pi / (max_len ** (2 * K[None] / embed_size))
```

```python
    ).to(indices.device)
    pos_embedding_cos = torch.cos(
        indices[..., None] * math.pi / (max_len ** (2 * K[None] / embed_size))
    ).to(indices.device)
    pos_embedding = torch.cat([pos_embedding_sin, pos_embedding_cos], axis=-1)
    return pos_embedding

## (6) Time Embedding
def get_timestep_embedding(timesteps, embedding_dim, max_positions=10000):
    assert len(timesteps.shape) == 1
    timesteps = timesteps * max_positions
    half_dim = embedding_dim // 2
    emb = math.log(max_positions) / (half_dim - 1)
    emb = torch.exp(
        torch.arange(half_dim, dtype=torch.float32, device=timesteps.device) * -emb
    )
    emb = timesteps.float()[:, None] * emb[None, :]
    emb = torch.cat([torch.sin(emb), torch.cos(emb)], dim=1)
    if embedding_dim % 2 == 1:  # zero pad
        emb = F.pad(emb, (0, 1), mode="constant")
    assert emb.shape == (timesteps.shape[0], embedding_dim)
    return emb

## (5) Edge Feature Network
class EdgeFeatureNet(nn.Module):
    def __init__(self, module_cfg):
        super(EdgeFeatureNet, self).__init__()
        self._cfg = module_cfg

        self.c_s = self._cfg.embed.c_s
        self.c_z = self._cfg.embed.c_z
        self.feat_dim = self._cfg.embed.feat_dim

        self.linear_s_p = nn.Linear(self.c_s, self.feat_dim)
        self.linear_relpos = nn.Linear(self.feat_dim, self.feat_dim)

        total_edge_feats = self.feat_dim * 3 + self._cfg.embed.num_bins * 2 + 2

        self.edge_embedder = nn.Sequential(
            nn.Linear(total_edge_feats, self.c_z),
            nn.ReLU(),
            nn.Linear(self.c_z, self.c_z),
            nn.ReLU(),
            nn.Linear(self.c_z, self.c_z),
            nn.LayerNorm(self.c_z),
        )

    def embed_relpos(self, r):
        d = r[:, :, None] - r[:, None, :]
        pos_emb = get_index_embedding(d, self.feat_dim, max_len=2056)
        return self.linear_relpos(pos_emb)

    def _cross_concat(self, feats_1d, num_batch, num_res):
        return torch.cat([
            torch.tile(feats_1d[:, :, None, :], (1, 1, num_res, 1)),
            torch.tile(feats_1d[:, None, :, :], (1, num_res, 1, 1)),
        ], dim=-1).float().reshape([num_batch, num_res, num_res, -1])

    def forward(self, s, t, sc_t, edge_mask, flow_mask):
        # Input: [b, n_res, c_s]
        num_batch, num_res, _ = s.shape

        # [b, n_res, c_z]
        p_i = self.linear_s_p(s)
        cross_node_feats = self._cross_concat(p_i, num_batch, num_res)

        # [b, n_res]
        r = torch.arange(
            num_res, device=s.device).unsqueeze(0).repeat(num_batch, 1)
        relpos_feats = self.embed_relpos(r)

        dist_feats = calc_distogram(
            t, min_bin=1e-3, max_bin=20.0, num_bins=self._cfg.embed.num_bins)
        sc_feats = calc_distogram(
            sc_t, min_bin=1e-3, max_bin=20.0, num_bins=self._cfg.embed.num_bins)

        all_edge_feats = [cross_node_feats, relpos_feats, dist_feats, sc_feats]

        diff_feat = self._cross_concat(flow_mask[..., None], num_batch, num_res)
```

```python
113        all_edge_feats.append(diff_feat)
114
115        edge_feats = self.edge_embedder(torch.concat(all_edge_feats, dim=-1).to(torch.float))
116        edge_feats *= edge_mask.unsqueeze(-1)
117        return edge_feats
118
119
120 ## (4) Node Feature Network
121 class NodeFeatureNet(nn.Module):
122     def __init__(self, module_cfg):
123         super(NodeFeatureNet, self).__init__()
124         self._cfg = module_cfg
125         self.c_s = self._cfg.embed.c_s
126         self.c_pos_emb = self._cfg.embed.c_pos_emb
127         self.c_timestep_emb = self._cfg.embed.c_timestep_emb
128         embed_size = self.c_pos_emb + self.c_timestep_emb * 2 + 1
129
130         self.aatype_embedding = nn.Embedding(21, self.c_s) # Always 21 because of 20 amino
    acids + 1 for unk
131         embed_size += self.c_s + self.c_timestep_emb + self._cfg.num_aa_type
132
133         self.linear = nn.Sequential(
134             nn.Linear(embed_size, self.c_s),
135             nn.ReLU(),
136             nn.Linear(self.c_s, self.c_s),
137             nn.ReLU(),
138             nn.Linear(self.c_s, self.c_s),
139             nn.LayerNorm(self.c_s),
140         )
141
142     def embed_t(self, timesteps, mask):
143         timestep_emb = get_timestep_embedding(
144             timesteps,
145             self.c_timestep_emb,
146             max_positions=2056
147         )[:, None, :].repeat(1, mask.shape[1], 1)
148         return timestep_emb * mask.unsqueeze(-1)
149
150     def forward(
151         self,
152         *,
153         t,
154         res_mask,
155         flow_mask,
156         pos,
157         aatypes,
158         aatypes_sc,
159     ):
160         # [b, n_res, c_pos_emb]
161         pos_emb = get_index_embedding(pos, self.c_pos_emb, max_len=2056)
162         pos_emb = pos_emb * res_mask.unsqueeze(-1)
163
164         # [b, n_res, c_timestep_emb]
165         input_feats = [
166             pos_emb,
167             flow_mask[..., None],
168             self.embed_t(t, res_mask),
169             self.embed_t(t, res_mask)
170         ]
171         input_feats.append(self.aatype_embedding(aatypes))
172         input_feats.append(self.embed_t(t, res_mask))
173         input_feats.append(aatypes_sc)
174         return self.linear(torch.cat(input_feats, dim=-1))
175
176
177 ## (3) Distance Embedder
178 class DistEmbedder(nn.Module):
179     def __init__(self, model_conf):
180         super(DistEmbedder, self).__init__()
181         torch.set_default_dtype(torch.float32)
182         self._model_conf = model_conf
183         self._embed_conf = model_conf.embed
184
185         edge_embed_size = self._model_conf.edge_embed_size
186
187         self.dist_min = self._model_conf.bb_ligand_rbf_d_min
188         self.dist_max = self._model_conf.bb_ligand_rbf_d_max
189         self.num_rbf_size = self._model_conf.num_rbf_size
190         self.edge_embedder = nn.Sequential(
191             nn.Linear(self.num_rbf_size, edge_embed_size),
192             nn.ReLU(),
```

```
193            nn.Linear(edge_embed_size, edge_embed_size),
194            nn.ReLU(),
195            nn.Linear(edge_embed_size, edge_embed_size),
196            nn.LayerNorm(edge_embed_size),
197        )
198
199        mu = torch.linspace(self.dist_min, self.dist_max, self.num_rbf_size)
200        self.mu = mu.reshape([1, 1, 1, -1])
201        self.sigma = (self.dist_max - self.dist_min) / self.num_rbf_size
202
203    def coord2dist(self, coord, edge_mask):
204        n_batch, n_atom = coord.size(0), coord.size(1)
205        radial = torch.sum((coord.unsqueeze(1) - coord.unsqueeze(2)) ** 2, dim=-1)
206        dist = torch.sqrt(
207                radial + 1e-10
208            ) * edge_mask
209
210        radial = radial * edge_mask
211        return radial, dist
212
213    def rbf(self, dist):
214        dist_expand = torch.unsqueeze(dist, -1)
215        _mu = self.mu.to(dist.device)
216        rbf = torch.exp(-(((dist_expand - _mu) / self.sigma) ** 2))
217        return rbf
218
219    def forward(
220        self,
221        rigid,
222        ligand_pos,
223        bb_ligand_mask,
224    ):
225        curr_bb_pos = all_atom.to_atom37(Rigid.from_tensor_7(torch.clone(rigid)))[-1][:, :,
     1].to(ligand_pos.device)
226
227        curr_bb_lig_pos = torch.cat([curr_bb_pos, ligand_pos], dim=1)
228        edge_mask = bb_ligand_mask.unsqueeze(dim=1) * bb_ligand_mask.unsqueeze(dim=2)
229
230        radial, dist = self.coord2dist(
231                    coord=curr_bb_lig_pos,
232                    edge_mask=edge_mask,
233                )
234
235
236        edge_embed = self.rbf(dist) * edge_mask[..., None]
237        edge_embed = self.edge_embedder(edge_embed.to(torch.float))
238
239        return edge_embed
240
241
242 ## (2) Cross-Attentiom
243 class CrossAttention(nn.Module):
244    def __init__(self, query_input_dim, key_input_dim, output_dim):
245        super(CrossAttention, self).__init__()
246        self.out_dim = output_dim
247        self.W_Q = nn.Linear(query_input_dim, output_dim)
248        self.W_K = nn.Linear(key_input_dim, output_dim)
249        self.W_V = nn.Linear(key_input_dim, output_dim)
250        self.scale_val = self.out_dim ** 0.5
251        self.softmax = nn.Softmax(dim=-1)
252
253    def forward(self, query_input, key_input, value_input, query_input_mask=None,
     key_input_mask=None):
254        query = self.W_Q(query_input)
255        key = self.W_K(key_input)
256        value = self.W_V(value_input)
257
258        attn_weights = torch.matmul(query, key.transpose(1, 2)) / self.scale_val
259        attn_mask = query_input_mask.unsqueeze(-1) * key_input_mask.unsqueeze(-1).transpose(1,
     2)
260        attn_weights = attn_weights.masked_fill(attn_mask == False, -1e9)
261        attn_weights = self.softmax(attn_weights)
262        output = torch.matmul(attn_weights, value)
263
264        return output, attn_weights
265
266
267 ## (1) Protein-Ligand Network
268 class ProteinLigandNetwork(nn.Module):
269    def __init__(self, model_conf):
270        super(ProteinLigandNetwork, self).__init__()
```

```
271            torch.set_default_dtype(torch.float32)
272            self._model_conf = model_conf
273
274            # Input Node Embedder
275            self.node_feature_net = NodeFeatureNet(model_conf)
276
277            # Input Edge Embedder
278            self.edge_feature_net = EdgeFeatureNet(model_conf)
279
280            # 3D Molecule GNN
281            self.mol_embedding_layer = MolEmbedder(model_conf)
282
283            # Invariant Point Attention (IPA) Network
284            self.ipanet = ipa_pytorch.IpaNetwork(model_conf)
285
286            # Node Fusion
287            self.node_embed_size = self._model_conf.node_embed_size
288            self.node_embedder = nn.Sequential(
289                nn.Embedding(self._model_conf.num_aa_type, self.node_embed_size),
290                nn.ReLU(),
291                nn.Linear(self.node_embed_size, self.node_embed_size),
292                nn.LayerNorm(self.node_embed_size),
293            )
294            self.node_fusion = nn.Sequential(
295                nn.Linear(self.node_embed_size + self.node_embed_size, self.node_embed_size),
296                nn.ReLU(),
297                nn.Linear(self.node_embed_size, self.node_embed_size),
298                nn.LayerNorm(self.node_embed_size),
299            )
300
301            # Backbone-Substrate Fusion
302            self.bb_lig_fusion = CrossAttention(
303                    query_input_dim=self.node_embed_size,
304                    key_input_dim=self.node_embed_size,
305                    output_dim=self.node_embed_size,
306            )
307
308            # Edge Fusion
309            self.edge_embed_size = self._model_conf.edge_embed_size
310            self.edge_dist_embedder = DistEmbedder(model_conf)
311
312            # Amino Acid Prediction Network
313            self.aatype_pred_net = nn.Sequential(
314                    nn.Linear(self.node_embed_size, self.node_embed_size),
315                    nn.ReLU(),
316                    nn.Linear(self.node_embed_size, self.node_embed_size),
317                    nn.ReLU(),
318                    nn.Linear(self.node_embed_size, model_conf.num_aa_type),
319            )
320
321            if self._model_conf.flow_msa:
322                # Co-Evolution Embedder
323                self.msa_embedding_layer = CoEvoFormer(model_conf)
324
325                # Coevo-Backbone-Substrate Fusion
326                self.msa_bb_lig_fusion = CrossAttention(
327                    query_input_dim=model_conf.msa.msa_embed_size,
328                    key_input_dim=self.node_embed_size,
329                    output_dim=self.node_embed_size,
330                )
331
332                # Coevo Prediction Network
333                self.msa_pred = nn.Sequential(
334                    nn.Linear(self.node_embed_size, self.node_embed_size),
335                    nn.SiLU(),
336                    nn.Linear(self.node_embed_size, self.node_embed_size),
337                    nn.SiLU(),
338                    nn.Linear(self.node_embed_size, model_conf.msa.num_msa_vocab),
339                )
340
341            if self._model_conf.ec:
342                # EC Embedder
343                self.ec_embedding_layer = nn.Sequential(
344                    nn.Embedding(model_conf.ec.num_ec_class, model_conf.ec.ec_embed_size),
345                    nn.SiLU(),
346                    nn.Linear(model_conf.ec.ec_embed_size, model_conf.ec.ec_embed_size),
347                    nn.LayerNorm(model_conf.ec.ec_embed_size),
348                )
349
350                # EC-Backbone-Substrate Fusion
351                self.ec_bb_lig_fusion = CrossAttention(
```

```
352              query_input_dim=model_conf.ec.ec_embed_size,
353              key_input_dim=self.node_embed_size,
354              output_dim=self.node_embed_size,
355          )
356
357          # EC Prediction Network
358          self.ec_pred = nn.Sequential(
359              nn.Linear(self.node_embed_size, self.node_embed_size),
360              nn.SiLU(),
361              nn.Linear(self.node_embed_size, self.node_embed_size),
362              nn.SiLU(),
363              nn.Linear(self.node_embed_size, model_conf.ec.num_ec_class),
364          )
365
366      self.condition_generation = self._model_conf.guide_by_condition
367      if self.condition_generation:
368          # 2D Molecule GNN
369          self.guide_ligand_mpnn = MolEmbedder2D(model_conf)
370
371          # Backbone-Product Fusion
372          self.guide_bb_lig_fusion = CrossAttention(
373              query_input_dim=self.node_embed_size,
374              key_input_dim=self.node_embed_size,
375              output_dim=self.node_embed_size,
376          )
377
378  def forward(self, input_feats, use_context=False):
379      # Frames as [batch, res, 7] tensors.
380      bb_mask = input_feats["res_mask"].type(torch.float32)  # [B, N]
381      flow_mask = input_feats["flow_mask"].type(torch.float32)
382      edge_mask = bb_mask[..., None] * bb_mask[..., None, :]
383
384      n_batch, n_res = bb_mask.shape
385
386      # Encode Backbone Nodes with Input Node Embedder
387      init_bb_node_embed = self.node_feature_net(
388          t=input_feats["t"],
389          res_mask=bb_mask,
390          flow_mask=flow_mask,
391          pos=input_feats["seq_idx"],
392          aatypes=input_feats["aatype_t"],
393          aatypes_sc=input_feats["sc_aa_t"],
394      )
395
396      # Encode Backbone Edges with Input Edge Embedder
397      init_bb_edge_embed = self.edge_feature_net(
398          s=init_bb_node_embed,
399          t=input_feats["trans_t"],
400          sc_t=input_feats["sc_ca_t"],
401          edge_mask=edge_mask,
402          flow_mask=flow_mask,
403      )
404
405      # Masking Padded Residues
406      bb_node_embed = init_bb_node_embed * bb_mask[..., None]
407      bb_edge_embed = init_bb_edge_embed * edge_mask[..., None]
408
409      # AminoAcid embedding
410      bb_aa_embed = self.node_embedder(input_feats["aatype_t"]) * bb_mask[..., None]
411      bb_aa_embed = torch.cat([bb_aa_embed, bb_node_embed], dim=-1)
412      # Backbone-AminoAcid Fusion
413      bb_node_embed = self.node_fusion(bb_aa_embed)
414      bb_node_embed = bb_node_embed * bb_mask[..., None]
415
416      # Initialze Substrate Masking
417      lig_mask = input_feats["ligand_mask"]
418      lig_edge_mask = lig_mask[..., None] * lig_mask[..., None, :]
419      # Encode Substrate with 3D Molecule GNN
420      lig_init_node_embed, _ = self.mol_embedding_layer(
421              ligand_atom=input_feats["ligand_atom"],
422              ligand_pos=input_feats["ligand_pos"],
423              edge_mask=lig_edge_mask,
424          )
425      lig_node_embed = lig_init_node_embed * lig_mask[..., None]
426
427      # Backbone-Substrate Fusion
428      bb_lig_rep, _ = self.bb_lig_fusion(
429                          query_input=bb_node_embed,
430                          key_input=lig_node_embed,
431                          value_input=lig_node_embed,
432                          query_input_mask=bb_mask,
```

```
2052
2053  433                                        key_input_mask=lig_mask,
      434                            )
2054  435
2055  436          # Residue Connection
      437          bb_node_embed = bb_node_embed + bb_lig_rep
2056  438
2057  439          # Conditioning on Product Molecule
      440          if self.condition_generation:
2058  441              # Encode Product with 2D Molecule GNN
      442              guide_ligand_rep = self.guide_ligand_mpnn(
2059  443                              mol_atom=input_feats["guide_ligand_atom"],
2060  444                              mol_edge=input_feats["guide_ligand_edge_index"],
      445                              mol_edge_feat=input_feats["guide_ligand_edge"],
2061  446                              mol_atom_mask=input_feats["guide_ligand_atom_mask"],
2062  447                              mol_edge_mask=input_feats["guide_ligand_edge_mask"],
      448                          ).unsqueeze(1)
2063  449
2064  450              # Initialze Product Masking
      451              guide_ligand_mask = input_feats["guide_ligand_atom_mask"][:, 0:1]
2065  452              # Backbone-Product Fusion
2066  453              bb_guide_lig_rep, _ = self.guide_bb_lig_fusion(
      454                              query_input=bb_node_embed,
2067  455                              key_input=guide_ligand_rep,
2068  456                              value_input=guide_ligand_rep,
      457                              query_input_mask=bb_mask,
2069  458                              key_input_mask=guide_ligand_mask,
      459                          )
2070  460
2071  461              # Residue Connection
2072  462              bb_node_embed = bb_node_embed + bb_guide_lig_rep
      463
2073  464          # Initialze Backbone-Substrate Masking
2074  465          bb_ligand_mask = torch.cat([bb_mask, lig_mask], dim=-1)
      466          # Backbone-Substrate Distance Embedding
2075  467          bb_lig_edge = self.edge_dist_embedder(
2076  468              rigid=input_feats["rigids_t"],
      469              ligand_pos=input_feats["ligand_pos"],
2077  470              bb_ligand_mask=bb_ligand_mask,
      471          )
2078  472
2079  473          # Backbone-Backbone-Product Edge Fusion
2080  474          bb_edge_embed = bb_edge_embed + bb_lig_edge[:, :n_res, :n_res, :]
      475
2081  476          # Masking Padded Residues
2082  477          bb_node_embed = bb_node_embed[:, :n_res, :] * bb_mask[..., None]
      478          bb_edge_embed = bb_edge_embed[:, :n_res, :n_res, :] * edge_mask[..., None]
2083  479
2084  480          # Run IPA Network
      481          model_out = self.ipanet(bb_node_embed, bb_edge_embed, input_feats)
2085  482          node_embed = model_out["node_embed"] * bb_mask[..., None]
2086  483
2087  484          # Amino Acid Prediction with Amino Acid Prediction Network
      485          aa_pred = self.aatype_pred_net(node_embed) * bb_mask[..., None]
2088  486
2089  487          if self._model_conf.flow_msa:
      488              # Encode Coevo with Co-Evolution Embedder
2090  489              msa_mask = input_feats["msa_mask"]
      490              msa_embed = self.msa_embedding_layer(input_feats["msa_t"], msa_mask=msa_mask) *
2091          msa_mask[..., None] #[B, N_msa, N_token, D]
2092  491              msa_rep = msa_embed.sum(dim=1) / (msa_mask[..., None].sum(dim=1) + 1e-10) #[B, 1,
2093          D]
      492              _msa_mask = msa_mask[:, 0] #torch.ones_like(msa_rep[..., 0]).to(msa_embed.device)
2094  493
2095  494              # Coevo-Backbone Fusion
      495              msa_rep, _ = self.msa_bb_lig_fusion(
2096  496                              query_input=msa_rep,
2097  497                              key_input=node_embed,
      498                              value_input=node_embed,
2098  499                              query_input_mask=_msa_mask,
      500                              key_input_mask=bb_mask,
2099  501                          )
2100  502
2101  503              # Coevo Prediction with Coevo Prediction Network
      504              msa_pred = self.msa_pred(msa_rep)
2102  505
2103  506          if self._model_conf.flow_ec:
      507              # Encode EC with EC Embedder
2104  508              ec_embed = self.ec_embedding_layer(input_feats["ec_t"])
      509              ec_mask = torch.ones_like(ec_embed[..., 0]).to(ec_embed.device)
2105  510
      511              # EC-Backbone Fusion
```

```
            ec_rep, _ = self.ec_bb_lig_fusion(
                                query_input=ec_embed,
                                key_input=node_embed,
                                value_input=node_embed,
                                query_input_mask=ec_mask,
                                key_input_mask=bb_mask,
                            )

        # EC Prediction with EC Prediction Network
        ec_rep = ec_rep.reshape(n_batch, -1)
        ec_pred = self.ec_pred(ec_rep)

    # Main Network Ouput
    pred_out = {
        "amino_acid": aa_pred,
        "rigids_tensor": model_out["rigids"],
    }

    if self._model_conf.flow_msa:
        pred_out["msa"] = msa_pred * _msa_mask[..., None]

    if self._model_conf.flow_ec:
        pred_out["ec"] = ec_pred

    pred_out["rigids"] = model_out["rigids"].to_tensor_7()
    return pred_out
```

Listing 4: Pytorch Implementation of EnzymeFlow Main Network.

*Fun Fact:* While implementing enzyme-substrate and enzyme-product interactions by cross-attention fusion networks, we experimented with using PairFormer (with only 3-4 layers) as implemented in AlphaFold3 (Abramson et al., 2024). However, the computational load was immense—it would take years to run on our A40 GPU. Our fusion network turns to be a more efficient approach. It makes me wonder who has the resources to re-train AlphaFold3, given the heavy computational demands!

