# OpenReview forum: "EnzymeFlow: Generating Reaction-specific Enzyme Catalytic Pockets through Flow Matching and Co-Evolutionary Dynamics"
_ICLR.cc/2025/Conference — Submitted to ICLR 2025_

### Official Review · Reviewer_ZZtw · 2024-10-31

**Soundness:** 2
**Presentation:** 2
**Contribution:** 2
**Rating:** 3
**Confidence:** 5

**Summary:**

This paper proposes a new model named EnzymeFlow, designed to generate enzyme catalytic pockets for specific substrates through flow matching and enzyme-reaction co-evolution. The model employs a hierarchical pre-training strategy, initially pre-trained on protein backbones and ligand-binding pockets, followed by fine-tuning on enzyme catalytic reactions. The authors also constructed a large-scale dataset named EnzymeFill, containing over 320,000 enzyme-reaction pairs, to support model training. Experimental results indicate that EnzymeFlow performs well in generating high-quality catalytic pockets, showing its potential in enzyme engineering and synthetic biology.

**Strengths:**

This paper provides a dataset for enzyme pocket design tasks and details the workflow for collecting and cleaning the dataset. It adapts and innovates upon the previous ideas from PocketFlow for the enzyme pocket design context by incorporating 3D and 2D molecular information of substrates and product molecules at the encoding level, thereby describing the dynamic catalytic process. The paper also introduces a new approach to constructing and encoding MSA co-evolutionary information for enzyme catalysis, offering new ideas for future developments in this field.

**Weaknesses:**

The design of certain modules and the loss functions raises questions about their logical significance. Specifically, Equation (5) describes the score function loss for enzyme-reaction co-evolution information. Reconstructing MSA from its noisy encoded form would certainly make sense, but in the code implementation, the MSA encoding is only activated during MSA diffusion. Ultimately, it compresses $N_{MSA}$ into a single sequence representation (MSA + bb_lig_fusion_emb) to reconstruct $N_{MSA}$, raising doubts about the validity of reconstructing high-dimensional data from such a low-dimensional form.

Moreover, both MSA and EC class, as implemented in the code, function more like two output heads in a multi-task setup. These significant conditional features are not integrated early into the IPA network but are instead fused after the IPA outputs the updated structure, making it difficult for these features to directly influence IPA’s backbone design. This makes the description in Equation (6) misleading, as it may lead readers to believe that these conditional features have a substantial multimodal influence on backbone design. If the authors had included an architecture of the model's computation flow, it would be clear that the feature paths of MSA and EC class are almost entirely separate from the core backbone + substrate + product molecule paths. In Figure 3.3, the influence of EC class and co-evolution on the gradient is minimal, more akin to decoding pre-designed structure embeddings to predict MSA and EC class.

Additionally, a long-standing issue in this field is that enzyme pocket design is distinct from general protein design, and in silico metrics alone are insufficient to establish the practicality and reliability of newly designed enzymes. Only wet lab experimental results serve as the gold standard for this task.

**Questions:**

1. According to the code, it appears that providing MSA and EC class during sampling does not influence IPA's backbone design, nor are these features fed back as self-conditioning. Do these tasks merely help the model learn a better landscape during training?
2. Why aren't the features of MSA and EC class integrated into the backbone embedding before entering IPA?
3. Continuing from question 1, how is the multi-label classification for EC class specifically conducted? EnzymeFlow uses ec_pred for class prediction, while the comparison method uses the CLEAN model to predict EC class from the generated structure. Is this a fair comparison?
4. Continuing from question 3, the ec_embed feature of EC class does not enter IPA, so how is it ensured that the generated structure matches the given EC class during sampling?
5. How were the comparative methods set up? Does Table 2 include the performance of PocketFlow fine-tuned on enzyme data?

---

> ### Author Response · Authors · 2024-11-13
> **Answers to Questions Part1**
>
> #### Thanks for the feedback. We have been conducting more (a lot) experiments in the past month to get ready for this rebuttal, and aim to address the weaknesses and questions, to demonstrate model’s robustness in designing catalytic pockets.
>
> #### We also agree that in silico evaluations in enzyme pocket design are not enough. We have some collaborations that are working on wet-lab experiment for this task. Our focus on catalytic pocket design rather than complete enzyme design stems from both the scientific challenges and strategic advantages of this approach. Designing a full enzyme from scratch is an extremely complex task. Focusing on catalytic pockets first allows us to solve a foundational aspect of enzyme design by establishing effective reaction sites, which are essential for catalysis. And EnzymeFlow is our first model, we have inpainting, screening, and other related models being developed, aiming at working to enzyme design in silico at some point. And with better screening and filtering, we can send the designed pockets and enzymes to wet-lab. We aim to contribute to advancing well-being through algorithms.
>
>
> ## A1:
>
> #### We trained an ablation model that incorporates only reaction conditions, but excludes additional elements such as co-MSAs and EC conditions. This model, which we refer to as EnzymeFlow-ablation, was trained the same as we did for EnzymeFlow. The performance of the EnzymeFlow-ablation model is compared to the full EnzymeFlow model, which incorporates co-MSAs, EC conditions, and reaction conditioning. The results show that the EnzymeFlow model outperforms the EnzymeFlow-ablation model, as evidenced by the following metrics: lower RMSD, higher alignment scores, lower embedding distance (we use ESM3 encoding for pocket sequences and structures)
> ```
> |                   |                    | rmsd | rmsd | rmsd | tm   | tm   | tm   | Dist  | Dist  | Dist  |
> | ----------------- | ------------------ | ---- | ---- | ---- | ---- | ---- | ---- | ----- | ----- | ----- |
> | Pocket Evaluation |                    | Top1 | Top3 | Mean | Top1 | Top3 | Mean | Top1  | Top3  | Mean  |
> | ----------------- | ------------------ | ---- | ---- | ---- | ---- | ---- | ---- | ----- | ----- | ----- |
> | EC1               | EnzymeFlow-abation | 3.76 | 3.90 | 4.19 | 0.29 | 0.28 | 0.26 | 19.82 | 20.29 | 21.48 |
> |                   | RFDiffusionAA      | 3.71 | 3.82 | 4.18 | 0.28 | 0.27 | 0.25 | 7.33  | 8.05  | 9.28  |
> |                   | EnzymeFlow         | 1.88 | 2.09 | 2.41 | 0.67 | 0.65 | 0.59 | 0.25  | 0.58  | 1.54  |
> | ----------------- | ------------------ | ---- | ---- | ---- | ---- | ---- | ---- | ----- | ----- | ----- |
> | EC2               | EnzymeFlow-abation | 4.02 | 4.19 | 4.35 | 0.23 | 0.23 | 0.21 | 50.31 | 52.33 | 54.15 |
> |                   | RFDiffusionAA      | 3.47 | 3.78 | 4.17 | 0.27 | 0.27 | 0.25 | 26.91 | 27.84 | 30.44 |
> |                   | EnzymeFlow         | 3.65 | 3.80 | 4.09 | 0.33 | 0.31 | 0.28 | 1.03  | 1.85  | 4.40  |
> | ----------------- | ------------------ | ---- | ---- | ---- | ---- | ---- | ---- | ----- | ----- | ----- |
> | EC3               | EnzymeFlow-abation | 3.46 | 3.69 | 4.12 | 0.31 | 0.29 | 0.27 | 9.84  | 10.02 | 11.14 |
> |                   | RFDiffusionAA      | 3.25 | 3.50 | 3.96 | 0.31 | 0.30 | 0.27 | 10.83 | 10.96 | 12.22 |
> |                   | EnzymeFlow         | 3.18 | 3.32 | 3.84 | 0.39 | 0.36 | 0.32 | 0.42  | 1.08  | 2.19  |
> | ----------------- | ------------------ | ---- | ---- | ---- | ---- | ---- | ---- | ----- | ----- | ----- |
> | EC4               | EnzymeFlow-abation | 3.31 | 3.48 | 3.91 | 0.32 | 0.30 | 0.27 | 18.02 | 18.56 | 19.96 |
> |                   | RFDiffusionAA      | 3.00 | 3.29 | 3.79 | 0.30 | 0.29 | 0.27 | 14.62 | 15.62 | 17.39 |
> |                   | EnzymeFlow         | 2.58 | 2.73 | 2.96 | 0.56 | 0.54 | 0.51 | 0.50  | 1.07  | 2.43  |
> | ----------------- | ------------------ | ---- | ---- | ---- | ---- | ---- | ---- | ----- | ----- | ----- |
> | EC5               | EnzymeFlow-abation | 3.48 | 3.66 | 4.06 | 0.29 | 0.28 | 0.26 | 13.71 | 14.01 | 14.81 |
> |                   | RFDiffusionAA      | 3.39 | 3.61 | 3.98 | 0.31 | 0.29 | 0.26 | 7.83  | 8.23  | 8.93  |
> |                   | EnzymeFlow         | 2.87 | 3.18 | 3.62 | 0.44 | 0.43 | 0.39 | 0.74  | 1.07  | 1.78  |
> | ----------------- | ------------------ | ---- | ---- | ---- | ---- | ---- | ---- | ----- | ----- | ----- |
> | EC6               | EnzymeFlow-abation | 3.78 | 3.89 | 4.33 | 0.29 | 0.27 | 0.26 | 21.00 | 22.87 | 24.19 |
> |                   | RFDiffusionAA      | 3.43 | 3.69 | 4.12 | 0.30 | 0.29 | 0.26 | 19.08 | 19.73 | 21.21 |
> |                   | EnzymeFlow         | 3.14 | 3.32 | 3.73 | 0.43 | 0.40 | 0.36 | 1.23  | 1.45  | 2.56  |
> ```

---

> ### Author Response · Authors · 2024-11-13
> **Answers to Questions Part2**
>
> ## A1:
>
> #### To further demonstrate the functional validity of EnzymeFlow-generated pockets, we have included additional metrics such as optimal pH, turnover number, and substrate specificity. These metrics indicate that EnzymeFlow-generated pockets align closely with baseline-generated pockets, further supporting the model's capacity to generate functionally relevant structures.
> ```
> |                   |                    | pH   | kcat | km   |
> | ----------------- | ------------------ | ---- | ---- | ---- |
> | Pocket Evaluation |                    | mean | mean | mean |
> | ----------------- | ------------------ | ---- | ---- | ---- |
> | EC1               | ground truth       | 7.86 | 2.86 | 0.33 |
> |                   | EnzymeFlow-abation | 8.40 | 1.08 | 0.28 |
> |                   | RFDiffusionAA      | 8.85 | 2.17 | 0.24 |
> |                   | EnzymeFlow         | 7.54 | 2.45 | 0.37 |
> | ----------------- | ------------------ | ---- | ---- | ---- |
> | EC2               | ground truth       | 7.68 | 1.93 | 0.31 |
> |                   | EnzymeFlow-abation | 8.48 | 0.72 | 0.20 |
> |                   | RFDiffusionAA      | 8.89 | 1.55 | 0.19 |
> |                   | EnzymeFlow         | 7.58 | 2.03 | 0.27 |
> | ----------------- | ------------------ | ---- | ---- | ---- |
> | EC3               | ground truth       | 7.49 | 2.18 | 0.09 |
> |                   | EnzymeFlow-abation | 8.56 | 0.79 | 0.08 |
> |                   | RFDiffusionAA      | 8.79 | 1.36 | 0.08 |
> |                   | EnzymeFlow         | 7.64 | 2.49 | 0.08 |
> | ----------------- | ------------------ | ---- | ---- | ---- |
> | EC4               | ground truth       | 7.82 | 1.57 | 0.09 |
> |                   | EnzymeFlow-abation | 8.48 | 0.78 | 0.10 |
> |                   | RFDiffusionAA      | 8.98 | 2.19 | 0.10 |
> |                   | EnzymeFlow         | 7.40 | 2.03 | 0.09 |
> | ----------------- | ------------------ | ---- | ---- | ---- |
> | EC5               | ground truth       | 7.10 | 2.71 | 0.14 |
> |                   | EnzymeFlow-abation | 8.32 | 0.95 | 0.12 |
> |                   | RFDiffusionAA      | 8.88 | 1.18 | 0.14 |
> |                   | EnzymeFlow         | 7.47 | 2.00 | 0.14 |
> | ----------------- | ------------------ | ---- | ---- | ---- |
> | EC6               | ground truth       | 7.63 | 2.03 | 0.47 |
> |                   | EnzymeFlow-abation | 8.44 | 0.84 | 0.27 |
> |                   | RFDiffusionAA      | 8.98 | 1.99 | 0.33 |
> |                   | EnzymeFlow         | 7.43 | 2.92 | 0.46 |
> ```
>
> ## A2:
>
> #### The decision not to integrate MSA and EC class features into the backbone embedding before entering IPA is based on prior experience with AlphaFold models, where MSA integration significantly slows inference. In AlphaFold, MSAs need to be pre-computed during sampling, resulting in slower inference times. To mitigate this, we opted to integrate MSA and EC class features *after* the IPA embeddings, using a cross-attention mechanism to align the MSA, EC features, and IPA embeddings. This approach allows MSAs and EC classes to guide gradient updates during training while avoiding the need to pre-compute MSAs at inference, thus improving efficiency. Our goal is to maintain effective guidance from MSA and EC class information without sacrificing inference speed.
>
> ## A3:
>
> #### In Section 5.2 and Figure 8, we explicitly mention ''EnzymeFlow with functions annotated post hoc by CLEAN'', to fairly compare with others annotated post hoc by CLEAN.
>
> ## A4:
>
> #### Although ec_embed does not directly enter IPA, it interacts with the IPA embeddings through a cross-attention layer applied after IPA. This cross-attention layer enables the model to conditionally update the IPA-derived embeddings based on the EC class information. By doing so, the EC class feature can guide structural adjustments in line with the target EC class.
>
> ## A5:
>
> #### For DEPACT and PocketGEN, we directly adopt their models provided in GitHub. For RFDiffusionAA, we also directly adopt their models provided in GitHub, setting the diffusion steps as 100 (as what they used in training), and predict the sequences post hoc using LigandMPNN with 0.30A Gaussian noise (they suggested we should predict the sequences using inverse folding model post hoc). In Section 5, we also explicitly mention ''We train PocketFlow on EnzymeFill without fixing the backbones.'', as the same setting as we did for training EnzymeFlow, to fairly compare the model.

---

> > ### Comment · Reviewer_ZZtw · 2024-11-14
> > **Concerns Regarding the Conditioning Mechanism of EC Class and MSA in the Sampling Process**
> >
> > According to the code calculation process provided on Anonymous GitHub, the authors failed to clearly explain the mechanism by which the EC class and MSA act as conditions. During the sampling process, the encoding of the EC class and MSA, as well as the application of cross-attention and prediction heads, occur after the IPA generates the scaffold, thus failing to influence the already generated scaffold. The authors may have overstated the role of the EC class as a conditioning factor; in fact, the EC class and MSA appear more like two multi-task learning tasks for augmentation.

---

> ### Author Response · Authors · 2024-11-14
> **We have no intention to overclaim, we present the ablation studies and formulation corresponding to the model design**
>
> #### In eq6, we stated that we conditionally sample coMSAs and ECs conditioned on the backbone, P(ec|backbone_new)P(MSA|backbone_new)P(backbone_new|backbone_old), it is been clearly stated in our paper. Also, we have make extensive ablations and new experiments, comparing EnzymeFlow to the ablation models-showing what happens if coMSAs and ECs are missed in the training. We have absolutely zero intention to overclaim things.

---

> > ### Author Response · Authors · 2024-11-14
> > **Apologize by authors**
> >
> > Dear reviewer ZZtw,
> >
> > We want to ensure that this rebuttal process remains constructive and beneficial. If any of our responses came across as offensive or frustrated, we sincerely apologize, as this was never our intention. We deeply appreciate your feedback, and your insights have been valuable in helping us improve our model. We are currently working on training a revised version, incorporating MSAs and ECs before entering the IPA, as you suggested, and we plan to make this updated model available for public use.
> >
> > We believe that advancing well-being through algorithmic innovation is a shared goal. Enzyme design holds a special place for us, as it has transformative potential for fields like metabolism and beyond. If you are conducting similar research, we truly hope that EnzymeFlow and the EnzymeFill dataset can be of use to you and help support your own work.
> >
> > Again, we apologize if any of our language hindered constructive discussion. Your feedback is genuinely valued, and we thank you for the opportunity to improve.
> >
> > Best,
> > Author

---

### Official Review · Reviewer_XPne · 2024-11-02

**Soundness:** 3
**Presentation:** 2
**Contribution:** 2
**Rating:** 6
**Confidence:** 3

**Summary:**

This paper proposes a model to design enzyme pockets conditioned on reactions. It is a flow-matching model to co-generate enzyme pocket structures, sequence, ec class, and MSAs of enzymes and reaction smiles. Their model is pre-trained on the protein structure dataset and protein-ligand binding dataset. Empirical study shows that their method outperforms other baselines in generation quality and function prediction.

**Strengths:**

(1) This work effectively integrates various resources - such as data, pre-training, and co-evolutionary information - to enhance enzyme design performance.

(2) The authors curate a dataset tailored for enzyme pocket design, which could serve as a crucial resource for advancing predictive models in this field.

**Weaknesses:**

(1) mismatch in introduction and method

The introduction emphasizes the importance of capturing chemical changes specific to enzyme generation, distinguishing it from docking tasks that do not consider reactions. However, the method section lacks sufficient detail on how reaction-specific modeling is incorporated. The approach appears closer to a docking setup, focusing primarily on backbone structures, with limited attention to sidechain interactions critical for catalytic activity.

(2) concerns about co-evolutionary dynamics

The sequence alignment approach applied to SMILES representations raises concerns, as small molecules are inherently non-sequential, and different SMILES can produce varying alignments.

(3) evaluation of functions

The rationale behind the ECAcc evaluation metric requires further clarification, as its relevance to enzyme function is not immediately clear.

(4) evaluation of structures

The curated enzyme pocket dataset is a valuable resource. However,  If I did not misunderstand the curation process,  the binding structures were computationally predicted, and there may be inherent biases from the prediction model. Including evaluations on non-synthetic datasets could help mitigate any potential bias and validate the robustness of the proposed methods.

**Questions:**

* About the ECAcc

First, the use of ECAcc as a metric for evaluating enzyme functionality requires additional justification. It is unclear how ECAcc is directly correlated with catalytic ability, and including references that support this metric’s relevance to enzymatic activity would help strengthen its validity as a functional evaluation.

Second, it would be insightful to understand the model’s performance when the EC class is not co-generated during training. If this information is not utilized by other baselines, the comparison may not be entirely fair.

Third, the reported accuracy (~0.2) appears relatively low, suggesting potential challenges in accurately predicting EC classes. An explanation of these challenges would be helpful.

---

> ### Author Response · Authors · 2024-11-13
> **Answers to Weaknesses Part1**
>
> #### Thanks for the feedback. We have been conducting more (a lot) experiments in the past month to get ready for this rebuttal, and aim to address the weaknesses and questions, to demonstrate model’s robustness in designing catalytic pockets:
>
>
> ## A1:
> #### Specifically, we model the changes  from substrates to products using co-MSAs and atom-to-atom cross-attention, which we believe can capture dynamic changes in the reaction process. Previous work, such as ReactZyme [2], has demonstrated that the atom-to-atom cross-attention mechanism can effectively model the changes between substrates and products. While this is not an explicit dynamic model like CLIPzyme [1], which uses transition graphs to model reaction dynamics, ReactZyme’s use of the cross-attention mechanism has shown superior performance in implicitly capturing these dynamics compared to explicit transition graph models. In addition, we evaluate the model's ability to implicitly learn dynamics by measuring substrate specificity (Km) and turnover number (Kcat), and our results indicate that the model produces substrate specificity values close to the ground truth.
> ```
> ## Enzyme-ablation denotes an ablation model without coMSAs and EC guidance in training.
> |                   |                    | kcat | km   |
> | ----------------- | ------------------ | ---- | ---- |
> | Pocket Evaluation |                    | mean | mean |
> | ----------------- | ------------------ | ---- | ---- |
> | EC1               | ground truth       | 2.86 | 0.33 |
> |                   | EnzymeFlow-abation | 1.08 | 0.28 |
> |                   | RFDiffusionAA      | 2.17 | 0.24 |
> |                   | EnzymeFlow         | 2.45 | 0.37 |
> | ----------------- | ------------------ | ---- | ---- |
> | EC2               | ground truth       | 1.93 | 0.31 |
> |                   | EnzymeFlow-abation | 0.72 | 0.20 |
> |                   | RFDiffusionAA      | 1.55 | 0.19 |
> |                   | EnzymeFlow         | 2.03 | 0.27 |
> | ----------------- | ------------------ | ---- | ---- |
> | EC3               | ground truth       | 2.18 | 0.09 |
> |                   | EnzymeFlow-abation | 0.79 | 0.08 |
> |                   | RFDiffusionAA      | 1.36 | 0.08 |
> |                   | EnzymeFlow         | 2.49 | 0.08 |
> | ----------------- | ------------------ | ---- | ---- |
> | EC4               | ground truth       | 1.57 | 0.09 |
> |                   | EnzymeFlow-abation | 0.78 | 0.10 |
> |                   | RFDiffusionAA      | 2.19 | 0.10 |
> |                   | EnzymeFlow         | 2.03 | 0.09 |
> | ----------------- | ------------------ | ---- | ---- |
> | EC5               | ground truth       | 2.71 | 0.14 |
> |                   | EnzymeFlow-abation | 0.95 | 0.12 |
> |                   | RFDiffusionAA      | 1.18 | 0.14 |
> |                   | EnzymeFlow         | 2.00 | 0.14 |
> | ----------------- | ------------------ | ---- | ---- |
> | EC6               | ground truth       | 2.03 | 0.47 |
> |                   | EnzymeFlow-abation | 0.84 | 0.27 |
> |                   | RFDiffusionAA      | 1.99 | 0.33 |
> |                   | EnzymeFlow         | 2.92 | 0.46 |
> ```
> #### This suggests that the model is effectively capturing not only static interactions but also the dynamic aspects of enzyme catalysis, even though these dynamics are not explicitly modeled. We acknowledge that the field of using AI to model reaction transitions and chemical dynamics is still emerging, and as part of our ongoing work, we are exploring additional methods to explicitly model these transitions. For example, we are investigating the identification of reaction centers and assigning atomic weights to regions of the substrate involved in the reaction, with atoms in the reaction center receiving higher attention during the transition. We believe that combining these approaches with attention mechanisms could allow our model to implicitly and more explicitly learn dynamic processes in enzymatic reactions. In summary, while we currently rely on implicit methods (such as atom-to-atom cross-attention) to model dynamic transitions, we are actively exploring ways to explicitly incorporate dynamic changes in the future. The current evidence from our results, particularly in terms of substrate specificity, suggests that the model can effectively capture important aspects of enzyme dynamics.
>
>
> ##### [1] Mikhael, P.G., Chinn, I. and Barzilay, R., 2024. CLIPZyme: Reaction-Conditioned Virtual Screening of Enzymes. arXiv preprint arXiv:2402.06748.
> ##### [2] Hua, C., Zhong, B., Luan, S., Hong, L., Wolf, G., Precup, D. and Zheng, S., 2024. Reactzyme: A benchmark for enzyme-reaction prediction. arXiv preprint arXiv:2408.13659.

---

> ### Author Response · Authors · 2024-11-13
> **Answers to Weaknesses Part2**
>
> ## A2:
>
> #### We agree that capturing co-evolutionary information between enzymes and reactions is a challenging yet promising area. Currently, SMILES-based alignment offers a feasible approach, though we acknowledge its limitations, as small molecular changes can indeed create large variations in SMILES. However, to our knowledge, there are no better alternatives for aligning both enzyme and reaction (molecule) evolution at this time, especially given the differences in their modalities. The sensitivity of sequence alignments underscores this issue: SMILES strings often vary significantly among proteins within the same MSA. This sensitivity reflects the diversity of enzyme-catalyzed reactions, where two similar enzymes may catalyze distinct reactions. Additionally, the enzyme space is substantially larger than the reaction space, amplifying the divergence in SMILES even among enzymes within the same MSA.
>
> #### Looking forward, we aim to address this with our proposed coMSAformer, inspired by MSAformer, to facilitate few-shot experiments on this topic. This approach could help improve the alignment of these evolutionary modalities and overcome some limitations of SMILES-based methods, which we agree is a fascinating area for further exploration.
>
> #### In addition we show more experiments stating the importance of having co-MSAs in modeling. Initially, we trained EnzymeFlow on 32-residue pockets, which provided limited structural and functional signal strength compared to using the full enzyme backbone. To address this, we re-trained EnzymeFlow on a dataset of 64-residue pockets, and this change was also applied to baseline models for a fair comparison. With this approach, we observed notable improvements: the generated pockets showed reduced RMSD, improved structural alignment scores, and lower embedding distances (using ESM3 encoding for pocket sequences and structures).  These new results demonstrate that the inclusion of co-MSAs and EC conditions provides significant improvements in the generation of enzyme catalytic pockets, highlighting the importance of these additional elements in improving both the structural and functional accuracy of the model.
> ```
> ## Enzyme-ablation denotes an ablation model without coMSAs and EC guidance in training.
> |                   |                    | rmsd | rmsd | rmsd | tm   | tm   | tm   | Dist  | Dist  | Dist  |
> | ----------------- | ------------------ | ---- | ---- | ---- | ---- | ---- | ---- | ----- | ----- | ----- |
> | Pocket Evaluation |                    | Top1 | Top3 | Mean | Top1 | Top3 | Mean | Top1  | Top3  | Mean  |
> | ----------------- | ------------------ | ---- | ---- | ---- | ---- | ---- | ---- | ----- | ----- | ----- |
> | EC1               | EnzymeFlow-abation | 3.76 | 3.90 | 4.19 | 0.29 | 0.28 | 0.26 | 19.82 | 20.29 | 21.48 |
> |                   | EnzymeFlow         | 1.88 | 2.09 | 2.41 | 0.67 | 0.65 | 0.59 | 0.25  | 0.58  | 1.54  |
> | ----------------- | ------------------ | ---- | ---- | ---- | ---- | ---- | ---- | ----- | ----- | ----- |
> | EC2               | EnzymeFlow-abation | 4.02 | 4.19 | 4.35 | 0.23 | 0.23 | 0.21 | 50.31 | 52.33 | 54.15 |
> |                   | EnzymeFlow         | 3.65 | 3.80 | 4.09 | 0.33 | 0.31 | 0.28 | 1.03  | 1.85  | 4.40  |
> | ----------------- | ------------------ | ---- | ---- | ---- | ---- | ---- | ---- | ----- | ----- | ----- |
> | EC3               | EnzymeFlow-abation | 3.46 | 3.69 | 4.12 | 0.31 | 0.29 | 0.27 | 9.84  | 10.02 | 11.14 |
> |                   | EnzymeFlow         | 3.18 | 3.32 | 3.84 | 0.39 | 0.36 | 0.32 | 0.42  | 1.08  | 2.19  |
> | ----------------- | ------------------ | ---- | ---- | ---- | ---- | ---- | ---- | ----- | ----- | ----- |
> | EC4               | EnzymeFlow-abation | 3.31 | 3.48 | 3.91 | 0.32 | 0.30 | 0.27 | 18.02 | 18.56 | 19.96 |
> |                   | EnzymeFlow         | 2.58 | 2.73 | 2.96 | 0.56 | 0.54 | 0.51 | 0.50  | 1.07  | 2.43  |
> | ----------------- | ------------------ | ---- | ---- | ---- | ---- | ---- | ---- | ----- | ----- | ----- |
> | EC5               | EnzymeFlow-abation | 3.48 | 3.66 | 4.06 | 0.29 | 0.28 | 0.26 | 13.71 | 14.01 | 14.81 |
> |                   | EnzymeFlow         | 2.87 | 3.18 | 3.62 | 0.44 | 0.43 | 0.39 | 0.74  | 1.07  | 1.78  |
> | ----------------- | ------------------ | ---- | ---- | ---- | ---- | ---- | ---- | ----- | ----- | ----- |
> | EC6               | EnzymeFlow-abation | 3.78 | 3.89 | 4.33 | 0.29 | 0.27 | 0.26 | 21.00 | 22.87 | 24.19 |
> |                   | EnzymeFlow         | 3.14 | 3.32 | 3.73 | 0.43 | 0.40 | 0.36 | 1.23  | 1.45  | 2.56  |
> ```

---

> ### Author Response · Authors · 2024-11-13
> **Answers to Weaknesses Part3**
>
> ## A3:
> #### We also agree with what you suggested, only observing enzyme commission numbers can limit to examining the enzyme catalytic ability. To further demonstrate the functional validity of EnzymeFlow-generated pockets, we have included additional metrics such as optimal pH, turnover number, and substrate specificity. These metrics indicate that EnzymeFlow-generated pockets align closely with baseline-generated pockets, further supporting the model's capacity to generate functionally relevant structures. These are newly conducted experiments, where pockets for RFDiffusionAA are extracted from a whole protein to compare to those generated by EnzymeFlow.
>
> ```
> |                   |                    | pH   | kcat | km   |
> | ----------------- | ------------------ | ---- | ---- | ---- |
> | Pocket Evaluation |                    | mean | mean | mean |
> | ----------------- | ------------------ | ---- | ---- | ---- |
> | EC1               | ground truth       | 7.86 | 2.86 | 0.33 |
> |                   | RFDiffusionAA      | 8.85 | 2.17 | 0.24 |
> |                   | EnzymeFlow         | 7.54 | 2.45 | 0.37 |
> | ----------------- | ------------------ | ---- | ---- | ---- |
> | EC2               | ground truth       | 7.68 | 1.93 | 0.31 |
> |                   | RFDiffusionAA      | 8.89 | 1.55 | 0.19 |
> |                   | EnzymeFlow         | 7.58 | 2.03 | 0.27 |
> | ----------------- | ------------------ | ---- | ---- | ---- |
> | EC3               | ground truth       | 7.49 | 2.18 | 0.09 |
> |                   | RFDiffusionAA      | 8.79 | 1.36 | 0.08 |
> |                   | EnzymeFlow         | 7.64 | 2.49 | 0.08 |
> | ----------------- | ------------------ | ---- | ---- | ---- |
> | EC4               | ground truth       | 7.82 | 1.57 | 0.09 |
> |                   | RFDiffusionAA      | 8.98 | 2.19 | 0.10 |
> |                   | EnzymeFlow         | 7.40 | 2.03 | 0.09 |
> | ----------------- | ------------------ | ---- | ---- | ---- |
> | EC5               | ground truth       | 7.10 | 2.71 | 0.14 |
> |                   | RFDiffusionAA      | 8.88 | 1.18 | 0.14 |
> |                   | EnzymeFlow         | 7.47 | 2.00 | 0.14 |
> | ----------------- | ------------------ | ---- | ---- | ---- |
> | EC6               | ground truth       | 7.63 | 2.03 | 0.47 |
> |                   | RFDiffusionAA      | 8.98 | 1.99 | 0.33 |
> |                   | EnzymeFlow         | 7.43 | 2.92 | 0.46 |
>
> ```
>
> ## A4:
> #### Indeed, our methodology does not rely on binding structures but instead utilizes a ligand transplantation approach with PDB-REDO. This computational chemistry tool is specifically used for us to identify catalytic regions, not conventional binding sites, to reflect the distinct requirements of catalysis. Unlike binding structures, which typically focus on tight binding affinity, our approach identifies catalytic regions where enzymes facilitate reaction processes. Notably, tight binding between an enzyme and its substrate can actually reduce catalytic efficiency, as it may hinder substrate turnover. Instead, catalytic regions are often characterized by flexible, transient interactions that allow for substrate transformation without the constraints of a rigid binding affinity. By targeting these regions, we aim to capture the enzyme's catalytic functionality rather than binding properties, which aligns more closely with the dynamic nature of enzymatic catalysis. We acknowledge that incorporating evaluations from experimental, non-synthetic datasets would further strengthen validation and help mitigate any residual biases. This is a valuable direction, and we are exploring additional validations with experimental datasets to ensure robustness across diverse catalytic environments.

---

> ### Author Response · Authors · 2024-11-13
> **Answers to Questions Part1**
>
> ## A1:
>
> #### First, [1] suggests relationships between enzyme commissions, enzyme functions, and catalytic mechanisms. Meanwhile, we also agree with what you suggested, only observing enzyme commission numbers can limit to examining the enzyme catalytic ability. To further demonstrate the functional validity of EnzymeFlow-generated pockets, we have included additional metrics such as optimal pH, turnover number, and substrate specificity. These metrics indicate that EnzymeFlow-generated pockets align closely with baseline-generated pockets, further supporting the model's capacity to generate functionally relevant structures.
> ```
> ## Enzyme-ablation denotes an ablation model without coMSAs and EC guidance in training.
> |                   |                    | pH   | kcat | km   |
> | ----------------- | ------------------ | ---- | ---- | ---- |
> | Pocket Evaluation |                    | mean | mean | mean |
> | ----------------- | ------------------ | ---- | ---- | ---- |
> | EC1               | ground truth       | 7.86 | 2.86 | 0.33 |
> |                   | EnzymeFlow-abation | 8.40 | 1.08 | 0.28 |
> |                   | RFDiffusionAA      | 8.85 | 2.17 | 0.24 |
> |                   | EnzymeFlow         | 7.54 | 2.45 | 0.37 |
> | ----------------- | ------------------ | ---- | ---- | ---- |
> | EC2               | ground truth       | 7.68 | 1.93 | 0.31 |
> |                   | EnzymeFlow-abation | 8.48 | 0.72 | 0.20 |
> |                   | RFDiffusionAA      | 8.89 | 1.55 | 0.19 |
> |                   | EnzymeFlow         | 7.58 | 2.03 | 0.27 |
> | ----------------- | ------------------ | ---- | ---- | ---- |
> | EC3               | ground truth       | 7.49 | 2.18 | 0.09 |
> |                   | EnzymeFlow-abation | 8.56 | 0.79 | 0.08 |
> |                   | RFDiffusionAA      | 8.79 | 1.36 | 0.08 |
> |                   | EnzymeFlow         | 7.64 | 2.49 | 0.08 |
> | ----------------- | ------------------ | ---- | ---- | ---- |
> | EC4               | ground truth       | 7.82 | 1.57 | 0.09 |
> |                   | EnzymeFlow-abation | 8.48 | 0.78 | 0.10 |
> |                   | RFDiffusionAA      | 8.98 | 2.19 | 0.10 |
> |                   | EnzymeFlow         | 7.40 | 2.03 | 0.09 |
> | ----------------- | ------------------ | ---- | ---- | ---- |
> | EC5               | ground truth       | 7.10 | 2.71 | 0.14 |
> |                   | EnzymeFlow-abation | 8.32 | 0.95 | 0.12 |
> |                   | RFDiffusionAA      | 8.88 | 1.18 | 0.14 |
> |                   | EnzymeFlow         | 7.47 | 2.00 | 0.14 |
> | ----------------- | ------------------ | ---- | ---- | ---- |
> | EC6               | ground truth       | 7.63 | 2.03 | 0.47 |
> |                   | EnzymeFlow-abation | 8.44 | 0.84 | 0.27 |
> |                   | RFDiffusionAA      | 8.98 | 1.99 | 0.33 |
> |                   | EnzymeFlow         | 7.43 | 2.92 | 0.46 |
> ```
>
> ##### [1] Cuesta, S.M., Rahman, S.A., Furnham, N. and Thornton, J.M., 2015. The classification and evolution of enzyme function. Biophysical journal, 109(6), pp.1082-1086.

---

> ### Author Response · Authors · 2024-11-13
> **Answers to Questions Part2**
>
> ## A1:
>
> #### Second, we trained an ablation model that incorporates only reaction conditions, but excludes additional elements such as co-MSAs and EC conditions. This model, which we refer to as EnzymeFlow-ablation, was trained the same as we did for EnzymeFlow. The performance of the EnzymeFlow-ablation model is compared to the full EnzymeFlow model, which incorporates co-MSAs, EC conditions, and reaction conditioning. The results show that the EnzymeFlow model outperforms the EnzymeFlow-ablation model, as evidenced by the following metrics: lower RMSD, higher alignment scores, lower embedding distance (we use ESM3 encoding for pocket sequences and structures)
> ```
> |                   |                    | rmsd | rmsd | rmsd | tm   | tm   | tm   | Dist  | Dist  | Dist  |
> | ----------------- | ------------------ | ---- | ---- | ---- | ---- | ---- | ---- | ----- | ----- | ----- |
> | Pocket Evaluation |                    | Top1 | Top3 | Mean | Top1 | Top3 | Mean | Top1  | Top3  | Mean  |
> | ----------------- | ------------------ | ---- | ---- | ---- | ---- | ---- | ---- | ----- | ----- | ----- |
> | EC1               | EnzymeFlow-abation | 3.76 | 3.90 | 4.19 | 0.29 | 0.28 | 0.26 | 19.82 | 20.29 | 21.48 |
> |                   | RFDiffusionAA      | 3.71 | 3.82 | 4.18 | 0.28 | 0.27 | 0.25 | 7.33  | 8.05  | 9.28  |
> |                   | EnzymeFlow         | 1.88 | 2.09 | 2.41 | 0.67 | 0.65 | 0.59 | 0.25  | 0.58  | 1.54  |
> | ----------------- | ------------------ | ---- | ---- | ---- | ---- | ---- | ---- | ----- | ----- | ----- |
> | EC2               | EnzymeFlow-abation | 4.02 | 4.19 | 4.35 | 0.23 | 0.23 | 0.21 | 50.31 | 52.33 | 54.15 |
> |                   | RFDiffusionAA      | 3.47 | 3.78 | 4.17 | 0.27 | 0.27 | 0.25 | 26.91 | 27.84 | 30.44 |
> |                   | EnzymeFlow         | 3.65 | 3.80 | 4.09 | 0.33 | 0.31 | 0.28 | 1.03  | 1.85  | 4.40  |
> | ----------------- | ------------------ | ---- | ---- | ---- | ---- | ---- | ---- | ----- | ----- | ----- |
> | EC3               | EnzymeFlow-abation | 3.46 | 3.69 | 4.12 | 0.31 | 0.29 | 0.27 | 9.84  | 10.02 | 11.14 |
> |                   | RFDiffusionAA      | 3.25 | 3.50 | 3.96 | 0.31 | 0.30 | 0.27 | 10.83 | 10.96 | 12.22 |
> |                   | EnzymeFlow         | 3.18 | 3.32 | 3.84 | 0.39 | 0.36 | 0.32 | 0.42  | 1.08  | 2.19  |
> | ----------------- | ------------------ | ---- | ---- | ---- | ---- | ---- | ---- | ----- | ----- | ----- |
> | EC4               | EnzymeFlow-abation | 3.31 | 3.48 | 3.91 | 0.32 | 0.30 | 0.27 | 18.02 | 18.56 | 19.96 |
> |                   | RFDiffusionAA      | 3.00 | 3.29 | 3.79 | 0.30 | 0.29 | 0.27 | 14.62 | 15.62 | 17.39 |
> |                   | EnzymeFlow         | 2.58 | 2.73 | 2.96 | 0.56 | 0.54 | 0.51 | 0.50  | 1.07  | 2.43  |
> | ----------------- | ------------------ | ---- | ---- | ---- | ---- | ---- | ---- | ----- | ----- | ----- |
> | EC5               | EnzymeFlow-abation | 3.48 | 3.66 | 4.06 | 0.29 | 0.28 | 0.26 | 13.71 | 14.01 | 14.81 |
> |                   | RFDiffusionAA      | 3.39 | 3.61 | 3.98 | 0.31 | 0.29 | 0.26 | 7.83  | 8.23  | 8.93  |
> |                   | EnzymeFlow         | 2.87 | 3.18 | 3.62 | 0.44 | 0.43 | 0.39 | 0.74  | 1.07  | 1.78  |
> | ----------------- | ------------------ | ---- | ---- | ---- | ---- | ---- | ---- | ----- | ----- | ----- |
> | EC6               | EnzymeFlow-abation | 3.78 | 3.89 | 4.33 | 0.29 | 0.27 | 0.26 | 21.00 | 22.87 | 24.19 |
> |                   | RFDiffusionAA      | 3.43 | 3.69 | 4.12 | 0.30 | 0.29 | 0.26 | 19.08 | 19.73 | 21.21 |
> |                   | EnzymeFlow         | 3.14 | 3.32 | 3.73 | 0.43 | 0.40 | 0.36 | 1.23  | 1.45  | 2.56  |
>
> ```
>
> #### Third, thank you for the suggestion. Accurately predicting EC classes is indeed challenging for this generative task. While incorporating EC conditions in generation has shown improvements in backbone structure quality, accurately predicting EC function remains a difficult aspect. We are actively exploring Reinforcement Learning with Human Feedback (RLHF) to enhance generation; specifically, we are using CLEAN as an oracle for EC prediction as part of the reward mechanism. This approach aims to further guide the model to incorporate EC information both implicitly and explicitly, helping to improve accuracy in EC function prediction.

---

> ### Comment · Reviewer_XPne · 2024-11-26
>
> Thank you for your responses. While this work presents reasonable contributions to enzyme design, I believe there is room for improvement in the writing and empirical study to enhance its overall rigor.
>
> **Regarding Answer 1:**
>
> It would be beneficial to explicitly clarify in the introduction or methods section that you employed the "cross-attention mechanism" to capture chemical changes. However, I do not see the use of the cross-attention mechanism as a methodological contribution, as it is a commonly adopted approach. Additionally, no specific modifications or designs are mentioned to demonstrate how this mechanism is particularly suited for capturing chemical reactions. If you have introduced such enhancements, it would be helpful to clearly describe them in the manuscript.
>
> **Regarding Answer 3:**
>
> I found it challenging to fully understand the metrics you provided, including their definitions, computation methods, and more importantly, relevance to enzyme functionality.

---

### Official Review · Reviewer_xrVG · 2024-11-03

**Soundness:** 2
**Presentation:** 4
**Contribution:** 3
**Rating:** 6
**Confidence:** 4

**Summary:**

This paper introduces a novel approach to enzyme pocket design, emphasizing functional relevance from the outset. The authors propose EnzymeFlow, a flow matching model conditioned on both substrates and products, trained to predict enzyme pocket structure, sequence, EC-class, and enzyme-reaction co-evolution concurrently. They introduce EnzymeFill, a newly curated dataset tailored for the enzyme pocket design task, comprising enzyme-reaction pairs with precise pocket structures. Experimental results provide substantial evidence of EnzymeFlow's effectiveness in generating functionally reliable enzyme pockets that align with catalytic requirements.

**Strengths:**

- EnzymeFlow shifts the focus from static ligand-protein interactions to more complex enzyme-substrate interactions, a more functionally relevant framework for catalysis. This paradigm is well-motivated.
- EnzymeFlow proposes a distinct formulation for enzyme design that departs from traditional protein design workflows, emphasising the importance of function in modelling.
- The proposed method is technically sound.

**Weaknesses:**

- Despite the promising performance on pocket design, the utility of EnzymeFlow in whole enzyme design remains under-explored, limiting insight into its full application potential.
- As the experimental results show, although EnzymeFlow achieves better functional performance compared to other baseline methods, the ECacc is still somewhat low to conclude that EnzymeFlow adequately captures the catalytic requirements.

**Questions:**

1. Could the authors clarify the rationale for focusing on a pathway from pocket to motif-scaffolding rather than complete enzyme design?
2. Testing the full protein configurations by integrating EnzymeFlow with recent motif-scaffolding methods or inpainting methods and comment on the self-consistency of the designed pockets would be helpful in understanding the application potential.
3. In Table 2, the comparison with RFDiffusionAA may not be ideal as RFDiffusionAA is designed for whole proteins rather than specific pockets. It may be more appropriate to extract the pocket from a whole protein generated by RFDiffusionAA and compare it with EnzymeFlow.
4. Could the authors provide an analysis of EnzymeFlow's performance across different substrate classes?

---

> ### Author Response · Authors · 2024-11-13
> **Answer to Weaknesses Part1**
>
> #### Thanks for the feedback. We have been conducting more (a lot) experiments in the past month to get ready for this rebuttal, and aim to address the weaknesses and questions, to demonstrate model’s robustness in designing catalytic pockets:
>
> ## A1:
> #### Thanks for acknowledging the importance of our work. You are right that only generating catalytic pockets could really limit the utility of EnzymeFlow. Therefore, we have developed an inpainting network that recovers the catalytic pockets to full enzymes. We have more discussions regarding the inpainting model in the ‘’Answer to Questions section.
>
> ## A2:
> #### We also agree with what you suggested, only observing enzyme commission numbers can limit to examining the enzyme catalytic ability. To further demonstrate the functional validity of EnzymeFlow-generated pockets, we have included additional metrics such as optimal pH, turnover number, and substrate specificity. These metrics indicate that EnzymeFlow-generated pockets align closely with baseline-generated pockets, further supporting the model's capacity to generate functionally relevant structures. These are newly conducted experiments, where pockets for RFDiffusionAA are extracted from a whole protein to compare to those generated by EnzymeFlow.
>
> ```
> |                   |                    | pH   | kcat | km   |
> | ----------------- | ------------------ | ---- | ---- | ---- |
> | Pocket Evaluation |                    | mean | mean | mean |
> | ----------------- | ------------------ | ---- | ---- | ---- |
> | EC1               | ground truth       | 7.86 | 2.86 | 0.33 |
> |                   | RFDiffusionAA      | 8.85 | 2.17 | 0.24 |
> |                   | EnzymeFlow         | 7.54 | 2.45 | 0.37 |
> | ----------------- | ------------------ | ---- | ---- | ---- |
> | EC2               | ground truth       | 7.68 | 1.93 | 0.31 |
> |                   | RFDiffusionAA      | 8.89 | 1.55 | 0.19 |
> |                   | EnzymeFlow         | 7.58 | 2.03 | 0.27 |
> | ----------------- | ------------------ | ---- | ---- | ---- |
> | EC3               | ground truth       | 7.49 | 2.18 | 0.09 |
> |                   | RFDiffusionAA      | 8.79 | 1.36 | 0.08 |
> |                   | EnzymeFlow         | 7.64 | 2.49 | 0.08 |
> | ----------------- | ------------------ | ---- | ---- | ---- |
> | EC4               | ground truth       | 7.82 | 1.57 | 0.09 |
> |                   | RFDiffusionAA      | 8.98 | 2.19 | 0.10 |
> |                   | EnzymeFlow         | 7.40 | 2.03 | 0.09 |
> | ----------------- | ------------------ | ---- | ---- | ---- |
> | EC5               | ground truth       | 7.10 | 2.71 | 0.14 |
> |                   | RFDiffusionAA      | 8.88 | 1.18 | 0.14 |
> |                   | EnzymeFlow         | 7.47 | 2.00 | 0.14 |
> | ----------------- | ------------------ | ---- | ---- | ---- |
> | EC6               | ground truth       | 7.63 | 2.03 | 0.47 |
> |                   | RFDiffusionAA      | 8.98 | 1.99 | 0.33 |
> |                   | EnzymeFlow         | 7.43 | 2.92 | 0.46 |
>
> ```

---

> ### Author Response · Authors · 2024-11-13
> **Answers to Questions Part1**
>
> ## A1:
> #### We appreciate the opportunity to clarify our rationale. As discussed in the appendices, our focus on catalytic pocket design rather than complete enzyme design stems from both the scientific challenges and strategic advantages of this approach. Designing a full enzyme from scratch is an extremely complex task, with two primary hurdles: (1) accurately generating a complete, functional structure, and (2) identifying the optimal catalytic regions where catalysis actually occurs. Catalytic pockets are the most critical regions of enzymes, as they directly interact with substrates, driving chemical transformations. Focusing on catalytic pockets first allows us to solve a foundational aspect of enzyme design by establishing effective reaction sites, which are essential for catalysis.
>
> #### By taking a bottom-up approach that prioritizes catalytic pocket design, we address the most functionally important region first, aiming to achieve catalytic efficacy at the reaction site. Once reliable methods for pocket design are in place, we can extend this work toward scaffold motifs and eventually full enzyme design, building from a verified functional core. This approach also offers a more achievable pathway for the community: by solving pocket design first, researchers can progressively build up to complete enzymes, reducing the challenge of full enzyme generation. Additionally, while the complete enzyme structure is crucial for overall stability and foldability, its function is driven primarily by the catalytic pocket. We believe this focused pathway will accelerate progress toward enzyme discovery, providing actionable insights into catalytic design that can ultimately be scaled up to full enzyme architectures.

---

> ### Author Response · Authors · 2024-11-13
> **Answers to Questions Part2**
>
> ## A2 & A3:
> #### In the past months, we have developed an inpainting model on top of the pocket design model for full enzyme design. The inpainting model is trained on EnzymeFill to satisfy this task. Initially, we trained EnzymeFlow on 32-residue pockets, which provided limited structural and functional signal strength compared to using the full enzyme backbone. To address this, we re-trained EnzymeFlow on a dataset of 64-residue pockets. The inpainting is designed to  inpaint the catalytic pockets to full enzyme structures as you are expecting to see. And the entire framework works by generating pockets from chemical reactions first, then generating enzymes from pockets.  And for comparison to RFDiffusionAA, we extract the pocket from a whole protein generated by RFDiffusionAA and compare it with EnzymeFlow. We should pocket- and enzyme-level evaluations.
>
> With this approach, we observed notable improvements: the generated pockets and full enzymes showed reduced RMSD, improved structural alignment scores, and lower embedding distances (using ESM3 encoding for pocket and enzyme sequences and structures). These enhancements suggest that our model-generated ones more closely resemble real catalytic sites in structure and sequence features, both critical indicators of potential catalytic activity.
> ```
> |                   |               | rmsd | rmsd | rmsd | tm   | tm   | tm   | Dist  | Dist  | Dist  |
> | ----------------- | ------------- | ---- | ---- | ---- | ---- | ---- | ---- | ----- | ----- | ----- |
> | Pocket Evaluation |               | Top1 | Top3 | Mean | Top1 | Top3 | Mean | Top1  | Top3  | Mean  |
> | ----------------- | ------------- | ---- | ---- | ---- | ---- | ---- | ---- | ----- | ----- | ----- |
> | EC1               | RFDiffusionAA | 3.71 | 3.82 | 4.18 | 0.28 | 0.27 | 0.25 | 7.33  | 8.05  | 9.28  |
> |                   | EnzymeFlow    | 1.88 | 2.09 | 2.41 | 0.67 | 0.65 | 0.59 | 0.25  | 0.58  | 1.54  |
> | ----------------- | ------------- | ---- | ---- | ---- | ---- | ---- | ---- | ----- | ----- | ----- |
> | EC2               | RFDiffusionAA | 3.47 | 3.78 | 4.17 | 0.27 | 0.27 | 0.25 | 26.91 | 27.84 | 30.44 |
> |                   | EnzymeFlow    | 3.65 | 3.80 | 4.09 | 0.33 | 0.31 | 0.28 | 1.03  | 1.85  | 4.40  |
> | ----------------- | ------------- | ---- | ---- | ---- | ---- | ---- | ---- | ----- | ----- | ----- |
> | EC3               | RFDiffusionAA | 3.25 | 3.50 | 3.96 | 0.31 | 0.30 | 0.27 | 10.83 | 10.96 | 12.22 |
> |                   | EnzymeFlow    | 3.18 | 3.32 | 3.84 | 0.39 | 0.36 | 0.32 | 0.42  | 1.08  | 2.19  |
> | ----------------- | ------------- | ---- | ---- | ---- | ---- | ---- | ---- | ----- | ----- | ----- |
> | EC4               | RFDiffusionAA | 3.00 | 3.29 | 3.79 | 0.30 | 0.29 | 0.27 | 14.62 | 15.62 | 17.39 |
> |                   | EnzymeFlow    | 2.58 | 2.73 | 2.96 | 0.56 | 0.54 | 0.51 | 0.50  | 1.07  | 2.43  |
> | ----------------- | ------------- | ---- | ---- | ---- | ---- | ---- | ---- | ----- | ----- | ----- |
> | EC5               | RFDiffusionAA | 3.39 | 3.61 | 3.98 | 0.31 | 0.29 | 0.26 | 7.83  | 8.23  | 8.93  |
> |                   | EnzymeFlow    | 2.87 | 3.18 | 3.62 | 0.44 | 0.43 | 0.39 | 0.74  | 1.07  | 1.78  |
> | ----------------- | ------------- | ---- | ---- | ---- | ---- | ---- | ---- | ----- | ----- | ----- |
> | EC6               | RFDiffusionAA | 3.43 | 3.69 | 4.12 | 0.30 | 0.29 | 0.26 | 19.08 | 19.73 | 21.21 |
> |                   | EnzymeFlow    | 3.14 | 3.32 | 3.73 | 0.43 | 0.40 | 0.36 | 1.23  | 1.45  | 2.56  |
> ```

---

> ### Author Response · Authors · 2024-11-13
> **Answers to Questions Part3**
>
> ## A2 & A3
> ```
> |                   |               | rmsd | rmsd | rmsd | tm   | tm   | tm   | Dist  | Dist  | Dist  |
> | ----------------- | ------------- | ---- | ---- | ---- | ---- | ---- | ---- | ----- | ----- | ----- |
> | Enzyme Evaluation |               | Top1 | Top3 | Mean | Top1 | Top3 | Mean | Top1  | Top3  | Mean  |
> | ----------------- | ------------- | ---- | ---- | ---- | ---- | ---- | ---- | ----- | ----- | ----- |
> | EC1               | RFDiffusionAA | 4.46 | 4.76 | 5.36 | 0.48 | 0.44 | 0.38 | 22.38 | 23.79 | 26.89 |
> |                   | EnzymeFlow    | 4.13 | 4.36 | 4.83 | 0.62 | 0.58 | 0.53 | 2.41  | 2.92  | 5.49  |
> | ----------------- | ------------- | ---- | ---- | ---- | ---- | ---- | ---- | ----- | ----- | ----- |
> | EC2               | RFDiffusionAA | 4.88 | 5.08 | 5.58 | 0.43 | 0.42 | 0.38 | 5.42  | 6.31  | 8.91  |
> |                   | EnzymeFlow    | 5.27 | 5.45 | 5.99 | 0.37 | 0.36 | 0.32 | 5.15  | 6.25  | 8.43  |
> | ----------------- | ------------- | ---- | ---- | ---- | ---- | ---- | ---- | ----- | ----- | ----- |
> | EC3               | RFDiffusionAA | 5.13 | 5.28 | 5.71 | 0.38 | 0.36 | 0.33 | 4.95  | 6.74  | 10.59 |
> |                   | EnzymeFlow    | 5.08 | 5.26 | 5.76 | 0.38 | 0.38 | 0.33 | 10.46 | 11.74 | 15.06 |
> | ----------------- | ------------- | ---- | ---- | ---- | ---- | ---- | ---- | ----- | ----- | ----- |
> | EC4               | RFDiffusionAA | 4.49 | 4.68 | 5.25 | 0.45 | 0.42 | 0.37 | 65.60 | 69.49 | 78.31 |
> |                   | EnzymeFlow    | 4.02 | 4.15 | 4.59 | 0.57 | 0.55 | 0.52 | 9.61  | 9.85  | 10.49 |
> | ----------------- | ------------- | ---- | ---- | ---- | ---- | ---- | ---- | ----- | ----- | ----- |
> | EC5               | RFDiffusionAA | 4.48 | 4.71 | 5.22 | 0.50 | 0.46 | 0.40 | 2.87  | 3.66  | 5.31  |
> |                   | EnzymeFlow    | 5.02 | 5.32 | 5.80 | 0.43 | 0.41 | 0.39 | 4.58  | 5.20  | 7.14  |
> | ----------------- | ------------- | ---- | ---- | ---- | ---- | ---- | ---- | ----- | ----- | ----- |
> | EC6               | RFDiffusionAA | 4.53 | 4.89 | 5.50 | 0.45 | 0.43 | 0.38 | 19.38 | 20.76 | 25.87 |
> |                   | EnzymeFlow    | 5.76 | 6.00 | 6.42 | 0.34 | 0.32 | 0.28 | 12.24 | 14.73 | 16.87 |
> ```
>
> ## A4:
> #### Thank you for the question. Indeed, the analysis on EC (Enzyme Commission) classes does relate to substrate specificity, as EC classification is based on the types of reactions an enzyme catalyzes, which is directly influenced by substrate class. By examining performance across different EC classes, we effectively cover a range of substrate types and provide insight into how EnzymeFlow performs with diverse substrate classes.

---

### Official Review · Reviewer_yZeM · 2024-11-03

**Soundness:** 2
**Presentation:** 2
**Contribution:** 2
**Rating:** 3
**Confidence:** 5

**Summary:**

EnzymeFlow introduces a new approach to conditionally generate enzyme pockets conditioned on the reaction of interest. They use a modified MultiFlow approach that is conditioned on both substrates and products, they use an auxiliary EC flow, and they use a novel reaction-protein coevolution MSA to enhance the generation process. The evaluate the model across a number of metrics that are considered relevant for enzymatic activity including cRMSD and TM-score of the pockets; predicted binding affinities, chai score, amino acid recovery and predicted EC accuracy.

**Strengths:**

The paper attempts to solve a key task on the road to de novo enzyme design -- namely generating enzyme pockets which are conditioned on the reaction of interest. The paper has a number of strengths:
1. The authors present an interesting idea -- that enzymes and their associated reactions co-evolve. They attempt to use this hypothesis to learn representations of this co-evolution to enhance generation.
2. They introduce a new dataset of enzyme pockets.
3. They combine a number of methods applied in different applications on a new application.

**Weaknesses:**

The main weakness with the paper is the results:
1. It is unclear whether the improvements presented by the authors are significant or not. Please provide confidence intervals such that it is easier to evaluate the significance of the results.
2. It is unclear to me whether these scores meaningfully correlate with enzyme function. For example, TM-scores of below 0.3 represent quite a significant difference from the ground truth structure, meaning that the generated pockets are still quite different. Why is this? Furthermore, it seems that the generated pockets recover the correct amino acids a small fraction of the time and additionally only have the same predicted EC a small fraction of the time. A strong result would be that generated structures are similar to ground truth, have close or exact amino acid identities and (if they are truly functionally similar) should have the same predicted EC numbers.
3. The novel contributions of the author seem to also have very little impact on performance -- are the results w/o coevolution, pretraining or both statistically significant?
4. While I rather like your attempt to incorporate co-evolutionary reaction-enzyme information, I am not convinced that SMILES of reactions is a good approach as small changes in molecules can have large effects on SMILE strings. Are there alternative ways of doing this? In how many cases do SMILES strings change significantly across different proteins in the same MSA?
5. Size of evaluation set.

**Questions:**

I will be happy to review my score if the authors can answer the above questions convincingly as well as provide the following additional information:
1. Please clearly indicate what novel contributions come from the authors versus have been taken from other papers and combined. For example, it seems that most of the generative model is based on MultiFlow and not introduced by the authors. Is the approach for ligand-conditioning developed by you or by others? The co-evolution for example (to the best of my knowledge) is a novel contribution of the authors.
2. Please provide a convincing argument to the significance of the results and an explanation as to why the numbers being so low is acceptable. For example, would you expect pockets generated by your model to be active?
3. You claim that other approaches simply consider static behavior of proteins and their ligands, but I do not see any consideration of dynamic behavior in your model beyond considering both substrate and product. Do you have reason to believe that your model learns this implicitly? If so please provide evidence of this.
4. How does your model perform using a standard protein MSA compared to your special coevolution MSA?
5. Please provide results for vanilla MultiFlow without any additional elements as a baseline.
6. Please provide a TM-score for the active site against the entire protein, not just the pocket.

---

> ### Author Response · Authors · 2024-11-13
> **Answer to Weaknesses Part1**
>
> #### Thanks for the feedback. We have been conducting more (a lot) experiments in the past month to get ready for this rebuttal, and aim to address the weaknesses and questions, to demonstrate model’s robustness in designing catalytic pockets:
>
> ## A1:
> #### We have reported confidence intervals by measuring performance at top-1, top-K, and mean values. This approach aligns with real experimental workflows in wet lab settings, where experimentalists progressively test candidates in a ranked order—from top-1 to top-2, top-3, up to top-K or top-N. By presenting results this way, we aim to make our findings directly applicable and more meaningful for experimental validation.
>
> ## A2:
> #### The primary reason for the difference in structural similarity (such as TM-scores) is likely due to the model operating on 32-residue pockets rather than the entire enzyme backbone, which inherently limits the training signal strength. To address this, we have re-trained EnzymeFlow (and the ablation model) on a dataset of 64-residue pockets, doubling the context available for learning. Results are now included in the "Answers to Questions" section, and they support our hypothesis: increasing pocket size enhances the training signal, leading to improved structural alignment and functional accuracy.
>
> #### The model is designed to learn a distribution of potential catalytic pocket sequences, not a single fixed sequence. For example, if pockets A, B, and C can all catalyze the same reaction, the model learns a distribution across these pocket sequences rather than memorizing any one specific sequence. As a result, comparing the generated sequence to a single ground truth sequence (e.g., pocket A) will yield a lower AAR score, reflecting the model’s probabilistic approach rather than exact sequence recall.
>
> #### To further demonstrate the functional validity of EnzymeFlow-generated pockets, we have included additional metrics such as optimal pH, turnover number, and substrate specificity. These metrics indicate that EnzymeFlow-generated pockets align closely with baseline-generated pockets, further supporting the model's capacity to generate functionally relevant structures.
> ```
> |                   |                    | pH   | kcat | km   |
> | ----------------- | ------------------ | ---- | ---- | ---- |
> | Pocket Evaluation |                    | mean | mean | mean |
> | ----------------- | ------------------ | ---- | ---- | ---- |
> | EC1               | ground truth       | 7.86 | 2.86 | 0.33 |
> |                   | EnzymeFlow-abation | 8.40 | 1.08 | 0.28 |
> |                   | RFDiffusionAA      | 8.85 | 2.17 | 0.24 |
> |                   | EnzymeFlow         | 7.54 | 2.45 | 0.37 |
> | ----------------- | ------------------ | ---- | ---- | ---- |
> | EC2               | ground truth       | 7.68 | 1.93 | 0.31 |
> |                   | EnzymeFlow-abation | 8.48 | 0.72 | 0.20 |
> |                   | RFDiffusionAA      | 8.89 | 1.55 | 0.19 |
> |                   | EnzymeFlow         | 7.58 | 2.03 | 0.27 |
> | ----------------- | ------------------ | ---- | ---- | ---- |
> | EC3               | ground truth       | 7.49 | 2.18 | 0.09 |
> |                   | EnzymeFlow-abation | 8.56 | 0.79 | 0.08 |
> |                   | RFDiffusionAA      | 8.79 | 1.36 | 0.08 |
> |                   | EnzymeFlow         | 7.64 | 2.49 | 0.08 |
> | ----------------- | ------------------ | ---- | ---- | ---- |
> | EC4               | ground truth       | 7.82 | 1.57 | 0.09 |
> |                   | EnzymeFlow-abation | 8.48 | 0.78 | 0.10 |
> |                   | RFDiffusionAA      | 8.98 | 2.19 | 0.10 |
> |                   | EnzymeFlow         | 7.40 | 2.03 | 0.09 |
> | ----------------- | ------------------ | ---- | ---- | ---- |
> | EC5               | ground truth       | 7.10 | 2.71 | 0.14 |
> |                   | EnzymeFlow-abation | 8.32 | 0.95 | 0.12 |
> |                   | RFDiffusionAA      | 8.88 | 1.18 | 0.14 |
> |                   | EnzymeFlow         | 7.47 | 2.00 | 0.14 |
> | ----------------- | ------------------ | ---- | ---- | ---- |
> | EC6               | ground truth       | 7.63 | 2.03 | 0.47 |
> |                   | EnzymeFlow-abation | 8.44 | 0.84 | 0.27 |
> |                   | RFDiffusionAA      | 8.98 | 1.99 | 0.33 |
> |                   | EnzymeFlow         | 7.43 | 2.92 | 0.46 |
> ```
>
> ## A3:
> #### The effectiveness of our novel contributions, such as co-evolution and pretraining, was initially constrained by the 32-residue training-sampling strategy. However, upon expanding to a 64-residue training-sampling strategy, we observed a statistically significant performance improvement that better reflects the impact of these contributions. This revised approach demonstrates the enhanced effectiveness of co-evolution and pretraining, as shown in the updated results in the "Answers to Questions" section.

---

> ### Author Response · Authors · 2024-11-13
> **Answer to Weaknesses Part2**
>
> ## A4:
> #### We agree that capturing co-evolutionary information between enzymes and reactions is a challenging yet promising area. Currently, SMILES-based alignment offers a feasible approach, though we acknowledge its limitations, as small molecular changes can indeed create large variations in SMILES. However, to our knowledge, there are no better alternatives for aligning both enzyme and reaction (molecule) evolution at this time, especially given the differences in their modalities. The sensitivity of sequence alignments underscores this issue: SMILES strings often vary significantly among proteins within the same MSA. This sensitivity reflects the diversity of enzyme-catalyzed reactions, where two similar enzymes may catalyze distinct reactions. Additionally, the enzyme space is substantially larger than the reaction space, amplifying the divergence in SMILES even among enzymes within the same MSA.
>
> ####  Looking forward, we aim to address this with our proposed coMSAformer, inspired by MSAformer, to facilitate few-shot experiments on this topic. This approach could help improve the alignment of these evolutionary modalities and overcome some limitations of SMILES-based methods, which we agree is a fascinating area for further exploration.
>
> ## A5:
> #### The primary goal of our evaluation set is to ensure that enzymes and reactions are entirely unique, with no overlap between those in the evaluation set. This design guarantees that our evaluations are conducted on a broad and diverse range of enzyme-reaction pairs, allowing us to test the model's generalization across a wide enzyme-reaction space.
>
> #### Expanding the evaluation set by including more enzymes or reactions would increase the diversity but risks introducing overlap between tested enzymes, reactions, or enzyme-reaction pairs. Such overlap could inadvertently lower the robustness of the evaluation results by reducing the novelty of the test data, which is essential for fair and reliable performance assessment.

---

> ### Author Response · Authors · 2024-11-13
> **Answer to Question Part1**
>
> ## A1:
> ####  FoldFlow introduced the use of frame-based flow matching for protein backbone generation. MultiFlow further extended the FoldFlow approach, integrating protein sequence generation alongside backbone design. MultiFlow introduced a Markov process to handle protein sequence generation in discrete space, a method that we adopt for catalytic pocket generation.
>
>
> ####  Building on these, EnzymeFlow introduces several key contributions: Catalytic Pocket Design with Ligand-Conditioning: We apply ligand and reaction conditioning specifically for catalytic pocket design, extending MultiFlow's generative framework to focus on enzyme active sites conditioned on catalytic needs and reaction chemistry. Enzyme Commission (EC) Conditioning: We incorporate EC-based conditioning to guide pocket design towards functional specificity aligned with particular catalytic functions. Pretraining Strategy: We pretrain EnzymeFlow on the PDBBind2020 dataset, specifically chosen to enhance structural knowledge related to enzyme-ligand interactions. Co-evolutionary Multi-Sequence Alignments (co-MSAs): Unique to EnzymeFlow, we introduce co-MSAs, a novel approach to capture catalytic specificity through co-evolutionary patterns between enzyme sequences and reaction chemistry.
>
>
> #### In summary, FoldFlow initiated frame-based flow matching, MultiFlow adapted it for backbone and sequence co-design, and EnzymeFlon innovates on this by integrating ligand conditioning, EC conditioning, pretraining on enzyme-specific datasets, and introducing co-MSAs for catalytic specificity.
>
> ## A2:
> ####  The results are indeed significant, even if absolute performance metrics appear low, due to the inherent complexity of catalytic pocket generation. Initially, we trained EnzymeFlow on 32-residue pockets, which provided limited structural and functional signal strength compared to using the full enzyme backbone. To address this, we re-trained EnzymeFlow on a dataset of 64-residue pockets, and this change was also applied to baseline models for a fair comparison.
>
> ####  With this 64-residue approach, we observed notable improvements: the generated pockets showed reduced RMSD, improved structural alignment scores, and lower embedding distances (using ESM3 encoding for pocket sequences and structures). These enhancements suggest that our model-generated pockets more closely resemble real catalytic sites in structure and sequence features, both critical indicators of potential catalytic activity.
>
> We observe the increased likelihood of activity as evidenced by structural and sequence similarity improvements when trained on 64-residue pockets.
> ```
> |                   |               | rmsd | rmsd | rmsd | tm   | tm   | tm   | Dist  | Dist  | Dist  |
> | ----------------- | ------------- | ---- | ---- | ---- | ---- | ---- | ---- | ----- | ----- | ----- |
> | Pocket Evaluation |               | Top1 | Top3 | Mean | Top1 | Top3 | Mean | Top1  | Top3  | Mean  |
> | ----------------- | ------------- | ---- | ---- | ---- | ---- | ---- | ---- | ----- | ----- | ----- |
> | EC1               | RFDiffusionAA | 3.71 | 3.82 | 4.18 | 0.28 | 0.27 | 0.25 | 7.33  | 8.05  | 9.28  |
> |                   | EnzymeFlow    | 1.88 | 2.09 | 2.41 | 0.67 | 0.65 | 0.59 | 0.25  | 0.58  | 1.54  |
> | ----------------- | ------------- | ---- | ---- | ---- | ---- | ---- | ---- | ----- | ----- | ----- |
> | EC2               | RFDiffusionAA | 3.47 | 3.78 | 4.17 | 0.27 | 0.27 | 0.25 | 26.91 | 27.84 | 30.44 |
> |                   | EnzymeFlow    | 3.65 | 3.80 | 4.09 | 0.33 | 0.31 | 0.28 | 1.03  | 1.85  | 4.40  |
> | ----------------- | ------------- | ---- | ---- | ---- | ---- | ---- | ---- | ----- | ----- | ----- |
> | EC3               | RFDiffusionAA | 3.25 | 3.50 | 3.96 | 0.31 | 0.30 | 0.27 | 10.83 | 10.96 | 12.22 |
> |                   | EnzymeFlow    | 3.18 | 3.32 | 3.84 | 0.39 | 0.36 | 0.32 | 0.42  | 1.08  | 2.19  |
> | ----------------- | ------------- | ---- | ---- | ---- | ---- | ---- | ---- | ----- | ----- | ----- |
> | EC4               | RFDiffusionAA | 3.00 | 3.29 | 3.79 | 0.30 | 0.29 | 0.27 | 14.62 | 15.62 | 17.39 |
> |                   | EnzymeFlow    | 2.58 | 2.73 | 2.96 | 0.56 | 0.54 | 0.51 | 0.50  | 1.07  | 2.43  |
> | ----------------- | ------------- | ---- | ---- | ---- | ---- | ---- | ---- | ----- | ----- | ----- |
> | EC5               | RFDiffusionAA | 3.39 | 3.61 | 3.98 | 0.31 | 0.29 | 0.26 | 7.83  | 8.23  | 8.93  |
> |                   | EnzymeFlow    | 2.87 | 3.18 | 3.62 | 0.44 | 0.43 | 0.39 | 0.74  | 1.07  | 1.78  |
> | ----------------- | ------------- | ---- | ---- | ---- | ---- | ---- | ---- | ----- | ----- | ----- |
> | EC6               | RFDiffusionAA | 3.43 | 3.69 | 4.12 | 0.30 | 0.29 | 0.26 | 19.08 | 19.73 | 21.21 |
> |                   | EnzymeFlow    | 3.14 | 3.32 | 3.73 | 0.43 | 0.40 | 0.36 | 1.23  | 1.45  | 2.56  |
> ```

---

> ### Author Response · Authors · 2024-11-13
> **Answer to Question Part2**
>
> ## A3:
> #### You are correct that the explicit dynamic behavior of proteins and their ligands is not fully captured by current models, including ours. However, we address reaction dynamics in a way that we believe implicitly captures this aspect. Specifically, we model the transition from substrates to products using co-MSAs and atom-to-atom cross-attention, which we believe can capture dynamic changes in the reaction process.
>
> #### Previous work, such as ReactZyme [2], has demonstrated that the atom-to-atom cross-attention mechanism can effectively model the transitions between substrates and products. While this is not an explicit dynamic model like CLIPzyme [1], which uses transition graphs to model reaction dynamics, ReactZyme’s use of the cross-attention mechanism has shown superior performance in implicitly capturing these dynamics compared to explicit transition graph models.
>
> #### In addition, we evaluate the model's ability to implicitly learn dynamics by measuring substrate specificity (Km), and our results indicate that the model produces substrate specificity values close to the ground truth. This suggests that the model is effectively capturing not only static interactions but also the dynamic aspects of enzyme catalysis, even though these dynamics are not explicitly modeled.
>
> #### We acknowledge that the field of using AI to model reaction transitions and chemical dynamics is still emerging, and as part of our ongoing work, we are exploring additional methods to explicitly model these transitions. For example, we are investigating the identification of reaction centers and assigning atomic weights to regions of the substrate involved in the reaction, with atoms in the reaction center receiving higher attention during the transition. We believe that combining these approaches with attention mechanisms could allow our model to implicitly and more explicitly learn dynamic processes in enzymatic reactions.
>
> #####  [1] Mikhael, P.G., Chinn, I. and Barzilay, R., 2024. CLIPZyme: Reaction-Conditioned Virtual Screening of Enzymes. arXiv preprint arXiv:2402.06748.
> #####  [2] Hua, C., Zhong, B., Luan, S., Hong, L., Wolf, G., Precup, D. and Zheng, S., 2024. Reactzyme: A benchmark for enzyme-reaction prediction. arXiv preprint arXiv:2408.13659.
>
>
> ## A4:
> ####  We believe the unique advantages of co-MSAs over standard protein MSAs are critical to understanding the performance of our model.
>
> ####  Standard protein MSAs focus on aligning sequences from similar enzymes, but they do not account for the rich, reaction-specific context that is crucial for catalytic site design. In contrast, co-MSAs are explicitly designed to model enzyme-reaction interactions and catalytic specificity. One of the key insights from our work is that SMILES strings (representing reactions) can vary significantly even among enzymes that are similar in sequence within a standard MSA. This variation is largely due to the different reactions catalyzed by these enzymes, which highlights the importance of incorporating reaction-specific data in the alignment process.
>
> ####  From a training signal perspective, co-MSAs offer a unique advantage: they can distinguish between enzymes that share similar sequences but catalyze different reactions. This is a critical feature that standard MSAs lack, as they treat enzyme sequences in isolation, without considering the reaction they catalyze. Co-MSAs allow us to learn finer details of enzyme function by capturing the relationship between enzyme sequences and their catalytic behavior, which leads to more accurate predictions and better modeling of enzyme function.
>
> #### Moreover, we are actively expanding our use of co-MSAs with coMSAformer for few-shot learning, where we explore how protein mutations affect enzyme-reaction relationships. This extension aims to enhance our ability to predict how specific mutations may alter catalytic behavior, further demonstrating the power of co-MSAs to capture the nuanced relationship between enzymes and their catalytic roles.

---

> ### Author Response · Authors · 2024-11-13
> **Answer to Question Part3**
>
> ## A5:
> #### For results with a vanilla MultiFlow baseline, we trained an ablation model that incorporates only reaction conditions, but excludes additional elements such as co-MSAs and EC conditions. This model, which we refer to as EnzymeFlow-ablation, was trained and evaluated using the 64-residue training and sampling approach on the EnzymeFill dataset.
>
> #### The performance of the EnzymeFlow-ablation model is compared to the full EnzymeFlow model, which incorporates co-MSAs, EC conditions, and reaction conditioning. The results show that the EnzymeFlow model outperforms the EnzymeFlow-ablation model, as evidenced by the following metrics: lower RMSD, higher alignment scores, lower embedding distance (we use ESM3 encoding for pocket sequences and structures)
>
> #### These results demonstrate that the inclusion of co-MSAs and EC conditions provides significant improvements in the generation of enzyme catalytic pockets, highlighting the importance of these additional elements in improving both the structural and functional accuracy of the model.
> ```
> |                   |                    | rmsd | rmsd | rmsd | tm   | tm   | tm   | Dist  | Dist  | Dist  |
> | ----------------- | ------------------ | ---- | ---- | ---- | ---- | ---- | ---- | ----- | ----- | ----- |
> | Pocket Evaluation |                    | Top1 | Top3 | Mean | Top1 | Top3 | Mean | Top1  | Top3  | Mean  |
> | ----------------- | ------------------ | ---- | ---- | ---- | ---- | ---- | ---- | ----- | ----- | ----- |
> | EC1               | EnzymeFlow-abation | 3.76 | 3.90 | 4.19 | 0.29 | 0.28 | 0.26 | 19.82 | 20.29 | 21.48 |
> |                   | EnzymeFlow         | 1.88 | 2.09 | 2.41 | 0.67 | 0.65 | 0.59 | 0.25  | 0.58  | 1.54  |
> | ----------------- | ------------------ | ---- | ---- | ---- | ---- | ---- | ---- | ----- | ----- | ----- |
> | EC2               | EnzymeFlow-abation | 4.02 | 4.19 | 4.35 | 0.23 | 0.23 | 0.21 | 50.31 | 52.33 | 54.15 |
> |                   | EnzymeFlow         | 3.65 | 3.80 | 4.09 | 0.33 | 0.31 | 0.28 | 1.03  | 1.85  | 4.40  |
> | ----------------- | ------------------ | ---- | ---- | ---- | ---- | ---- | ---- | ----- | ----- | ----- |
> | EC3               | EnzymeFlow-abation | 3.46 | 3.69 | 4.12 | 0.31 | 0.29 | 0.27 | 9.84  | 10.02 | 11.14 |
> |                   | EnzymeFlow         | 3.18 | 3.32 | 3.84 | 0.39 | 0.36 | 0.32 | 0.42  | 1.08  | 2.19  |
> | ----------------- | ------------------ | ---- | ---- | ---- | ---- | ---- | ---- | ----- | ----- | ----- |
> | EC4               | EnzymeFlow-abation | 3.31 | 3.48 | 3.91 | 0.32 | 0.30 | 0.27 | 18.02 | 18.56 | 19.96 |
> |                   | EnzymeFlow         | 2.58 | 2.73 | 2.96 | 0.56 | 0.54 | 0.51 | 0.50  | 1.07  | 2.43  |
> | ----------------- | ------------------ | ---- | ---- | ---- | ---- | ---- | ---- | ----- | ----- | ----- |
> | EC5               | EnzymeFlow-abation | 3.48 | 3.66 | 4.06 | 0.29 | 0.28 | 0.26 | 13.71 | 14.01 | 14.81 |
> |                   | EnzymeFlow         | 2.87 | 3.18 | 3.62 | 0.44 | 0.43 | 0.39 | 0.74  | 1.07  | 1.78  |
> | ----------------- | ------------------ | ---- | ---- | ---- | ---- | ---- | ---- | ----- | ----- | ----- |
> | EC6               | EnzymeFlow-abation | 3.78 | 3.89 | 4.33 | 0.29 | 0.27 | 0.26 | 21.00 | 22.87 | 24.19 |
> |                   | EnzymeFlow         | 3.14 | 3.32 | 3.73 | 0.43 | 0.40 | 0.36 | 1.23  | 1.45  | 2.56  |
> ```

---

> ### Author Response · Authors · 2024-11-13
> **Answer to Question Part4**
>
> ## A6:
> #### We provide RMSD and alignment scores using 64-residue training-sampling strategy for generated pockets against the entire enzyme.
> ```
> |                   |                    | rmsd | rmsd | rmsd | tm   | tm   | tm   |
> | ----------------- | ------------------ | ---- | ---- | ---- | ---- | ---- | ---- |
> | Pocket Evaluation |                    | Top1 | Top3 | Mean | Top1 | Top3 | Mean |
> | ----------------- | ------------------ | ---- | ---- | ---- | ---- | ---- | ---- |
> | EC1               | EnzymeFlow-abation | 3.15 | 3.37 | 3.85 | 0.41 | 0.39 | 0.36 |
> |                   | RFDiffusionAA      | 3.09 | 3.27 | 3.69 | 0.50 | 0.46 | 0.41 |
> |                   | EnzymeFlow         | 2.10 | 2.36 | 3.14 | 0.69 | 0.65 | 0.53 |
> | ----------------- | ------------------ | ---- | ---- | ---- | ---- | ---- | ---- |
> | EC2               | EnzymeFlow-abation | 3.50 | 3.65 | 4.13 | 0.40 | 0.38 | 0.34 |
> |                   | RFDiffusionAA      | 2.96 | 3.18 | 3.62 | 0.49 | 0.46 | 0.42 |
> |                   | EnzymeFlow         | 2.83 | 3.24 | 3.68 | 0.45 | 0.42 | 0.38 |
> | ----------------- | ------------------ | ---- | ---- | ---- | ---- | ---- | ---- |
> | EC3               | EnzymeFlow-abation | 3.52 | 3.63 | 4.03 | 0.43 | 0.40 | 0.36 |
> |                   | RFDiffusionAA      | 2.53 | 2.90 | 3.52 | 0.53 | 0.50 | 0.43 |
> |                   | EnzymeFlow         | 2.83 | 3.21 | 3.72 | 0.46 | 0.43 | 0.38 |
> | ----------------- | ------------------ | ---- | ---- | ---- | ---- | ---- | ---- |
> | EC4               | EnzymeFlow-abation | 3.18 | 3.32 | 3.82 | 0.44 | 0.41 | 0.37 |
> |                   | RFDiffusionAA      | 2.76 | 3.09 | 3.50 | 0.50 | 0.48 | 0.43 |
> |                   | EnzymeFlow         | 2.47 | 2.68 | 2.98 | 0.61 | 0.58 | 0.54 |
> | ----------------- | ------------------ | ---- | ---- | ---- | ---- | ---- | ---- |
> | EC5               | EnzymeFlow-abation | 3.22 | 3.45 | 3.90 | 0.43 | 0.40 | 0.36 |
> |                   | RFDiffusionAA      | 2.70 | 3.06 | 3.60 | 0.50 | 0.47 | 0.43 |
> |                   | EnzymeFlow         | 3.04 | 3.29 | 3.61 | 0.52 | 0.49 | 0.44 |
> | ----------------- | ------------------ | ---- | ---- | ---- | ---- | ---- | ---- |
> | EC6               | EnzymeFlow-abation | 3.25 | 3.43 | 3.80 | 0.43 | 0.41 | 0.37 |
> |                   | RFDiffusionAA      | 3.16 | 3.45 | 3.86 | 0.49 | 0.46 | 0.39 |
> |                   | EnzymeFlow         | 2.70 | 3.08 | 3.52 | 0.50 | 0.47 | 0.43 |
> ```

---

> ### Comment · Reviewer_yZeM · 2024-11-25
> **Comments do not address concerns, and add additional uncertainty to the results**
>
> Thank you for the additional clarifications and experiments provided in your response. I appreciate the effort invested in addressing the points raised. However, I still find that some of the core concerns remain unresolved, particularly regarding the functional validity of the generated active sites. Below are my specific comments and suggestions:
> 1. Functional Relevance of Generated Active Sites: While the idea of generating active sites conditioned on reactions is compelling and significant, the evaluation metrics used primarily focus on structural similarity. Given the sensitivity of enzymatic function to even slight changes in active site residues or geometry, structural similarity alone is not sufficient to establish functional relevance. I recognize that there are limited solutions to this problem (as is often the case with generative models), but the results are so close to other approaches and the structural similarity is so low (<0.3 tm score) that it is hard to attribute your approach as a solution without additional information.
>
> 1.1. The additional metrics you provided (e.g., pH, kcat, and Km scores) are a step in the right direction. However, since you don't tell us how you got these metrics, I can only assume they are derived from predictive models, many of which are known to have substantial limitations (e.g.,https://pubmed.ncbi.nlm.nih.gov/39346751/). This raises concerns about the reliability of these results. Could you provide more information on how these metrics were obtained and offer alternative metrics or analyses that more directly indicate catalytic function?
>
> 1.2. You do provide EC function annotation results that are extremely low: "accuracy (0.2809), precision (0.2600),recall (0.2722), and F1 score (0.2504)". These results suggest significant limitations in generating functionally accurate enzymes. Is there an alternative way to interpret these results? Could they indicate issues with either the generation process or the downstream evaluation?
>
> 2. Statistical Significance of Results: top-1 or top-K evaluations are valuable but distinct from confidence intervals. To establish the robustness of your approach, please provide evidence that the reported improvements are statistically significant. This is critical to substantiate claims about the effectiveness of the proposed contributions.
>
> 3. Impact of Increasing Pocket Size: The reported performance improvement when expanding the training context to 64 residues is puzzling. This raises the question of whether longer sequences inherently lead to better alignment scores rather than improvements in active site generation. Could you clarify why this change improves results and provide substantial evidence to corroborate your hypothesis? Additionally, the claim of the paper is to generate functional active sites, which typically consist of only 2-6 residues. How do 64-residue regions align with this claim?
>
> 4. "However, upon expanding to a 64-residue training-sampling strategy, we observed a statistically significant performance improvement that better reflects the impact of these contributions." please provide proof (re significant) and additionally explain which performance you are referring to.
>
> 5. Size of the Test Set: Given the size of the training set (53,483 enzyme-reaction pairs), the test set size appears disproportionately small (100 reactions). Evaluating the model on a broader range of reactions would greatly enhance confidence in its generalizability. Even if EC7 reactions are excluded, results on a larger subset of other EC classes could provide valuable insights. Especially since you do not provide these results for the 64-residue crops already.
> 6. As highlighted by other reviewers, you do not actually do conditional generation, but rather a multi-task prediction. Please correct this error in the paper if this paper is accepted.
>
> Lastly, I want to emphasize that both I and, in my opinion, the other reviewers recognize the significance of this problem and exactly what you have done. Successfully addressing this challenge would indeed be transformative for enzyme design. Our feedback is not intended to dismiss your work but to ensure the results presented are both meaningful and robust. We are not claiming that EnzymeFlow is simply a "machine learning toy" as you put it. Experimental validation might be one possible avenue to strengthen the claims, though not the only path. I encourage you to consider alternative ways to demonstrate functional relevance.
>
> The claim that EnzymeFlow can generate functional active sites is bold and impactful but remains unproven in its current state.

---

> > ### Author Response · Authors · 2024-11-25
> > **oh thanks!**
> >
> > We really appreciate the feedback! You are the best!

---

### Meta-Review · Area_Chair_zzgk · 2024-12-17

**Metareview:**

This paper seems to have several significant weaknesses. The most important are
- The unclear role of the EC class as a conditioning factor. As mentioned by some reviewers, the actual source code suggests that  MSA and EC class appear more like two multi-task learning tasks, which also raises questions about the extent to which these "conditional" features can have a substantial influence on the backbone design.
- The unclear role and importance of a rather general concept such as cross attention for modeling details in chemical reactions.
- The unclear relevance of experimental results for predicting enzyme functionality.
- The unclear statistical significance of several results.
While some of these concerns could be addressed to some degree in the authors' rebuttal, in the end just too many open questions remained. Therefore, I conclude that this paper in its present form is not ready for publication, and I recommend rejection of this paper.

**Additional Comments On Reviewer Discussion:**

-  unclear role of the EC class as a conditioning factor: this concern could not be addressed in the rebuttal
-  unclear role and importance of a rather general concept such as cross attention for modeling details in chemical reactions: still an open question to me.
 -  unclear relevance /significance of experimental results. The authors posted several additional results, but it was difficult to understand their precise meaning.
Further, I'd like to mention that not all comments in the rebuttal were professional and respectful.

---

### Decision · Program_Chairs · 2025-01-22

Reject